# Domesticated cannabinoid synthases amid a wild mosaic cannabis pangenome

Ryan C. Lynch[1,8 ✉], Lillian K. Padgitt-Cobb[1,8 ✉], Andrea R. Garfinkel[2], Brian J. Knaus[3], Nolan T. Hartwick[1], Nicholas Allsing[1], Anthony Aylward[1], Philip C. Bentz[4], Sarah B. Carey[4], Allen Mamerto[1], Justine K. Kitony[1], Kelly Colt[1], Emily R. Murray[1], Tiffany Duong[1], Heidi I. Chen[1], Aaron Trippe[2], Alex Harkess[4], Seth Crawford[2], Kelly Vining[3] & Todd P. Michael[1,5,6,7 ✉]

*Cannabis sativa* is a globally important seed oil, fibre and drug-producing plant species. However, a century of prohibition has severely restricted development of breeding and germplasm resources, leaving potential hemp-based nutritional and fibre applications unrealized. Here we present a cannabis pangenome, constructed with 181 new and 12 previously released genomes from a total of 144 biological samples including both male (XY) and female (XX) plants. We identified widespread regions of the cannabis pangenome that are surprisingly diverse for a single species, with high levels of genetic and structural variation, and propose a novel population structure and hybridization history. Across the ancient heteromorphic X and Y sex chromosomes, we observed a variable boundary at the sex-determining and pseudoautosomal regions as well as genes that exhibit male-biased expression, including genes encoding several key flowering regulators. Conversely, the cannabinoid synthase genes, which are responsible for producing cannabidiol acid and delta-9-tetrahydrocannabinolic acid, contained very low levels of diversity, despite being embedded within a variable region with multiple pseudogenized paralogues, structural variation and distinct transposable element arrangements. Additionally, we identified variants of acyl-lipid thioesterase genes that were associated with fatty acid chain length variation and the production of the rare cannabinoids, tetrahydrocannabivarin and cannabidivarin. We conclude that the *C. sativa* gene pool remains only partially characterized, the existence of wild relatives in Asia is likely and its potential as a crop species remains largely unrealized.

Cannabis (*C. sativa* L., cannabis) is an ancient domesticated plant with widespread archaeological evidence for seed (achene) and fibre utilization dating to 8,000 years ago in East Asia, and earlier occurrences found up to 12,000 years ago[1,2], rivalling that of important crops such as wheat, barley, maize and rice. Cannabis was originally a multipurpose crop in Asia, where the same plants were utilized as a source of fibre, food and drugs[2,3]. Over time, cannabis spread globally and single or dual-use-type cultivars were developed, eventually giving rise to divergent hemp and drug-type populations of the twentieth century[4]. Prior to the early 1900s, cannabis was an important commodity across Asia, Europe and the New World, and was used to produce fibres used in sails, ropes, clothing and paper. However, competition from other fibre crops, entanglement with drug laws, and the eventual development of synthetic fibres led to a decline in production. In recent decades, the use of cannabis has shifted to specialized applications, including niche seed oils and drug production, where it continues to hold significant economic and cultural importance today[5].

Throughout history and around the world, cannabis has undergone cycles of "cultivation, consumption, and crackdown"[6]. Modern prohibition originated in the USA during the early twentieth century[7], but by 1961 had spread to a majority of countries[8]. Prohibition eliminated the fibre and food uses of cannabis for decades, but gave rise to a high-value illegal market for phytocannabinoid-based drugs, which are derived from glandular trichomes. Although more than 100 phytocannabinoids have been identified, only a limited number are produced in significant quantities, which are used to classify plants by chemotype: delta-9-tetrahydrocannabinolic acid (THCA; type I), cannabidiolic acid (CBDA; type III), balanced CBDA and THCA (type II), cannabigerolic acid (CBGA; type IV) and cannabinoid-free (type V)[9]. Although tetrahydrocannabinol (THC), the primary intoxicant, remains a controlled substance, a majority of US states and many countries now allow medical or adult use of cannabis products. Separately, the 2014 and 2018 US Farm Bills facilitated hemp production and research in plants that produce less than 0.3% THC on US soil,

[1]The Plant Molecular and Cellular Biology Laboratory, The Salk Institute for Biological Studies, La Jolla, CA, USA. [2]Oregon CBD, Independence, OR, USA. [3]Department of Horticulture, Oregon State University, Corvallis, OR, USA. [4]HudsonAlpha Institute for Biotechnology, Huntsville, AL, USA. [5]Department of Cell and Developmental Biology, School of Biological Sciences, University of California San Diego, La Jolla, CA, USA. [6]Science and Conservation, San Diego Botanical Garden, Encinitas, CA, USA. [7]Center for Marine Biotechnology and Biomedicine, University of California San Diego, La Jolla, CA, USA. [8]These authors contributed equally: Ryan C. Lynch, Lillian K. Padgitt-Cobb. ✉e-mail: rlynch@colorado.edu; lilliankpc@gmail.com; toddpmichael@gmail.com

generating opportunities for improved non-THC drug, grain and fibre applications.

The haploid cannabis genome is relatively small in size (around 750 Mb), yet its complexity is driven by a high proportion (approximately 79%) of transposable elements (TEs) and substantial heterozygosity (single nucleotide polymorphisms (SNPs): greater than 2%). The CBDRx (cs10) reference genome, derived from the high-cannabinoid (HC) cannabidiol (CBD) hemp lineage related to the well-known anti-epileptic 'Charlotte's web' cultivar[10], resolved the arrangement of cannabinoid synthase genes as a single full-length copy of *CBDAS* nested within conserved 70 to 80-kb tandem TE arrays. Furthermore, HC hemp lines such as CBDRx emerged through the introgression of the *CBDAS* locus into a predominantly marijuana (MJ) genetic background, thereby leveraging high-potency alleles to enhance CBD production[11]. However, initial comparison of published cannabis genomes suggests substantial genomic dynamism across use types[11-16], raising key unresolved questions about the global extent of genetic diversity. Additionally, the role of hybridization in shaping genome architecture and allele transmission remains unclear, highlighting the need for further high-quality assemblies and population-scale genomic analyses. Here we have built a comprehensive framework for exploring genetic diversity in this multi-use crop by creating a cannabis pangenome using haplotype-resolved, chromosome-scale assemblies.

## The cannabis pangenome

Cannabis is often classified as a monospecific genus[17], although debate remains regarding the status of *Cannabis indica* Lam. and *Cannabis ruderalis*, the latter of which is thought to be the source of the day-neutral (DN; autoflowering) flowering type[18]. We addressed the diversity of cannabis by building the pangenome with samples selected from multiple sources to cover use types, history, sex expression and agronomic traits (Extended Data Fig. 1 and Supplementary Fig. 1). The cannabis pangenome comprises 181 new PacBio assemblies and 12 previously published genomes, representing 144 biological samples, including 78 haplotype-resolved, chromosome-scale assemblies and 103 contig-level assemblies. We highlight an $F_1$ hybrid (ERBxHO40_23; EH23) between two phenotypically and genetically divergent parents to clarify features of the genome that have been missed in previous studies (Fig. 1a, Extended Data Figs. 2 and 3 and Supplementary Note 1).

All genomes are of high quality, with an average N50 of 7.5 Mb, and BUSCO[19] genome and proteome completeness scores of 97% and 95%, respectively (Extended Data Fig. 4). The average haploid genome length was 781 Mb with around 35,000 protein-coding genes per genome (Supplementary Tables 1, 2 and 3). Consistent with a predominantly outcrossing behaviour, the SNP-based heterozygosity ranged between 1% and 2.5% (Supplementary Fig. 2). The assemblies are also high quality structurally, resolving previous TE placement issues (Supplementary Fig. 3) and revealing centromere regions, telomere length, large structural variations (SVs), fine-scale genetic architecture of important genes such as the cannabinoid synthases, as well as the sex-determining region (SDR) and pseudoautosomal region (PAR) of the Y chromosome (Fig. 1a,b), the largest chromosome in the *Cannabis* genome (Extended Data Fig. 5).

We constructed comprehensive *Cannabis* pangenomes using both reference-based and reference-free approaches. A reference-based pangenome graph was generated with Minigraph-Cactus (MGC)[20] using the 78 chromosome-scale, haplotype-resolved genomes. For a reference-free approach, we built a *k*-mer matrix with PanKmer[21] using all 193 genomes and a graph-based representation with PanGenome Graph Builder (PGGB)[22]. Owing to the high memory demands of PGGB, we selected a subset of 16 genomes for graph generation (Extended Data Fig. 6 and Methods). SVs detected by MGC and PGGB closely matched those from pairwise whole-genome alignments. Mapping rates for a diverse short-read dataset[2] were similar between the MGC pangenome graph (95.09%) and the linear EH23a reference

genome (95.0%), indicating that both approaches effectively captured variation.

## The pangenome reveals five populations

The taxonomy, history and nomenclature of the cannabis genus have long been debated[23]. Owing to its wide phenotypic and geographic diversity, it has been classified either as a multi-species interbreeding complex or as a single species with subspecies designations. We calculated the collector's curve to evaluate the completeness and diversity of the pangenome using shared gene-based orthogroups as well as shared *k*-mers (Fig. 1c,d). The curve suggested that we captured the majority of cannabis orthogroup diversity at around 100–125 genomes (Fig. 1c), although significant global genomic variation remains uncharacterized (Fig. 1d), possibly owing to the recent TE activity. Collector's curves for the 78 haplotype-resolved, chromosome-scale assemblies revealed similar but more attenuated diversity–sample relationships (Supplementary Fig. 4). Across all pangenome samples we found that 23% of genes were 'core' (present in all genomes), 55% were 'nearly-core' (95–99% of genomes), 21% were 'shell' (5–94% of genomes), and a small fraction were classified as 'cloud' (0.4%) or 'unique' (0.7%) (Fig. 1e and Supplementary Fig. 5). Gene Ontology (GO) terms related to terpene biosynthesis and defence response were some of the most frequently enriched among core genes (Extended Data Fig. 7, Supplementary Note 2 and Supplementary Table 4), although both showed substantial variation at the sequence level (Extended Data Figs. 7 and 8).

Cannabis has not undergone a whole-genome duplication since the ancient lambda event approximately 100 million years ago[13]. This suggests that its extensive genomic diversity arose not through recent whole-genome duplications or hybridization-driven allopolyploidy, but through tandem gene duplication and other local duplication mechanisms (Supplementary Fig. 6 and Supplementary Note 3). Comparisons between populations using pairwise average $F_{st}$ (fixation index) values based on phased SNPs indicated that some cannabis populations exhibited levels of genetic differentiation that were similar to interspecies comparisons, such as in strawberry[24] ($F_{st} = 0.20$ for MJ versus hemp; Supplementary Table 5). Specific genes with high $F_{st}$ SNPs were linked to environmental response, with circadian, light signalling and flowering time genes exhibiting an above-average $F_{st}$ (0.42) (Supplementary Table 6). Notably, *GIGANTEA* (*GI*)[25], a highly conserved, typically single-copy gene that has a central role in the circadian clock that regulates daily period length, flowering time and cell elongation, contained a SNP with the fifth-highest $F_{st}$ (0.77, MJ versus hemp). Separately, using a test for selective sweeps across 20-kb SNP windows (XP-CLR, MJ versus hemp), *GI* was again found within a significant region of the X chromosome. Finally, a broader analysis of gene family diversity revealed substantial variation at the *GI* locus between the HC hemp and hemp populations (Supplementary Fig. 7). These findings highlight the effect of selection on key agronomic genes[26] that may underlie differentiation of traits such as flowering and internode elongation (fibre length), which contrast markedly between hemp and MJ populations.

Drug-type populations from North America that produce high levels of cannabinoids are thought to have originated from regions of Southeast and Central Asia, and were brought to the western hemisphere via the Caribbean and South America; however, most of what is known about these ancestral populations is based on limited historical accounts and speculation[5]. A broad split of drug-type samples into two groups, one aligned with Asian hemp and one with European hemp, was suggested by the *k*-mer-based hierarchical clustering using the PanKmer pangenome (Fig. 1f,g and Extended Data Fig. 1). Both groups contained MJ and HC hemp samples, which were thought to have largely MJ ancestry with a recent history of introgression breeding for *CBDAS* genes, perhaps from European hemp origins[11]. However, using a phased SNP-based structure with all MJ samples treated as a single population, the TreeMix model inferred a highest-likelihood

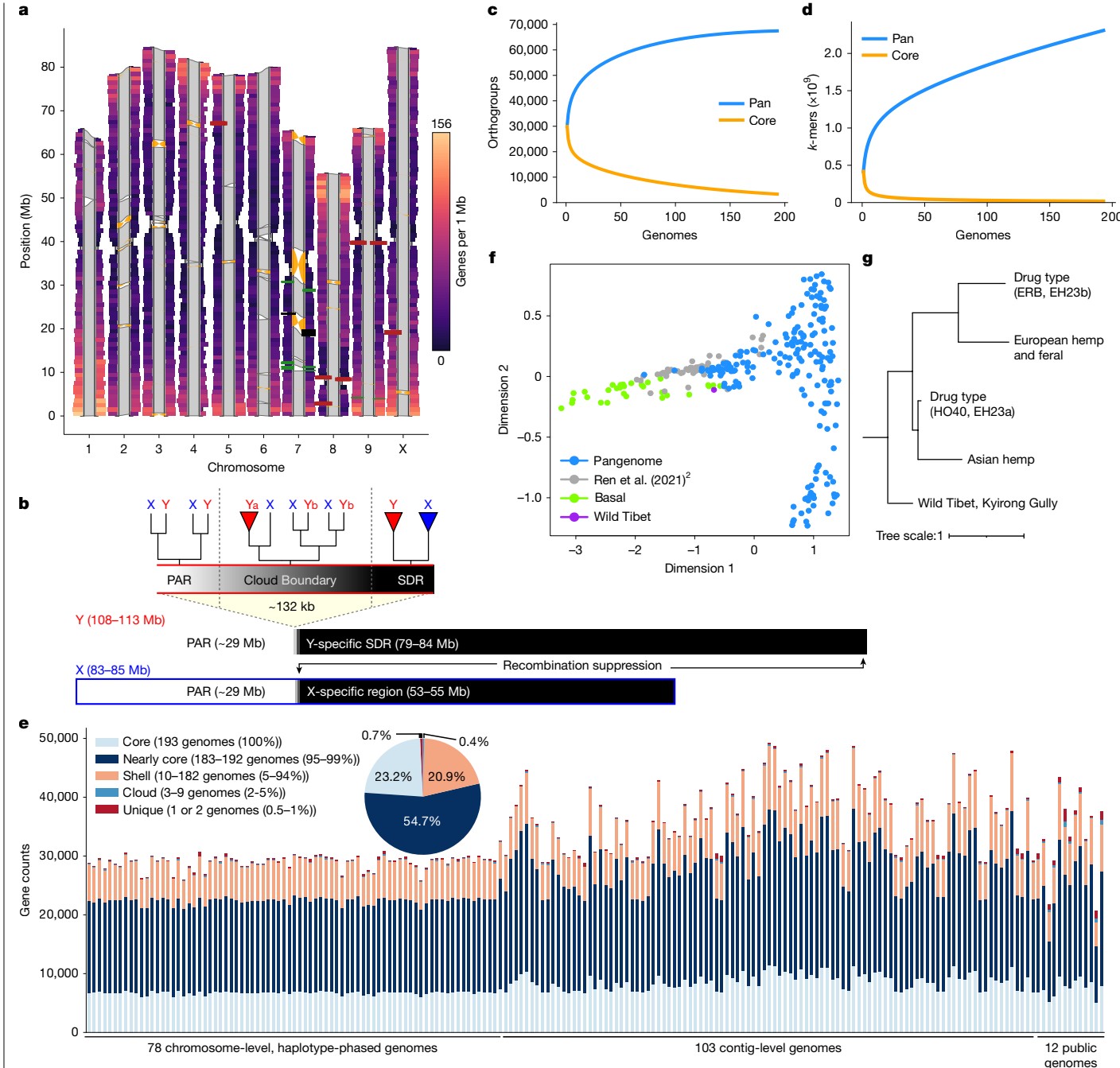

**Fig. 1 | Cannabis pangenome architecture uncovers at least five populations.** **a**, Genomic features of ten chromosome pairs in EH23. One million base pair rectangular windows extend from each haplotype at a width proportional to the absence of the CpG motif (high CpG content in centromeric and telomeric regions shown as constrictions). Each rectangular window is coloured by gene density, with warm colours indicating high gene density and cool colours indicating low gene density. Each haplotype pair is connected by polygons indicating structural arrangement, with grey for syntenic regions and orange connecting inversions. Rectangles along each haplotype indicate select loci, including 45S (26S, 5.8S and 18S) RNA arrays (firebrick red), 5S RNA arrays (black) and cannabinoid synthases (forest green). **b**, Summary of sex chromosomes based on haplotype-resolved XY assemblies[29,34]. Phylogenetic analysis of XY homologues revealed variation in SDR-linked versus PAR-linked genes on the Y chromosome, as indicated by a clade of Y-linked homologues (Ya) versus a clade containing both X- and Y-linked homologues (Yb), respectively. Tip triangles indicate collapsed monophyletic clades of X or Y homologues. The X-specific region does not undergo recombination with the Y chromosome (although it undergoes recombination in XX females). **c**, Collector's curve using shared gene orthogroup membership. **d**, Collector's curve using shared 31-mers. **e**, Gene membership across all pangenome samples. **f**, Hierarchical clustering of Jaccard similarity scores based on 31-mers reveals a structure of at least 5 groups in the pangenome. Each drug-type group contains both MJ and HC hemp samples (Supplementary Table 1). Scale bar represents distance of maximum Jaccard dissimilarity. **g**, 31-mer-based multidimensional scaling plot of all pangenome assemblies (blue), Wild Tibet assembly (purple) and global diversity panel of short-read samples[2] (green, 'basal' population from Asia; grey, other populations).

phylogeny that included six gene flow (migration) events between Asian hemp, HC hemp and European hemp, as well as MJ and HC hemp samples (Supplementary Fig. 8). These results may partially explain the European and Asian groupings of drug-type samples found by our

k-mer clustering analysis, and reflect the effects of historical hybridization breeding between Asian and European hemp that is documented in the breeding literature[27]. In addition to the two drug-type populations and separate European and Asian hemp populations, the k-mer

clustering showed significant divergence between the single available wild Tibetan assembly from all other domesticated and feral lines,[13] suggesting that wild *Cannabis* relatives still exist in remote regions of Asia[2]. Indeed, *k*-mer based hierarchical clustering of the pangenome assemblies combined with short reads from samples collected across Europe and Asia recapitulated the original authors' finding that samples from Asia described as 'drug-type feral' and 'basal' represent distinct populations[2] (Fig. 1f and Supplementary Fig. 9). Ultimately, refining hypotheses about domestication, biogeography and use-type history will require broader sampling of Asian and historical specimens, along with careful delineation of wild and feral populations.

## Sex chromosome evolution

Sex expression in cannabis has long puzzled biologists[28]. Although most populations are dioecious, with separate male (XY) and female (XX) plants, monoecious (XX) forms also exist, which exhibit variable ratios of male and female flowers. The Cannabaceae sex chromosomes originated in a common ancestor of *Cannabis* and *Humulus* more than 36 million years ago (Ma)[29]—earlier than previous estimates[30]—making them among the oldest known in flowering plants[31]. Despite their ancient origin, cannabis sex chromosomes have been shaped by human selection on sexually dimorphic traits[32]. In drug-type populations, males produce few glandular trichomes, and pollination reduces cannabinoid yield in female plants, leading to reduced use (or elimination) of males in breeding programmes (Methods). By contrast, hemp seed production requires pollen, and male plants enhance bast fibre yield and quality. Additionally, European monoecious fibre cultivars, such as Santhica (SAN) and KC Dora (KCDv1), were developed to improve mechanized harvesting efficiency of both fibre and seeds, adding another layer of artificial selection[31].

Unlike most angiosperms, cannabis has a heteromorphic XY pair, with a Y chromosome that is approximately 30% larger than the X chromosome (Fig. 1b, Extended Data Figs. 4 and 5). Recombination occurs in the PAR but is suppressed throughout the SDR on the Y chromosome. The SDR spans 79–84 Mb out of the approximately 110 Mb Y chromosome, making it one of the largest SDRs in plants, with 840–1,160 genes (Supplementary Figs. 10 and 11 and Supplementary Tables 7 and 8). By contrast, the PAR covers only around 29 Mb, yet hosts 1,900–1,980 genes, including many important flowering genes, such as *FLOWERING LOCUS T* (*FT*), *CONSTANS* (*CO*) and *GI*. Theory predicts that after initial recombination suppression, the SDR expands in a stepwise manner owing to selection linking genes to the SDR that are beneficial to males but deleterious to females[33]. Alternatively, neutral processes, reflected in synonymous substitution rates ($K_s$), can drive SDR expansions. $K_s$ values along the SDR showed a continuous pattern of gene addition from the PAR boundary to the centromere[29], suggesting that recombination suppression near the centromere at least partially caused expansion. Using *k*-mers and X–Y orthologue phylogenies, we identified two distinct SDR haplotypes: Ya, shared by six samples, and Yb, found in two samples (Fig. 1b). These haplotypes differed at the SDR–PAR boundary, separated by 5 conserved gene models spanning approximately 51 kb (GVA-21-1003-002)[34] to 132 kb (Kompolti), with all others spanning 61–62 kb. The gene located nearest to the PAR–SDR boundary in the Ya haplotype (within the PAR in Yb; Fig. 1b) is *TRANSCRIPTION ELONGATION FACTOR* (*SPT5*), which is known to interact with *FLOWERING LOCUS C* (*FLC*) via *FRIGIDA* during cold-induced flowering in *Arabidopsis*[35]. This suggests that selection on flowering time genes has facilitated a stepwise shift in recombination suppression and SDR expansion, which may explain why male flower development begins before female flowering onset in some varieties. Polymorphisms in the SDR–PAR boundary signal that the hemp gene pool hosts ancestral diversity of sexually antagonistic genes, which may underlie useful variation in flowering timing[36].

Furthermore, gene expression profiling of Ace High (AH3M) male and female tissue found biased expression of more than 7,000 genes in male flowers across all chromosomes, spanning many functions including pollen development. This contrasted with biased expression of genes in male leaf (approximately 1,400 genes), female leaf (approximately 3,700 genes) and female flower (approximately 3,900 genes) tissue (Extended Data Fig. 9). Whereas gene expression in the X chromosome was fairly uniform, gene density and expression in the Y chromosome were skewed toward the PAR. Of note, a substantial proportion of genes in the PAR (38%, around 750 genes) showed biased expression in male flowers, compared with only 6% (94 genes) in the SDR. Although the SDR encodes one or more unidentified sex-determining genes for male flower development, the majority of the required transcriptional network for male or female flower expression is broadly distributed across all chromosomes.

## TEs shape the pangenome

TEs had a major role in shaping the cannabis genome, particularly in the proliferation of intronless cannabinoid synthase genes, which are embedded within 70–80-kb conserved TE cassettes[11]. On average, TEs comprised 68% of each genome, with long terminal repeat retrotransposons (LTR-RTs) representing 50% of the total (Fig. 2a and Supplementary Tables 1 and 9). Genes on average were located near TEs (443–613 bp from TEs; Supplementary Table 10). Different TE types showed distinct insertion patterns: DNA transposons (for example, Mutator and Helitron) were inserted within 500 bp upstream of coding regions, whereas LTR-RTs were more evenly distributed flanking genes (Supplementary Fig. 12). Genes involved in transposition, transcription, recombination and DNA repair were frequently associated with Ty3-long terminal repeats (LTRs), whereas defence and metabolite biosynthesis genes were enriched near Ty1-LTRs (Supplementary Table 11). Many intact TEs were estimated to have inserted into the genome within the past 100,000 years, suggesting that ongoing diversification may be driven by hybridization and stress factors, particularly in $F_1$ and MJ populations (Fig. 2a–c). One such factor is clonal propagation, which is a common practice in modern MJ production but is rarely used in hemp field cultivation.

Despite 4 million years of sustained activity and a recent burst of LTR proliferation (Fig. 2b,c), the cannabis genome has maintained a smaller haploid genome size (approximately 750 Mb) than that of its sister genus *Humulus*, which ranges from 1,700 Mb in *Humulus japonicus* to 2,700 Mb in *Humulus lupulus*[37]. Solo LTRs reflect genome purging and can be formed by ectopic recombination, which occurs in the internal sequence of a complete LTR-RT[38]. The high solo:intact ratio observed in cannabis (Fig. 2d–g) is likely to contribute to its compact genome size by mitigating TE accumulation. Ty1-LTRs displayed the highest solo:intact ratio within the SDR of the Y chromosome (Fig. 2d,f), suggesting the initial expansion of this region was driven by TE insertions that preceded deletion events by ectopic recombination (Fig. 2i,j). DNA methylation also prevents uncontrolled TE proliferation by silencing expression[39]. We found that TE methylation levels were higher than genome-wide averages, although population-specific differences were detected (Supplementary Fig. 13 and Supplementary Table 12). We detected expressed TE transcripts in the EH23 $F_1$ hybrid, indicating ongoing TE activity (Supplementary Fig. 14). On the Y chromosome, the PAR and the SDR exhibited distinct patterns of gene expression and intact TE expression (Extended Data Fig. 9d,f), with the SDR showing increased methylation levels (Fig. 2h), consistent with its degenerate, gene-poor nature. Several TE families are actively transcribed, and many insertions are evolutionarily recent; however, TE frequency profiles varied distinctly among populations (Supplementary Figs. 15 and 16). The combination of recent divergence times for certain TE types (Fig. 2b,c), their enrichment near genes, and their population-specific distributions suggests that TEs contribute to both gene evolution and the regulation of adaptive responses in cannabis.

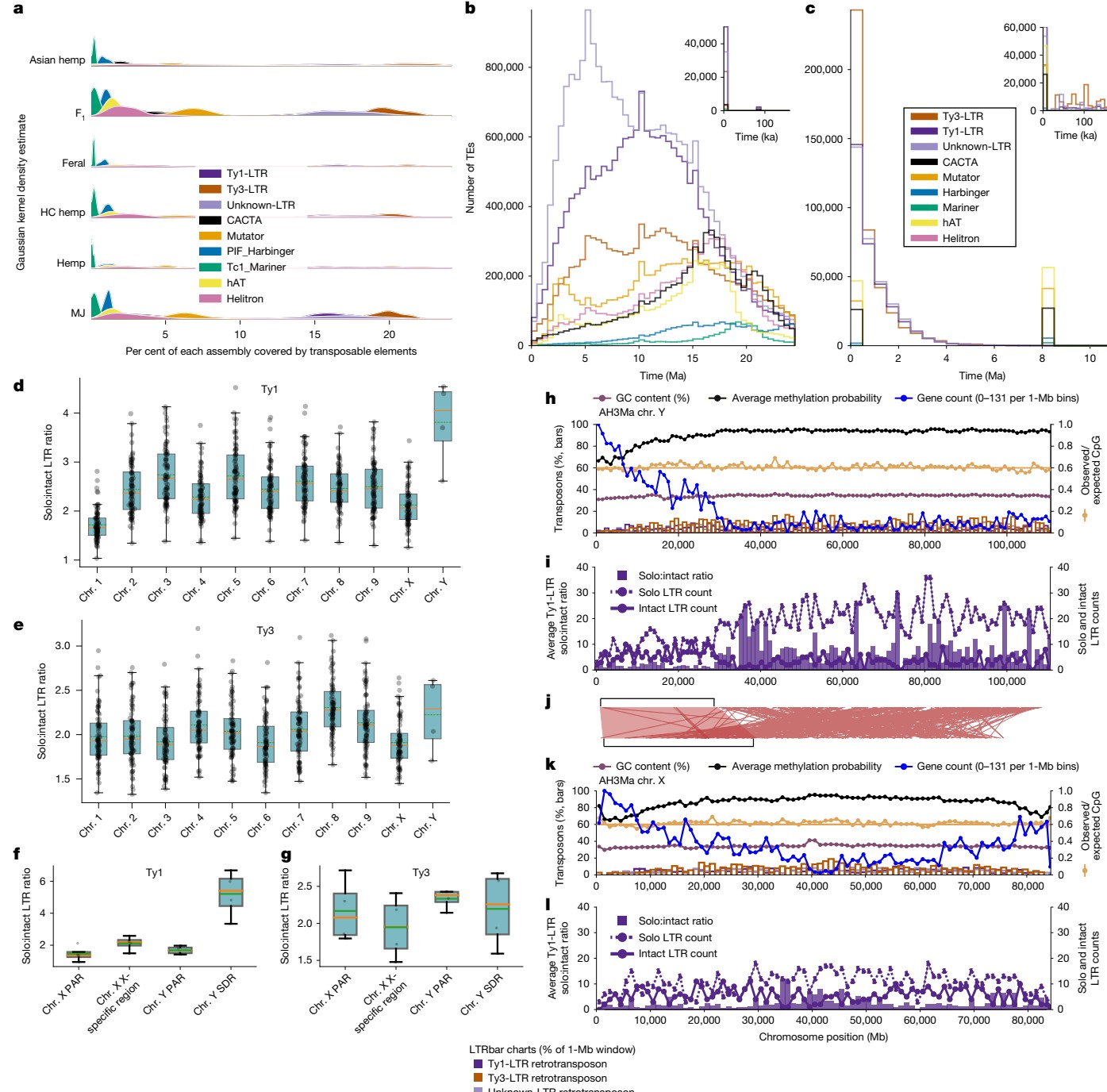

**Fig. 2 | TEs shape the cannabis pangenome. a**, Percentage of genome covered by TEs, using 78 chromosome-level, haplotype-resolved genomes, grouped by population. The y axis shows the Gaussian kernel density estimation. **b**, Across the pangenome, the age distribution of fragmented TEs, with inset showing their distribution within the past 100,000 years. In the inset, the highest density occurs since 10 thousand years ago (ka). **c**, Age distribution of intact TEs, with inset showing distribution within the last 100,000 years. In the inset, the highest density occurs within 10 kya. **d**, Average solo:intact ratio for Ty1-LTR elements in 78 chromosome-level, haplotype-resolved genomes, grouped by chromosome. For all box plots, the green dashed line is the mean and the orange solid line is the median. The lower and upper edges of the box correspond to the lower and upper quartiles. The vertical lines (whiskers) extending from the box reflect the minimum and maximum values in the dataset. Each scatter point represents an individual genome. **e**, Average solo:intact ratio for Ty3-LTR elements. **f**, Average

solo:intact ratio for Ty1-LTR elements in the sex chromosomes grouped according to boundary (PAR, X-specific region or SDR). **g**, Average solo:intact ratio for Ty3-LTR elements in the sex chromosomes grouped according to boundary (PAR, X-specific region or SDR). **h**, Genomic landscape plot for the AH3Mb Y chromosome, showing density of LTRs, methylation, CpG content and transcripts across the chromosome. **i**, Genomic landscape plot for the AH3Mb Y chromosome, showing the ratio of solo:intact Ty1-LTRs across the chromosome. **j**, Visualization of whole-genome alignments between the AH3Ma X and Y chromosomes. The bracketed region with high similarity is the PAR. **k**, Genomic landscape plot for the AH3Ma X chromosome, showing density of LTRs, methylation, CpG content and transcripts across the length of the chromosome. **l**, Genomic landscape plot for the AH3Ma X chromosome, showing the ratio of solo:intact Ty1-LTRs across the length of the chromosome.

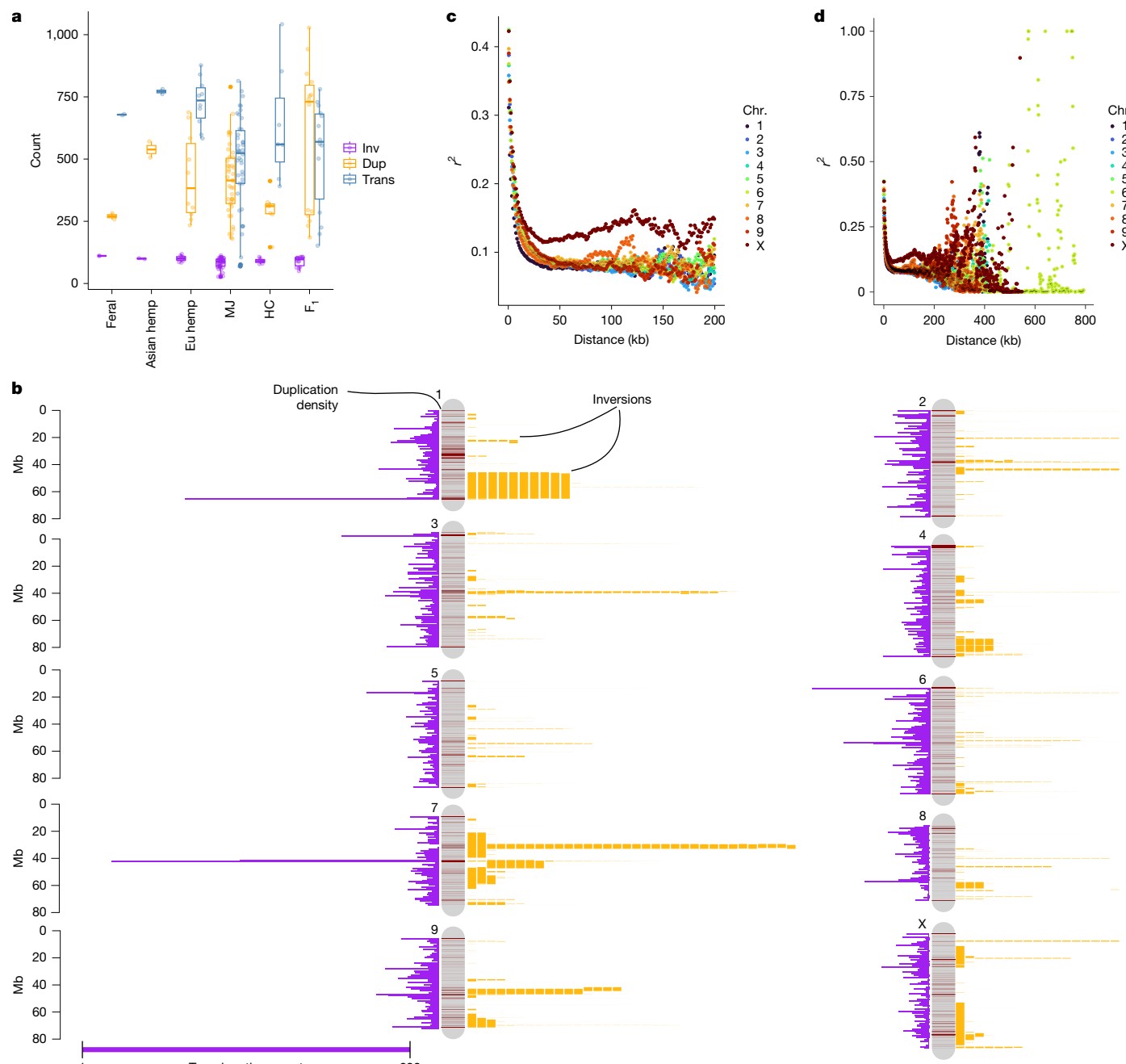

**Fig. 3 | Structural variants occur at different frequencies in populations and are non-randomly distributed across the genome. a**, Frequencies of inversions (inv), duplications (dup) and translocations (trans) by population. European hemp, Asian hemp and MJ populations differ significantly in average translocation and duplication counts, but not in inversions. Each box plot represents the median (centre line), two hinges (quartiles) and two whiskers (1.5× the interquartile range (IQR). **b**, Non-random genomic distribution of translocations (purple histograms), duplications (dark red bands) and inversions (mapped as length-scaled yellow bars on the right side of chromosomes, with each bar equal to one inversion). **c**, LD plot limited to 200-kb interactions, highlighting the general decay curves, with the X chromosome exhibiting a markedly reduced decay rate. Collectively, across the 78 haplotype-resolved, chromosome-scale assemblies, LD decay plots showed decay to half of the maximum $r^2$ value around 10 kb, which is similar to wild outcrossing soy and rice populations. **d**, LD decay plots extended to 800 kb. The increase in long-range (hundreds of kb to Mb) LD patterns found in certain SNP pairs further underscored the importance of using accurately phased genome assemblies and consideration of SVs for mapping and improvement efforts.

## SVs drive innovation

Given the high abundance of young, active TEs in cannabis, we examined their role in shaping pangenome SVs (Fig. 3). SV counts varied most in translocations and duplications, mirroring population-specific TE abundance (Fig. 2a), whereas inversions showed the least variation (86 per genome on average) (Fig. 3a and Extended Data Fig. 10). However, inversion sizes ranged from 200 bp to 25 Mb (average 304 kb), forming a multi-modal distribution, suggesting that multiple evolutionary forces shaped inversions of different lengths. Whereas the SNP heterozygosity ranged between 1 and 2.5% in the pangenome, the heterozygosity (variable regions) when including SVs and non-alignable regions was on average 20.6% of total genome length (Supplementary Fig. 17), highlighting the extent of previously uncharacterized genomic variation in cannabis.

TEs frequently caused small-to-medium translocations, duplications and inversions, whereas larger inversions arose at breakpoints that were

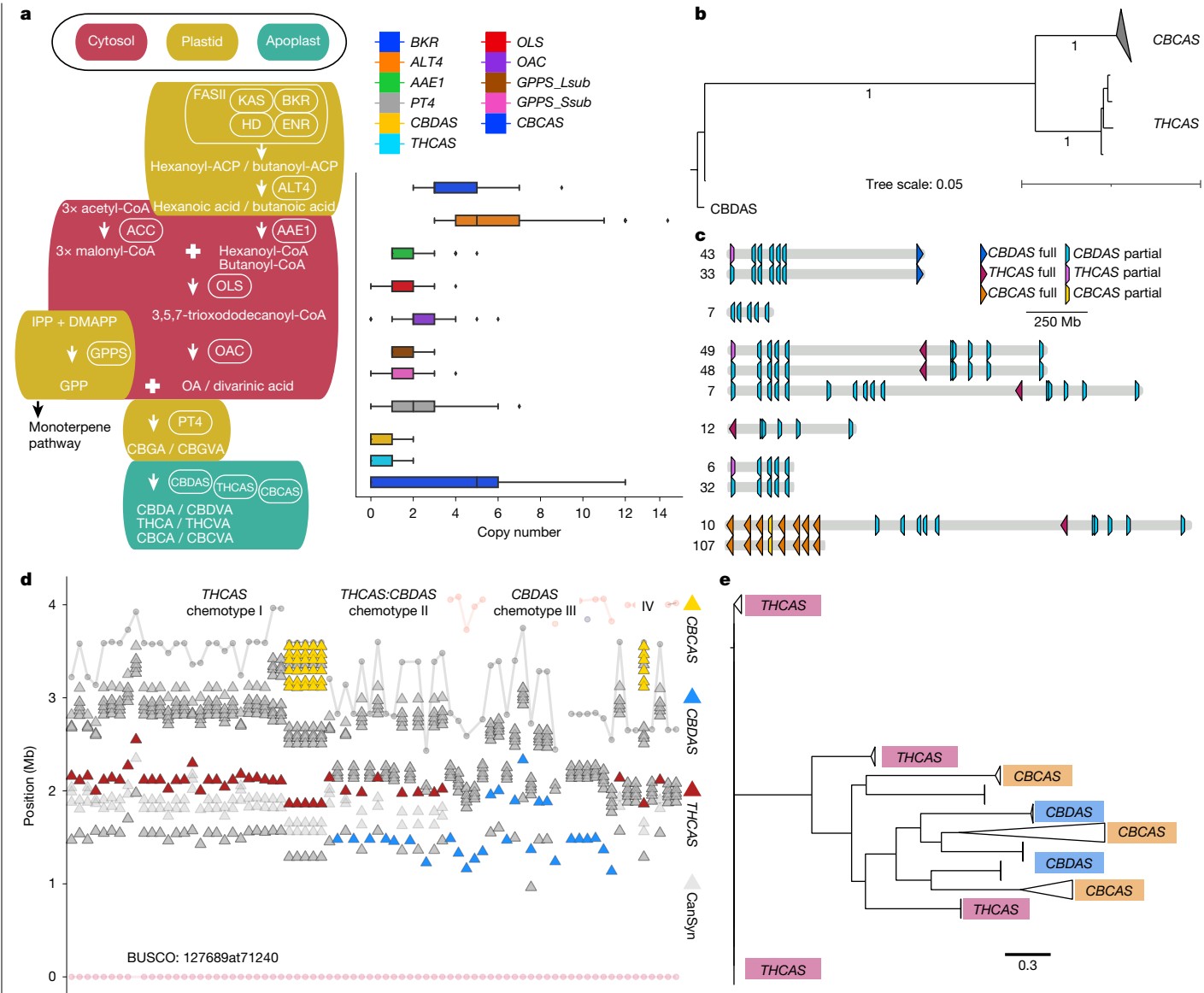

**Fig. 4 | The cannabinoid biosynthesis pathway is domesticated yet shows contrasting patterns of genetic diversity and synteny. a**, Cannabinoid biosynthesis pathway and gene copy numbers across the pangenome, per assembly. The left and right edges of the box plots correspond to the lower and upper quartiles, and the line in the box is the median. The horizontal lines that extend outward from the box (whiskers) reflect the minimum and maximum values in the dataset. Each scatter point represents an individual genome. AAE, acyl-activating enzyme; ACC, acetyl-CoA carboxylase; ACP, acyl carrier protein; CBCVA, cannabichromevarinic acid; CBDVA, cannabidivarinic acid; CBGVA, cannabigerovarinic acid; CoA, coenzyme; DH, dehydratase; DMAPP, dimethylallyl pyrophosphate; ENR, enoyl-ACP reductase; FASII, type II fatty acid synthase; GPPS, geranyl diphosphate synthase; IPP, isopentenyl diphosphate; KAS, β-ketoacyl-acyl carrier protein synthase; OAC, olivetolic acid cyclase; OLS, olivetolic acid synthase; THCVA, tetrahydrocannabivarin acid. **b**, Consensus maximum-likelihood phylogeny of aligned coding sequences from cannabinoid synthases, with the proportion of 100 bootstrap replicates shown on branches where values are greater than 0.75. Each branch tip represents a distinct cluster of synthases within greater than 99% identity of 859 total synthases from across all 193 pangenome samples. **c**, Summary of common cannabinoid synthase cassette arrangements, with the number of occurrences in the pangenome shown on the left. Full, full-length synthase gene models; partial, truncated lower-stringency synthase alignments that probably represent pseudogenes. **d**, Synthase cassettes exhibit variation in synteny, as seen in BUSCO anchored local alignment of chr. 7. Red triangle, *THCAS* cassettes; blue triangles, *CBDAS* cassettes; yellow triangle, *CBCAS* cassettes; grey triangles, low-stringency synthase matches (pseudogenes); grey and pink circles, BUSCOs. **e**, Maximum-likelihood tree of helitron DNA TE sequences flanking (2 kb upstream or downstream) cannabinoid synthase genes in the 78 haplotype-resolved, chromosome-scale assemblies.

enriched with segmental duplications and inverted repeats[40] (Extended Data Fig. 10). SV hotspots on chromosome (chr.) 1, chr. 4 and chr. 7 overlapped common inversion breakpoints and TE-enriched regions (Fig. 3b). Analysis of TEs within SV breakpoints (500 bp upstream and downstream, 1 kb total) revealed population-specific TE enrichment patterns. In MJ genomes, duplications frequently contained three DNA TE families and Ty3-LTR-RTs (Supplementary Table 13; *P* < 0.05, Welch's

*t*-test). Only Harbinger and Mutator DNA TEs were enriched at duplication breakpoints in other populations, whereas feral hemp duplications showed no significant TE enrichment, suggesting recent TE activity or alternative SV formation mechanisms. Inversions covered up to 7% of the genome, surpassing values observed in multi-species comparisons, such as soybean and grapevine[41]. Given the population-specific interplay of TEs and SVs, as well as their frequent proximity to genes,

our findings revealed a diverse set of mechanisms driving cannabis genome evolution, many of which were undetected in previous resequencing efforts.

Segregation distortion has been observed across multiple regions of the cannabis genome[16], mirroring patterns detected in the $F_1$ EH23 hybrid (Extended Data Fig. 3), which suggests that SVs may contribute to allele transmission biases[42]. Long inversions, such as the one found on chr. 1 (19.5 Mb in length; Fig. 3b), may function as a supergene, perhaps maintained as a balanced polymorphism through associative overdominance[43]. Indeed, the 17 instances of this inversion were found to be heterozygous in 15 samples and homozygous in 1. This inverted region contained around 1,203 genes, spanning many functions, including the core circadian and flowering time gene *PSEUDO RESPONSE REGULATOR 3* (*PRR3*), which has been implicated in the 'autoflower' DN behaviour in cannabis[44] as well as in flowering time variation associated with range expansion in major crops (soybean and sorghum) and natural populations[45–47]. *PRR3* contained a high-$F_{st}$ SNP (0.61) as well as biased expression in our $F_1$ EH23 hybrid that was recessive for the DN trait (Extended Data Fig. 3). We found that pairwise SNP $r^2$ values and local principal component analysis (PCA) plots of this area suggested some level of haplotype formation and increased linkage disequilibrium (LD; >10 kb) across this region, especially at the interior breakpoint (Fig. 3c,d and Supplementary Fig. 18). However, these were not obvious signals of complete differentiation or recombination suppression as has been shown in other species[48].

## Domesticated cannabinoid pathway

Cannabis is the only prolific producer of cannabinoids, although other plants (such as liverworts) and fungi synthesize smaller quantities[49]. Although key enzymes in the cannabinoid biosynthetic pathway have been identified (Fig. 4a and Supplementary Fig. 19), the genomic organization of the final step in this pathway remained unresolved owing to the complexities of the cannabis genome (Supplementary Fig. 20). This mystery was clarified with the discovery of full-length *THCAS*, *CBDAS* and *CBCAS* genes nested within conserved TE cassettes, arranged in arrays on chr. 7[11]. However, it was unclear whether this TE-mediated arrangement of synthase genes was conserved across the cannabis pangenome.

Cannabinoid synthases duplicated and neofunctionalized from the ancestral Berberine bridge enzyme-like (BBE-like) family of genes on chr. 7, then were ultimately reduced to a limited set of functional *THCAS* and *CBDAS* alleles through the domestication process[11,50] (Fig. 4b,c and Supplementary Fig. 21). Across the pangenome, each haploid genome hosted a maximum of one full-length *THCAS* or *CBDAS*, which were arranged in similar arrays of TE cassettes, most of which contained synthase pseudogenes. These cannabinoid synthase cassettes were found in a limited number of arrangements with association to specific TEs (Fig. 4c,e, Supplementary Figs. 22 and 23 and Supplementary Table 14), which suggested that selection had linked a small range of functional alleles to pseudogene cassette haplotypes. As a result, most *THCAS* and *CBDAS* genes were non-syntenic, and associated with inversions between cannabis types, but were generally located within a region constrained to about 1.5 Mb on chr. 7 (Figs. 1a and 4d). Whereas the cannabis pangenome exhibits high genomic variation, the conserved structure of the *THCAS* and *CBDAS* loci suggests that these regions are under strong selective pressure.

Full-length *CBCAS* paralogues were typically 15–20 Mb from the chr. 7 centromere, but owing to a genomic inversion, sometimes appeared within about 1.2 Mb of *THCAS* (Fig. 4d). *CBCAS* occurred in 56% (110 out of 193) of genomes, in arrays of 1–15 copies (Supplementary Fig. 22). Although *CBCAS* is capable of producing cannabichromenic acid (CBCA) in yeast[16], analysis of more 59,000 cannabis samples detected almost no CBCA, probably owing to low natural levels[51]. In EH23, *CBCAS* expression was low across all tissues, suggesting that CBCA accumulation has not been under strong selection, potentially owing to human preference for THC and CBD (Supplementary Fig. 19).

## Varin cannabinoids and fatty acid genes

*In planta* cannabinoid alkyl side-chain length can vary from one to at least seven carbons, with five carbons being the most common in modern gene pools[52]. Three-carbon side-chain cannabinoids (propyl; tetrahydrocannabivarin (THCV), cannabivarin (CBDV) and cannabigerovarin (CBGV)) are much less common, but have attracted interest as novel therapeutic agents[53]. Prior studies have characterized the polygenic nature of this trait, and associated the *β-keto acyl carrier protein reductase* (*BKR*) gene with varin cannabinoid production, but left open at least one step needed for a complete biosynthetic hypothesis[54]. We extended the model for varin cannabinoid production by identifying a complex of acyl-lipid thioesterase (*ALT3* and *ALT4*) genes located near the beginning of chr. 7 that were associated with varin production in our $F_2$ mapping population and were contained within a common haplotype in our *k*-mer-based crossover analysis of trios (Fig. 5, Supplementary Note 1, Supplementary Figs. 24–26 and Supplementary Tables 15 and 16). There was high *ALT* gene copy number variation in cannabis, ranging from 2–14 copies (considering both phased and unphased assemblies) across 4 chromosomes (Fig. 4a). Most plant genomes contain 4–5 *ALT* homologues, and some contain only a single homologue (for example, *Brassica rapa* and *Glycine max*)[55]. Additionally, *ALT* protein sequence variation in cannabis was notable, with distinct orthogroup membership of each *ALT4* in EH23a and EH23b genomes (Fig. 5b,c), despite these genes being located at similar positions (Fig. 5b). Since the shortest known fatty acid product of a plant fatty acyl-thioesterase is a 6:0 fatty acid generated by the *Arabidopsis ALT4*, the EH23a *ALT4* allele is a lead candidate for further experimentation. However, given the crossover locations (Fig. 5a), potential for linkage disequilibrium and short-read mapping issues in this region, any of these *ALT3* and *ALT4* trans-duplicated genes (or splice variants) could be causal for varin cannabinoid production. Alternatively, they may have overlapping sub-functionalized substrate specificities, which would pose challenges for further mapping and improvement efforts[56].

Although the *BKR* gene on chr. 4 was identified previously in a genome-wide association study, the pangenome showed that a 2-bp deletion produced a 6-exon loss-of-function gene model, which lacked catalytic active site residues (Fig. 5e). Thus, reduction or loss of function in this gene is probably required to increase the butyryl-acyl carrier protein pool, which one of the *ALT3* or *ALT4* gene products then hydrolyses to butyric acid, leading to varin cannabinoid biosynthesis (Fig. 4a). Since cannabis hosts *BKR* genes on chr. 3 and chr. 4, loss of catalytic function of one copy is unlikely to fully terminate iterative fatty acid chain synthesis, which could also explain why varin cannabinoids are only found in certain ratios with pentyl cannabinoids[52,54]. Across the pangenome, the EH23a 6-exon *BKR* variant was exclusively found in HO40 pedigree samples (high varin); all other samples, except one 8-exon version of *BKR* in the seed oil cultivar Finola (low varin producer) were 11- or 12-exon models. The phylogenetic relationships of the predicted *BKR* proteins showed that the 6-exon gene may be closer to certain Asian hemp, European hemp and feral variants (Fig. 5f). However, one of the 11-exon gene clades contained the varin-producing AutoCBDV genome, and the potential varin producer Durban Poison, which could be reduced-function variants. Some reports suggest that there is no defined geographic origin associated with the varin chemical phenotype[57]. However, other studies report plants that contain high levels of varin cannabinoids from the southern regions of Africa and certain regions of Asia[52,58]. Collectively, the *BKR* gene phylogeny and whole-genome *k*-mer-based clustering analysis suggest an Asian origin for varin cannabinoid genes used in this breeding project (Fig. 1f,g). Deeper understanding of these biosynthetic pathways enhances our

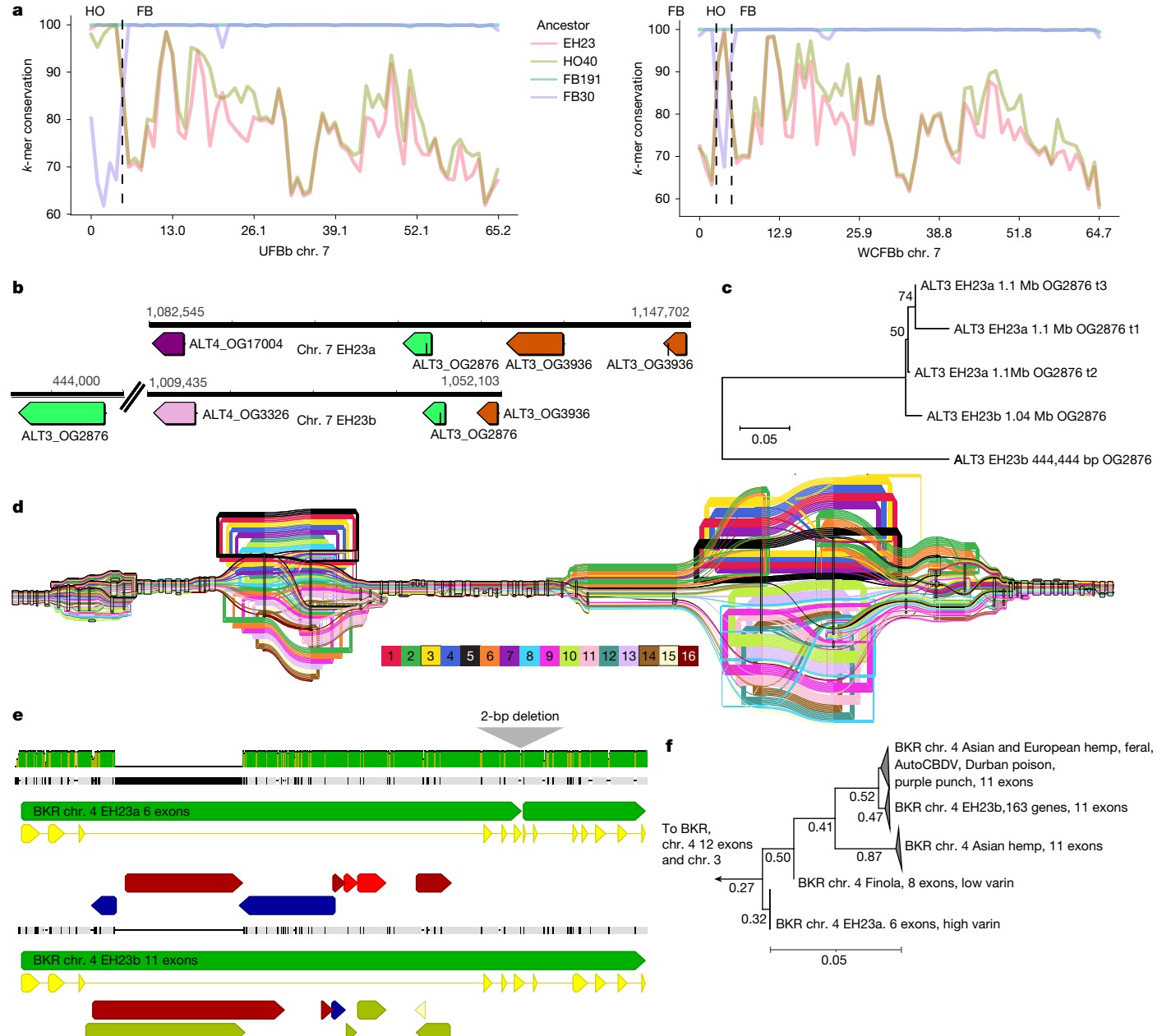

**Fig. 5 | *ALT* gene *trans*-duplication and diversification explains varin cannabinoid phenotype in cannabis. a**, PanKmer crossover analysis identifies the specific breakpoints on chr. 7 (vertical dashed lines) for the *ALT* gene haplotypes in relation to cannabinoid synthases. UFBb has one crossover at 5 Mb that breaks the linkage between HO40 (HO) *THCAS* and the varin haplotype *ALT* genes, whereas WCFBb has two crossovers, which result in an absence of the HO40 *ALT* alleles. **b**, *ALT3* and *ALT4* arrangements on chr. 7 of EH23a and EH23b. **c**, Protein-based neighbour-joining phylogeny, showing relationships between the three *ALT3* orthogroup OG2876 members on chr. 7, including the three alternative splice variants (t1, t2 and t3) from the EH23a gene model, with the proportion of 100 bootstrap replicates shown on branches where values are greater than 0.50. **d**, Sequence tube map visualization of variation at *ALT4* from the 16 haplotype graph pangenome incorporating the following colour-coded assemblies: 1, AH3Ma; 2, AH3Mb; 3, BCMa; 4, BCMb; 5, EH23a; 6, EH23b; 7, GRMa; 8, GRMb; 9, KCDv1a; 10, KCDv1b; 11, KOMPa; 12, KOMPb; 13, MM3v1a; 14, SAN2a; 15, SAN2b; 16, YMv2a. **e**, *BKR* 6-exon and 11-exon gene models and local nucleic acid alignment for EH23a and EH23b, with close up of the 2-bp deletion that truncates the 6-exon model. Green arrows, gene models; yellow arrows, coding sequences; red, blue, white and olive arrows, TEs. Green vertical bars represent per cent identity for the alignment. **f**, BKR protein-based neighbour-joining phylogeny from 772 pangenome gene models, with the proportion of 100 bootstrap replicates shown on branches where values are greater than 0.25.

ability to select and optimize diverse cannabinoid production and suggests a path toward improvement of seed oil lipid profiles.

## Conclusions

Our analysis of 193 cannabis genomes revealed that global diversity remains undersampled, with Asian germplasm notably underrepresented. Despite its phenotypic similarity to European hemp, Asian hemp carries highly divergent genomic regions, some of which align more closely with North American drug-type cannabis, suggesting undiscovered wild relatives and unresolved taxonomy. TE activity and hybridization, rather than whole-genome duplication, drive cannabis genome evolution. SVs uncover previously hidden diversity missed by short-read sequencing. Whereas cannabinoid synthase genes show limited variation, genes related to fatty acid metabolism, growth, defence and terpene biosynthesis exhibit extensive diversity and copy number

variation. We assembled fully phased cannabis X and Y chromosomes, identifying a variable SDR–PAR boundary and unique male-specific homologues on the large Y chromosome that may influence flowering time and development, offering new targets for breeding.

Finally, the discovery of extensive variation in fatty acid biosynthesis genes (for example, *ALT* and *BKR*) suggested that cannabis has untapped potential for lipid metabolism. Given the overlap between cannabinoid biosynthesis and seed oil pathways, hybridizing diverse parental lines beyond the conventional Northern European hemp seed oil gene pool could yield novel lipid profiles and traits. The conservation and utilization of Asian hemp and wild cannabis will be critical for advancing cannabis breeding and the development of agronomic and pharmaceutical potential.

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

## Methods

### Plant material

*C. sativa* pangenome samples were selected from multiple sources to maximize the genetic diversity, history and agronomic value. A large portion of the pangenome comes from the Oregon CBD (OCBD) breeding programme that includes elite cultivars; foundational marijuana lines potentially originating from the 1970s to the present; and elite trios used for different aspects of the breeding programme (Extended Data Figs. 1 and 2, Supplementary Table 1 and Supplementary Fig. 1). The remaining cultivars come from the US Department of Agriculture (USDA) Germplasm Resource Information Network (GRIN) and German Federal Genebank (IPK Gatersleben) repositories, as well as collections made by the Salk Institute from various breeders. The pangenome includes European and Asian fibre and seed hemp, feral populations, North American marijuana (type I) and North American high cannabinoid yielding (CBD or CBG) hemp (type III and IV). Additional cannabinoid diversity is represented with chemotypes presenting high expression of pentyl or propyl (varin) homologues of CBD or THC, and cannabinoid-free (type V) plants. Flowering time variation is also captured with the inclusion of both regular short-day and day-neutral (autoflowering) phenotypes (Supplementary Table 1).

### EH23 phased, haplotype-resolved, chromosome-scale anchor genome

EH23a (HO40) and EH23b (ERB) are haplotype-resolved assemblies for ERBxHO40_23, an $F_1$ resulting from a cross between parents, ERB and HO40, both proprietary female inbred lines from OCBD. ERB is a DN (autoflower), type III (CBDA-dominant) plant that is part of the drug-type group more closely related to European HC hemp. HO40 is type I propyl cannabinoid (THCVA and THCA)-producing, short-day flowering responsive, and is part of the drug-type marijuana group (MJ) with a closer affinity to Asian hemp. The genetically female (XX) ERB plant was induced to produce male flowers by treatment with silver thiosulfate and used to pollinate HO40. One individual from the $F_1$ populations (ERBxHO40_23) was selected for genome sequencing. Initial genome size estimates of ERB × HO40_23 using flow cytometry estimated a diploid genome size of 1445.6 Mb (722.8 Mb haploid genome size). High molecular weight (HMW) DNA was extracted from leaf tissue. Following DNA extraction and library preparation (see 'HMW DNA isolation and genome sequencing') HiFi reads were generated on the Pacific Bioscience (PacBio) Sequel II. Hifiasm v0.16.1[59] was then used in conjunction with Hi-C reads to produce initial assemblies. After assembly, Hi-C reads were aligned to the Hifiasm_HiC contigs using the Juicer v1.6.2 pipeline[60] followed by ordering and orientation utilizing version 180922 of the 3D-DNA pipeline[61]. The scaffolded assemblies were then manually corrected using Juicebox v1.11.08[62].

### EH23 $F_2$ population

In addition to the whole-genome sequencing data described above, ERBxHO40_23 was self-pollinated using silver thiosulfate induced masculinization of select flowers, to create an $F_2$ mapping population. From this $F_2$ population, individuals were scored for autoflower and varin content, and sequenced using Illumina 100 bp reads by NRGene (Nrgene Technologies). Illumina WGS genotyping runs were performed on 288 plants from this population, plus the ERBxHO40_23 parent. Trim_galore was used to trim sequences using: --2 colour 20, resulting in 271 individuals for analysis[63]. On average samples had 8.5× coverage. Minimap was used to align each sample to EH23b.softmasked.fasta. Freebayes was used to call variants: -g 4500 -0 -n 4 --trim-complex-tail --min-alternate-count 3[64]. Bcftools was used to filter on QUAL > 20 scores (99% chance variant exists)[65]. Finally, Vcftools[66] tools was then used to further filter SNPs: --remove-indels --minGQ 20 --maf 0.25 --max-missing 1 --min-alleles 2 --max-alleles 2 --stdout –recode[66]; only

sites that were scored as heterozygous (0/1) in ERBxHO40_23 sample were retained, resulting in 93,251 SNPs.

### EH23 $F_2$ cannabinoid HPLC methods

High-performance liquid chromatography (HPLC) was conducted according to the protocol thoroughly described previously[67] to determine relative propyl and pentyl cannabinoid content in all the plants used in this study, including $F_2$ progeny. In short, mature flower tissue was collected from each individual, frozen at −80 C and homogenized, before cannabinoids were extracted in methanol.

### EH23 RNA sequencing

ERBxHO40-21 seedlings were grown under controlled environmental conditions. Various tissues were collected during the development of the plants, including early and late flowers, foliage, foliage under a 12-h inductive light regimen, roots and shoot tips. Total RNA extraction was done using the QIAGEN RNeasy Plus Kit following manufacturer protocols. Total RNA was quantified using Qubit RNA Assay and TapeStation 4200. Prior to library prep, we performed DNase treatment followed by AMPure bead clean up and QIAGEN FastSelect HMR rRNA depletion. Library preparation was done with the NEBNext Ultra II RNA Library Prep Kit following manufacturer protocols. Then these libraries were run on the NovaSeq6000 platform in 2× 150-bp configuration.

### EH23 haplotype expression analysis

We measured gene expression levels using Salmon v1.6.0[68]. In brief, the raw paired end short reads from sequencing were mapped to the CDSs from both haplotypes (EH23a and EH23b) and the abundance was estimated in transcripts per million (TPM) for downstream analysis. Mapping rates were calculated with samtools flagstat[65]. The minimum TPM threshold for a given gene was ≥0.1. Haplotype gene pairs were identified by reciprocal best hits and synteny using blastp and MCScanX[69], and only genes shared between both haplotypes were included. A minimum of ≥95% sequence similarity and a threshold of 5 TPM difference between haplotypes was imposed. Visualization was performed using a combination of Matplotlib[70], SciPy[71] and NumPy[72], and expression values are shown in heat maps as $\log_2$TPM to represent log fold change. Enrichment of Biological Processes GO Terms was performed with topGO[73] with the following parameters: resultWeight <- runTest(topGOdata, algorithm = "weight01", statistic = "fisher"). A multiple test correction was performed with the following command: fullResults$p.adj <- p.adjust(as.numeric(fullResults$weightFisher), method = "fdr"). The background gene universe included all genes with a GO term from either EH23a or EH23b.

### Ace High sex-biased gene expression analysis

We collected flower and leaf tissue from four Ace High plants, two male and two female, at the same developmental time point, at 08:00 and 20:00, for a total of 16 samples. Since Ace High males flower several weeks before female plants under normal outdoor conditions, plants were germinated and grown under long days and transferred to inductive short-day conditions for flowering, which resulted in both male and female plants developing flowers at the same time. Samples were collected at two times of day to capture all transcripts regardless of their circadian or diurnal expression[74]. RNA was extracted with the Qiagen Plant RNA kit. Library prep was performed with the Oxford Nanopore Technologies (ONT) full-length cDNA kit. We aligned full-length cDNA to the haplotype-resolved Ace High (AH3Ma/b) genomes with minimap2 (v2.24)[75] and gene expression was measured using Salmon v1.6.0[68]. Sex-biased expression was assigned for all tissue-specific male and female samples (leaf and flower from two male plants (plants A and B, collected at 08:00 and 20:00) and two female plants (plants C and D, collected at 08:00 and 20:00)). Each sex-specific tissue had four replicates (for example, gene expression measurements from male flowers sampled from two male plants at two different time points

were averaged). Two categories of biased expression were defined: first, average expression that was higher (at least 5.0 TPM greater) in male or female samples, relative to the other sex; and second, male or female-only expression, where genes were not expressed in one sex (0.0 TPM for all replicates), but had an average of at least 1.0 TPM expression in the other sex. For GO term analysis with topGO[73], both categories of biased gene expression were combined. Fully syntenic genes were identified in the set of four genomes with X and Y chromosomes (AH3Ma/b, BCMa/b, GRMa/b and KOMPa/b) using genespace, and were grouped according to location in the PAR, SDR or X-specific region.

## Hi-C library preparation and sequencing

For the Dovetail Omni-C library, chromatin was fixed in place with formaldehyde in the nucleus and then extracted. Fixed chromatin was digested with DNAse I, chromatin ends were repaired and ligated to a biotinylated bridge adapter followed by proximity ligation of adapter containing ends. After proximity ligation, crosslinks were reversed and the DNA purified. Purified DNA was treated to remove biotin that was not internal to ligated fragments. Sequencing libraries were generated using NEBNext Ultra enzymes and Illumina-compatible adapters. Biotin-containing fragments were isolated using streptavidin beads before PCR enrichment of each library. The library was sequenced on an Illumina HiSeqX platform to produce ~30× sequence coverage. Then HiRise used (see read-pair above) MQ > 50 reads for scaffolding. Additional Hi-C libraries were generated using Phase Genomics Proximo Hi-C Kit (Plant) version 4.

## HMW DNA isolation and genome sequencing

All samples were sequenced on a PacBio Sequel II. For samples sourced from 'Michael' (Supplementary Table 1), HMW DNA was isolated using Carlson Lysis buffer and Qiagen Genomic tips as described in the ONT Protocol 'Plant leaf gDNA' *Arabidopsis* method. The DNA was further size-selected for fragments longer than 10–25 kb using the ONT Short Fragment Eliminator Kit (EXP-SFE001). HMW DNA was then confirmed by Tapestation Genomic DNA ScreenTape (Agilent 5067-5365) or Femto Pulse Genomic DNA 165 kb Kit (Agilent FP-1002-0275). For samples sourced from 'OCBD' (Supplementary Table 1), HMW DNA was isolated using a modified protocol[76]. In brief, samples were ground in a mortar and pestle with liquid nitrogen, two chloroform:isoamyl wash cycles were performed, and Total Pure NGS beads (Omega Biotek) were used as a substitute from the original protocol. Genomic DNA (gDNA) quality and purity was then assessed using a NanoDrop One (ThermoFisher) prior to starting library preparation. Continuous long read (CLR) libraries were made using the Pacbio protocol PN101-693-800 V1. Size selections on gDNA were made using the Blue Pippin U1 High Pass 30–40 kb cassette with a 30–40 kb base pair starting threshold to produce fragment distributions of 60–90 kb. HiFi circular consensus sequencing (CCS) libraries were prepared according to the PacBio protocol (PN 101-853-100 V5). Sheared gDNA fragment distributions with a modal peak ~18 kb were produced using g-Tubes from Covaris and Blue Pippin S1 High Pass 6–10 kb cassettes to remove everything under 10 kb in size.

## Pangenome assembly and scaffolding

All genomes labelled Hifiasm_HiC, Hifiasm_Trio_RagTag, Hifiasm_RagTag, and Hifiasm (Supplementary Table 1) were assembled using Hifiasm v0.16.1[59]. When available, Hi-C data and HiFi parental trio data were also incorporated into the assembly process defining the Hifiasm_HiC and Hifiasm_Trio_RagTag types respectively. CLR assemblies were generated using FALCON Unzip from PacBio SMRT Tools 9.0 Suite[77] and CCS labelled genomes were assembled with HiCanu v2.2[78]. After assembly, Hi-C reads were aligned to the Hifiasm_HiC contigs using the Juicer v1.6.2 pipeline[60] followed by ordering and orientation utilizing version 180922 of the 3D-DNA pipeline[61]. The scaffolded assemblies were then manually corrected using Juicebox v1.11.08[62]. Hifiasm_RagTag and Hifiasm_Trio_RagTag assemblies were scaffolded using the split

chromosomes of the 24 Hi-C scaffolded genomes and error checked with yak-0.1 (github.com/lh3/yak). Sourmash v4.6.1[79] was used to generate a Jaccard similarity matrix between the chromosomes and each un-scaffolded assembly, and the most similar version of chromosome 1 through X was concatenated to generate a reference for scaffolding via RagTag v2.1.0[80]. If the similarity matrix identified the Y chromosome as the best match, the assembly remained un-scaffolded. BUSCO v5.4.3[79] with the eudicots_odb10 dataset and assembly-stats v1.0.1 (https://github.com/sanger-pathogens/assembly-stats) were used on all assemblies to measure completeness and contiguity.

**Reference-based graph construction with Minigraph-cactus.** The graph pangenome of all 78 scaffolded and softmasked assemblies was generated with Minigraph-Cactus[20]. We used the cactus-pangenome command within an Apptainer (v1.1.8) Image[81] (https://quay.io/comparative-genomics-toolkit/cactus:v2.6.7-gpu) and the following parameter flags: --reference EH23a EH23b --vcf --vcfReference EH23a EH23b --giraffe --chrom-og --chrom-vg --viz --gfa --gbz. The seqFile input as well as the output graph in various formats (vg, paf, hal, etc.) can be found at https://resources.michael.salk.edu. We also compiled variants across the pangenome in terms of each assembly's coordinates by using vg deconstruct -a -C (vg tools v1.61.0 "Plodio") to derive vcf files from the Minigraph-Cactus gfa output and then using vcfbub --max-ref-length 100000 --max-level 0 to flatten nested variants and remove those >100 kb in length (see 78csatHaps_minigraphcactus_<assembly>.vcf.gz)[20,82,83].

## Reference-free graph construction with PGGB

**Input sequences and orientation.** We generated two versions of each PGGB graph, one with the fasta files provided in the 'Assembly files' table and in the JBrowse instance at https://resources.michael.salk.edu (mixed-orientation) and one with fasta files in which the sequences have been consistently oriented to match the plus strand of the corresponding homologous chromosome in EH23a (consistent-orientation).

For PGGB graph 16csatAsms, we generated one graph per autosomal chromosome from the following 16 scaffolded and softmasked assemblies: AH3Ma, AH3Mb, BCMa, BCMb, EH23a, EH23b, GRMa, GRMb, KCDv1a, KCDv1b, KOMPa, KOMPb, MM3v1a, SAN2a, SAN2b and YMv2a. We generated one combined fasta file per chromosome as inputs for PGGB (see 16csatAsms_chr[1-9]_combined.fa.gz and 16csatAsms_chr[1-9]-oOrient_combined.fa.gz for the consistent- and mixed-orientation fasta inputs, respectively, at resources.michael.salk.edu). We constructed per chromosome graphs instead of a single graph for the entirety of all assemblies combined due to the computational requirements for analysing genomes of this size and repetitive content (Extended Data Fig. 6).

For PGGB graph 13csatSexChroms, the 13 scaffolded and softmasked sex chromosome sequences AH3Ma.chrX, AH3Mb.chrY, BCMa.chrX, BCMb.chrY, EH23a.chrX, GRMa.chrY, GRMb.chrX, KCDv1a.chrX, KCDv1b.chrX, KOMPa.chrX, KOMPb.chrY, SAN2a.chrX and SAN2b.chrX were combined into one fasta file (see 13csatSexChromsCombined_filtOrientation.fa.gz and 13csatSexChromsCombined_origOrientation.fa.gz for the consistent- and mixed-orientation fasta inputs, respectively, at https://resources.michael.salk.edu).

## Graph generation

Nextflow v24.04.3.5916[84] was used to run the nf-core/pangenome v1.1.2 - canguro deployment[85,86] of PGGB[22] within the nextflow singularity profile. All default PGGB settings were used for graph generation. For PGGB graph 13csatSexChroms, the flag --vcf_spec was used to compile sequence variation across the pangenome relative to each assembly's coordinates, and each vcf was further processed with vcfbub --max-ref-length 100000 --max-level 0 to flatten nested variants and remove those >100 kb in length[20] (see 13csatSexChroms_pggb-fOrient_<assembly>.vcfbub.vcf.gz and

13csatSexChroms_pggb-oOrient_<assembly>.vcfbub.vcf.gz files for vcfs from graphs generated with consistent- and mixed-orientation input fastas, respectively, at https://resources.michael.salk.edu). For PGGB graph 16csatAsms, PGGB was run without the flag --vcf_spec and, instead, vg deconstruct -a was used to compile sequence variation across the pangenome from the final gfa file for each autosomal chromosome (vg tools v1.61.0 "Plodio")[82,83]. Per-autosome vcf files were concatenated into a single file for each assembly using bcftools[65] and then processed with vcfbub --max-ref-length 100000 --max-level 0 to flatten nested variants and remove those >100 kb in length[20] (see 16csatAsms_pggbByChrom_<assembly>.vcf.gz and 16csatAsms_pggbByOriginalChrom_<assembly>.vcf.gz for vcfs from graphs generated with consistent- and mixed-orientation input fastas, respectively, at resources.michael.salk.edu). Identical parameters were used for each pair of graphs generated with consistent- and mixed-orientation inputs.

### Visualization
Visualizations of the graph pangenomes were generated from the FINAL_GFA files of the PGGB pipeline run on consistent-orientation input fastas. Vg files were derived from gfa files using vg convert[82,83]. Then prepare_vg.sh and prepare_chunks.sh were used to visualize the pangenome variation at regions of interest in a local instance of the Sequence Tube Map server (https://github.com/vgteam/sequenceTube-Map.git, cloned on 4 September 2024).

### Short-read mapping to graph pangenome
Short-read sequences from the EH23 $F_2$ population and Ren et al.[2] were aligned to the pangenome graph with vg giraffe (example command: vg giraffe -Z {input.inputGBZ} -d {input.inputDist} -m {input.inputMin} -f {input.inputR1} -f {input.inputR2} -t {threads} > {output.outputFile})[87]. Summary statistics were collected with vg stats[82] (example command: vg stats -a {input.inputGAM} {input.inputGBZ} > {output.outputFile}). Calculate read support from GAM file with vg pack[82] (example command: vg pack -x {input.inputGBZ} -g {input.inputGAM} -Q 5 -t {threads} -o {output.outputFile}). Variants for the $F_2$ mapping population were called with vg call[88] (example command: vg call --gbz {input.inputGBZ} -k {input.inputPack} -S EH23b -t {threads} > {output.outputFile}). Downstream processing of VCF files was performed with BCFtools[65] (example commands: (1) bcftools view -a -f PASS merged.sorted.vcf.gz > merged.sorted.a.PASS.vcf.gz; (2) bcftools norm --fasta-ref EH23b.softmasked.fasta -m -any merged.sorted.a.PASS.vcf.gz > merged.sorted.a.PASS.normed.vcf.gz; (3) bcftools norm --fasta-ref EH23b.softmasked.fasta --rm-dup exact merged.sorted.a.PASS.normed.vcf.gz > merged.sorted.a.PASS.normed_no_dups.vcf.gz). Filtering of the pangenome graph-based VCF file to compare with the linear reference-based VCF file was performed with VCFtools[66] (example command: vcftools --remove-indels --minGQ 20 --maf 0.25 --max-missing 0.3 --min-alleles 2 --max-alleles 2 --stdout --recode --gzvcf merged.sorted.a.PASS.normed_no_dups.vcf.gz > merged.sorted.a.PASS.normed_no_dups.more_filter_missing0.3.vcf.gz).

### Graph pangenome data availability
Input and output files for the graph pangenomes described above (78csatHaps generated by Minigraph-Cactus, and 16csatAsms and 13csatSexChroms generated by PGGB) are available at https://resources.michael.salk.edu. Vcf files have been added as tracks to the *Cannabis* genomes JBrowse instance at https://resources.michael.salk.edu.

### Base-calling methylated cytosines
Genomic reads from the raw ONT FAST5 files generated from *Cannabis* sequencing samples were used for methylation calling. Genome assemblies generated for the same individuals were used as references for alignment. FAST5 data were converted to POD5 format using the pod5 software package (https://github.com/nanoporetech/pod5-file-format). Methylation calling was performed with ONT base-calling software Dorado version 0.3.4 (https://github.com/nanoporetech/dorado/). Dorado uses the raw POD5 data and a reference to identify methylated cytosines. This was performed with the super high accuracy (SUP) base-calling model trained for R9.4.1 or R10.4.1 pore type and 400 bps translocation speed, according to the sequencing conditions for each line. The assembled genomes generated from each sample were used as references to generate an aligned BAM file with MM/ML tags containing 5mC and 5hmC methylation calls. These were then piled up with modkit (https://github.com/nanoporetech/modkit), and the piled-up calls (aggregating 5mC with 5hmC) were used for calculating genome-wide methylation frequencies across all CG sites.

### Gene and repeat prediction
Gene model prediction involved a multi-step pipeline and was applied to all assemblies.
(1) We first curated a repeat library using RepeatModeler[89] on a small number of high-quality *Cannabis* assemblies and pre-existing repeat libraries. We used OrthoFinder (v2.5.4)[90] to group repeats for deduplication. The final repeat library included 10% of the sequences from each repeat orthogroup (minimum 1 sequence) for a total of 6,262 sequences from 5,793 groups.
 a. Finola (GCA_003417725.2)
 b. CBDRx (GCF_900626175.2)
 c. Purple_Kush (GCA_000230575.5)
 d. ERBxHO40_23
 e. ERBxHO40_23
 f. I3
 g. JL (GCA_013030365.1)
 h. ERB_F3
 i. Cannbio-2 (GCA_016165845.1)
 j. W103
 k. JL_Mother (GCA_012923435.1)
 l. FB30
 m. TS1_3_v1
 n. HO40
(2) For all 193 genomes, repeats were masked with RepeatMasker (v4.1.2)[91] using the repeat library (above).
(3) We predicted gene models with the TSEBRA pipeline (using Braker v2.1.6)[92]. We developed a Snakemake workflow for running TSEBRA, available here: https://gitlab.com/salk-tm/snake_tsebra. We incorporated a variety of pre-existing protein libraries from cannabis and other organisms as evidence: (a) *Arabidopsis thaliana*; (b) *Theobroma cacao*; (c) *G. max*; (d) *Rhamnella rubrinervis*; (e) *Ziziphus jujuba*; (f) *Trema orientale*; (g) *Vitis vinifera*; (h) *Prunus persica*; (i) *Morus notabilis*; (j) *C. sativa*; (k) *H. lupulus*.
(4) RNA-seq libraries (Supplementary Table 2) were aligned with either hisat2 (v2.2.1)[93] for short-read mapping, or minimap2 (v2.24)[75] for full-length cDNA. Short-read Illumina data was trimmed with fastp[94]. The expression data was incorporated into the TSEBRA pipeline as gene model evidence.
(5) Putative functional annotations of gene models were assigned using eggnog-mapper (v2.0.1)[95].
(6) Overall gene model quality and completeness was assessed by comparing genome BUSCO (v5.4.3)[96] scores to proteome BUSCO scores on the eudicots_ocdb10 dataset (Supplementary Table 1: https://doi.org/10.6084/m9.figshare.25869319.v2).
(7) EDTA v1.9.6[97] was also utilized to identify TEs in the cannabis pangenome with the following command: EDTA.pl --genome {inputFastaFile} --anno 1 --threads 32.

### Ideogram methods
Ideograms for each pair of chromosomes for the 78 chromosome-level, haplotype-phased genomes were created using ggplot2 [https://ggplot2.tidyverse.org] in R (www.R-project.org) (Fig. 1 and Extended

Data Fig. 5). The length of each chromosome was determined using 'nuccomp.py' (https://github.com/knausb/nuccomp) and used with ggplot::geom_rect() to initialize the plot. One million base pair windows were created for each chromosome where the number of CpG motifs were counted for each window with the program motif_counter.py (https://github.com/knausb/nuccomp). The CpG count was converted into a rate by dividing by the window size; this also accommodated the last window of each chromosome, which was less than one million base pairs in size. These rates were scaled by subtracting the minimum rate and then dividing by the maximum rate (the maximum rate after subtracting the minimum rate), on a per chromosome basis. In order to visually emphasize the enrichment of the CpG motif in the centromeric region, an inverse of the CpG rate was taken by taking one and subtracting the CpG rate for each window. This scaled, inverse CpG rate was used for the width of each one mbp window and coloured based on gene density using the viridis magma palette (https://doi.org/10.5281/zenodo.4679424).

Structural variation among each pair of chromosomes was determined using minimap2[75] alignments. The minimap2 comparisons were annotated using SyRI[98]. The syntenous and inverted regions were plotted using ggplot2::geom_polygon() in a manner inspired by plotsr[99] but implemented in R (github.com/ViningLab/CannabisPangenome).

The location of candidate loci within EH23 haplotypes A and B were determined using BLASTN[100]. Query sequences were as follows: CBCA synthase (LY658671.1), CBDA synthase (AB292682, AB292683, AB292684), THCA synthase (AB212829, AB212830), and olivetolic acid cyclase (NC_044376.1:c4279947-4279296, NC_044376.1:c4272107-4271242). These sequences were combined with centromeric, telomeric and rRNA sequences in the file blastn_queries_rrna_cann.fasta (https://github.com/ViningLab/CannabisPangenome). BLASTN was called with the following options: -task megablast -evalue 0.001 -perc_identity 90 -qcov_hsp_perc 90. Tabular results (subject chromosome, subject start of alignment, subject end of alignment) from BLASTN were read into R and plotted on ideograms with ggplot2::geom_rect() (https://ggplot2.tidyverse.org).

## Centromere and telomere analysis

ONT and PacBio based long read-based genome assemblies enable the assembly of some of the highly repetitive centromeres and telomeres sequences[101]. Centromeres were identified by searching genomes using tandem repeat finder (TRF; v4.09) using modified settings (1 1 2 80 5 200 2000 -d -h)[102]. Tandem repeats were reformatted, summed and plotted to find the highest copy number tandem repeat per our previous methods to identify centromeres[101] (Extended Data Fig. 5c).

Telomeres were estimated using two different methods. First, the TRF output was queried for repeats with the period of 7 for the 14 different version of the canonical telomere base repeat: AAACCCT, AACCCTA, ACCCTAA, CCCTAAA, CCTAAAC, CTAAACC, TAAACCC, TTTAGGG, TTAGGGT, TAGGGTT, AGGGTTT, GGGTTTA, GGTTTAG and GTTTAGG: (grep -a 'PeriodSize=7' *.genome.fasta.1.1.2.80.5.200.2000.dat.gff | grep -a 'Consensus=AAACCCT\|Consensus=AACCCTA\|Consensus=ACCCTAA\|Consensus=CCCTAAA\|Consensus=CCTAAAC\|Consensus=CTAAACC\|Consensus=TAAACCC\|Consensus=TTTAGGG\|Consensus=TTAGGGT\|Consensus=TAGGGTT\|Consensus=AGGGTTT\|Consensus=GGGTTTA\|Consensus=GGTTTAG\|Consensus=GTTTAGG' -). Second, we searched raw ONT and PacBio reads for telomere sequences using our TeloNum algorithm[103]. Although the results were variable across the pangenome assemblies, in general, telomere sequence was found at the end of the chromosome with an average length of 16 kb for PacBio assemblies and 60 kb for ONT assemblies. The differences between ONT and PacBio telomere length most likely reflected the input read length of >100 kb and 15–20 kb, respectively. TeloNum analysis of the raw reads supported the distributions from the assemblies consistent with most chromosomes having telomere sequence while being shorter than the actual size. *Cannabis* telomeres are on the longer side for a eudicot and could be explained by its predominantly clonal propagation for medicinal uses[104].

Centromere sequence was identified based on the hypothesis that it will be the most abundant repeat in the genomes that also has a higher-order repeat (HOR) structure[101,105]. Two different repeats with HOR were identified in the PacBio HiFiasm assemblies, whereas only one was found in the ONT assemblies and the previous CBDRx assembly, which is based on ONT sequence[11]. The highest copy number repeat was 370 bp that varied between 20–30 Mb (2–4% of the total genome) with HOR at 740 and 1,110 bp (Extended Data Fig. 5). The second highest, and the only one found in the ONT assemblies, was a 237 bp repeat that varied between 3–5 Mb (0.4–1.0% of the total genome) and had HOR at 474 and 711 bp (Extended Data Fig. 5). Mapping of the 370-bp repeat to the chromosome-resolved genomes revealed that this repeat was primarily located at the end of the chromosomes next to the telomere sequence, which suggested that it may be related to the CS-1 sub-telomeric repeat[106]. Comparison of the putative 370-bp centromeric repeat and the CS-1 sub-telomeric repeat showed they are the same repeat element. By contrast, the putative 237-bp centromeric repeat predominantly was found on chr. 6 and chr. 8 in the predicted centromere region (Fig. 1a and Extended Data Fig. 5). However, smaller 237-bp arrays were found on all chromosomes across the assemblies in the predicted centromere region (based on CpG, methylation, gene content and TEs) with most assemblies having small arrays on chr. 6 and chr. 8.

## Ribosomal DNA detection and quantification

Ribosomal DNA (rDNA) 45S (18S, 5.8S and 26S) and 5S sequences were identified in the CBDRx/CS10 assembly (LOC115701787 5.8S, LOC115701759 18S, LOC115701762 26S and LOC115721558 5S) and used to BLAST against the pangenome assemblies (Fig. 1a and Extended Data Fig. 5). Across the scaffolded genomes the 45S array was predominantly located on the acrocentric end of chr. 8, and the 5S was located exclusively on chr. 7 between the cannabinoid synthase cassette array, consistent with published results with fluorescence in situ hybridization[106]. However, partial arrays were found in some assemblies on all of the chromosomes (Extended Data Fig. 5). The distribution of the partial arrays on different chromosomes could reflect variability across the genomes since some share similar locations across assemblies. Most arrays are found on the un-scaffolded contigs, suggesting that these variable arrays across different chromosomes could be the result of mis-assemblies. In general, there are on average 1,000 45S and 2,000 5S arrays in the cannabis genome; some assemblies have the 5S array completely assembled on chr. 7.

## Allele frequency methods

Genotype data in the VCF format[107] was input into R using vcfR[108]. Allele and heterozygous counts were made with vcfR. Wright's $F_{IS}$ was calculated[109] to provide the deviation in heterozygosity from our random, Hardy–Weinberg, expectation. Wright's $F_{IS}$ was calculated as (HS − HO)/HS, where HO is the observed number of heterozygotes divided by their number and HS is the number of heterozygotes we expect based on the allele frequencies, calculated as the frequency of the first allele multiplied by the frequency of the second multiplied by two and divided by their number. Scatter plots were generated using ggplot2. Graphical panels were assembled into a single graphic using ggpubr (https://cran.r-project.org/package=ggpubr).

## PanKmer genome analysis

Using PanKmer, we constructed two 31-mer indexes: a 'full' index of 193 *Cannabis* assemblies and a 'scaffolded-only' index of 78 scaffolded assemblies, using the 'pankmer index' command with default parameters. We calculated and plotted pairwise Jaccard similarities for all assemblies in the full index using 'pankmer adj-matrix' followed by 'pankmer clustermap --metric jaccard'. We calculated and plotted a

collector's curves for both the full and scaffolded-only indexes using the 'pankmer collect' command with default parameters. All scripts used for this analysis can be found on GitHub.

## Analysis of gene-based pangenome

We define the gene-based pangenome as the set of all gene families (orthogroups) with a representative in at least one genome of the pangenome. For each of 193 (as well as the 78 chromosome-level, haplotype-phased genomes, as a separate set) *C. sativa* genomes, the primary transcript of each high-confidence gene prediction was chosen as a representative. The proteins corresponding to each primary transcript were clustered into orthogroups using Orthofinder (v.2.5.4, see Orthofinder and synteny analysis section below)[90]. The set of primary transcript CDS were merged into a single FASTA file, and exact duplicates were removed with SeqKit (2.7.0)[110]. Among primary transcripts, likely contaminants were determined by identifying transcripts predicted on contigs where fewer than 90% of predictions were annotated as either 'viridiplantae' or 'eukaryote' according to eggNOG-mapper (v2.1.12)[95], and were removed. To mitigate the problem of unannotated genes, we aligned coding sequences of all primary transcripts to each of the 193 (78) cannabis genomes using minimap2 (v2.26)[75] with parameters 'minimap2 -c -x splice' to generate a PAF file with CIGAR strings for each genome. For each genome, if an aligned CDS sequence had a mapping quality of at least 60, had a number of CIGAR matches at least 80% of the query length, and did not overlap a directly annotated gene, it was considered an unannotated gene and its orthogroup was marked as present in the target genome. The set of orthogroups that had at least one representative present in all 193 (78) genomes were considered to be the core genome, the remaining orthogroups were considered to be the variable genome. The presence or absence of each orthogroup in each genome was recorded in a table (see Data availability). All scripts for this analysis are available from GitHub.

## Haplotypes, orthogroups and scores

In pangenomics, collector's curves (pangenome rarefaction) show the relationship of the number of haplotypes (here $H$) to the number of gene families or orthogroups (here $X$).

Given the $X$ orthogroups distributed across $H$ haplotypes, let the score $s_x \in [0, H]$ of an orthogroup $x$ be the number of haplotypes in which $x$ is present. For any score $s$ let $P(s)$ be the number of orthogroups with score equal to $s$.

$$P(s) = \sum_{x \in x_0 \ldots x_X} I_{s_x = s}(x)$$

Where $I_{s_x}$: $\{x_0 \ldots x_X\} \to \{0,1\}$ is the indicator function on $\{x \in x_0 \ldots x_X : s_x = s\}$.

## The collector's curves

The collector's curve $C(h)$: $[1, H] \to [0, X]$ is the expected number of orthogroups that will be present in a subset of $h$ haplotypes randomly drawn from the total set of $H$. It can be calculated by:

$$C(h) = \sum_{s \in 1 \ldots H} 1 - P(s) \prod_{i \in 0 \ldots h-1} \frac{H - s - i}{H - i}$$

The expected number of core orthogroups $C^\wedge(h)$ can be estimated by

$$C^\wedge(h) = \sum_{s \in 1 \ldots H} P(s) \prod_{i \in 0 \ldots h-1} \frac{s - i}{H - i}$$

Each of these is a special case of a general formula for the expected number of orthogroups with a score of at least $n$, based on the hypergeometric survival function:

$$C_n(h) = \sum_{s \in 1 \ldots H} P(s) S_{hyp}(n, H, s, h)$$

Where $S_{hyp}$ is the hypergeometric survival function or the hypergeometric cumulative distribution function subtracted from 1:

$$S_{hyp}(n, H, s, h) = 1 - CDF_{hyp}(n, H, s, h)$$

Where for clarity, the hypergeometric probability mass function (PMF) is:

$$PMF_{hyp}(n, H, s, h) = \frac{\binom{h}{n}\binom{H-s}{h-n}}{\binom{H}{h}}$$

With binomial coefficients defined as:

$$\binom{h}{n} = \frac{h!}{n!\,(h-n)!}$$

And, conventionally, the cumulative distribution function (CDF$_{hyp}$) is:

$$CDF_{hyp}(n, H, s, h) = \sum_{n_i \leq n} PMF_{hyp}(n_i, H, s, h)$$

So defined, we can see that the pan-genome collector's curve $C(h)$ is equivalent to $C_1(h)$, while the core genome collector's curve $C^\wedge(h)$ is equivalent to $C_h(h)$:

$$C(h) = C_1(h)$$

$$C^\wedge(h) = C_h(h)$$

## *k*-mer based collector's curves

The definition of the collector's curve is agnostic to the unit of genomic sequence, so the calculation of a *k*-mer based curve is identical to the orthogroup based curve, excepting that $X$ will be the number of *k*-mers and $x$ will represent a *k*-mer, rather than an orthogroup.

## *k*-mer analysis of pangenome assemblies and global diversity short-read libraries

Trim_galore was used to trim Illumina short-read sequences from Ren et al.[2] using: --2 colour 20[63]. These reads were next filtered for low abundance reads (trim-low-abund.py -C 10 -M 5e9), and then used to make a *k*-mer sketch (sourmash sketch dna -p scaled=1000,k = 31)[79]. All pangenome assemblies were also analysed for 31-mer frequencies (sourmash sketch dna -p scaled=1000,k = 31). Finally, all pairwise samples of Illumina read and pangenome assemblies were compared (sourmash compare -p 64 *.sig -k 31). The 31-mer distances were then plotted in R using (hclust(dist(sourmash_comp_matrix), method = "average")).

## Identification of pangenome core and dispensable genes

We assigned core and dispensable (nearly-core, cloud, shell, private) genes based on orthogroup membership (https://github.com/padgittl/ CannabisPangenomeAnalyses/tree/main/CoreDispensableGenes). Core genes were defined as being present in 100% of genomes (193 genomes), nearly-core genes were defined as being present in 95–99% of genomes (183–192 genomes), shell genes were found in 5–94% of genomes (10–182 genomes), cloud genes were found in 2–5% of genomes (3–9 genomes), and unique genes were found in 0.5–1% of genomes (1–2 genomes)[111]. This analysis was performed on all 193 genomes (Fig. 1e) and also visualized according to population (Supplementary Fig. 5). For the contig-level assemblies (103 genomes), only contigs with similarity to the ten chromosomes of EH23a were included. Gene sets were filtered to include only genes that were present on the

ten chromosomes and contigs homologous to the chromosomes. We performed an analysis of functional enrichment with topGO[73] for each of the core, shell, cloud, nearly-core, and unique gene groupings for each genome, where the background gene set was all genes with a GO term for a given genome. Among the core genes, the most common significant GO term in the pangenome was sesquiterpene biosynthetic process (GO:0051762), which was significant in all but one genome (PBBK), followed by GO:0045338 farnesyl diphosphate metabolic process, which was absent in three genomes (public genomes: CANN, FIN and PBBK) (Supplementary Table 4). This analysis was restricted to high-confidence gene models predicted with the TSEBRA pipeline. By contrast, the collector's curve analysis of gene content also included unannotated genome regions lacking gene model predictions, but with similarity to known genes, as a way to capture unsampled diversity (Fig. 1c,d and Supplementary Fig. 4; see also 'Analysis of gene-based pangenome').

### Repeat analysis

**Calculation of divergence time in TEs.** Estimates of divergence time shown (Fig. 2b,c) were calculated using the equation $T = (1 - \text{identity})/2\mu$, where identity was obtained from EDTA output GFF3 files described previously[97]. We used a substitution rate ($\mu$) of $6.1 \times 10^{-9}$ from *Arabidopsis*[112,113]. This analysis was performed on all genomes.

**Identification of solo to intact LTR-RT ratio.** To identify solo LTRs and intact LTR-RTs, we used the EDTA pipeline on 193 cannabis genomes[97]. We identified solo LTRs by first collecting the set of LTRs that were not assigned as intact LTR-RTs, which are retrieved on the basis of 'method=homology' in the attribute column of the TEanno.gff3 file. We applied thresholds to isolate solo LTRs from truncated and intact LTRs, as well as internal sequences of LTR-RTs. These thresholds include a minimum sequence length of 100 bp, 0.8 identity relative to the reference LTR, and a minimum alignment score[114] of 300. We also required that the four adjacent LTR-RT annotations did not have the same LTR-RT ID[115]. Further, we required a minimum distance of 5,000 bp to the nearest adjacent solo-LTR, intact LTR or internal sequence[116]. Last, we kept solo-LTR sequences that fell within the 95th percentile for LTR lengths[117]. Overall, this method represents a modified approach based on the solo_finder.pl script from LTR_retriever[114] and the LTR_MINER script[116] with guidance from the github page for LTR_retriever (https://github.com/oushujun/LTR_retriever/issues/41).

**Enrichment of TEs flanking genomic features.** The method presented as part of PlanTEnrichment[118] was adapted for the cannabis pangenome to assess TE enrichment both upstream and downstream of different genomic features, including cannabinoid synthase genes. The goal of the analysis was to identify TEs that are significantly associated with a specific category of genomic feature. In brief, 'X' represents a specific type of TE and 'Y' encompasses all TEs. The total number of X located upstream or downstream of a specific genomic feature (for example, cannabinoid synthases) is denoted as $a$; the total number of X located upstream or downstream of all genomic features (for example, all genes) is $b$; the total number of Y located upstream or downstream of a specific genomic feature (cannabinoid synthases) is $c$; and the total number of Y located upstream or downstream of all genomic features (all genes) is $d$. An enrichment score (ES) is defined as $ES = (a/b)/(c/d)$, and the $P$ value is defined as $p = (a+b)!(c+d)!(a+c)!(b+d)!/(a!b!c!d!N!)$, where $N$ is the sum of $a$, $b$, $c$ and $d$. A multiple test correction[119] was performed on the $P$ values using the Python library statsmodels[120]. Significance threshold cut-offs included a false discovery rate (FDR) < 0.05 and ES ≥ 2. We used bedtools intersect[121] to collect and survey the set of TEs located 1 kb upstream or downstream of the genomic feature category of interest. An example command: bedtools intersect -a assemblyID_genomic_feature_coord_file.txt -b assemblyID.TE.gff3 -wo > assemblyID_intersect_results.txt.

**Distance between genes and TEs.** The median and mean distances between genes and each of the TE categories was calculated using bedtools sort (bedtools sort -i genome.TEs.bed > genome.sorted.TEs.bed) and bedops closest-features (command: closest-features --closest --header --dist genome.sorted.genes.bed genome.sorted.TEs.bed > genome.closest_features.bed)[122]. To obtain the initial pre-sorted BED file for genes, the following command was used: cat genes.gff3 | grep mRNA | grep '\.chr' | awk '{print $1"\t"$4"\t"$5"\t"$7"\t"$3"\t"$9}' > genome.genes.bed. For TEs, the following command was used: cat genome.EDTA.TEanno.gff3 | grep '\.chr' | awk '{print $1"\t"$4"\t"$5"\t"$7"\t"$3"\t"$9}' > genome.TEs.bed. To calculate mean and median values, the built-in Python statistics module was used.

**Enrichment of genes associated with different categories of TEs.** We performed a GO term enrichment analysis to identify genes that were statistically significantly located near different types of TEs on the full pangenome. To identify genes near TEs, we first created a concatenated, sorted bed file with both gene and TE coordinates to find the nearest TE for a given gene, while excluding cases where the closest genomic feature to a given gene was another gene. For scaffolded genomes, genes and TEs were restricted to the ten chromosomes. For contig-level assemblies, genes were included if they were on a contig with similarity to one of the ten EH23a chromosomes. Next, we identified gene/TE pairs using bedops closest-features[122]. We performed a GO enrichment test for each genome separately using topGO with parameters algorithm = 'weight01', statistic = 'fisher', and Benjamini–Hochberg multiple test correction with FDR < 0.05[73]. The background gene universe for statistical comparison was the set of all genes with a GO term for a given genome. To assess broad patterns, only GO terms that were significant in at least five genomes were considered further. This analysis included the full set of genomes (Supplementary Table 11).

**Phylogeny of TEs surrounding cannabinoid synthases.** The genomic coordinates for the 2 kb flanking distance surrounding copies of *CBCAS, CBDAS* and *THCAS* for the 78 scaffolded assemblies were retrieved with bedtools flank (bedtools flank -i assemblyID_synthase_coords.bed -g chromSizes.txt -l 2000 -r 2000 > assemblyID_flanking_2000.bed). Next, the TEs contained in this flanking region were retrieved using bedtools intersect (bedtools intersect -a assemblyID_flanking_2000.bed -b assemblyID.EDTA.TEanno.gff3 -wo > assemblyID_intersect_2000.bed)[121]. The genomic sequences for each of the TE types identified with bedtools intersect were collected in a fasta file and aligned with mafft (mafft --auto helitron.fasta > helitron_aln.fasta)[107]. A maximum-likelihood tree was constructed with FastTree (FastTree -nt -gtr -gamma helitron_aln.fasta > helitron_aln.tree)[123]. The tree was visualized with FigTree[124]. To reduce redundancy in the full set of LTRs, CD-HIT was applied to the set of sequences, prior to multiple sequence alignment (cd-hit-est -i Ty1_LTRs.fasta -o Ty1_LTRs.cdhit.fasta -c 1)[125].

**Expression analysis of active TEs in EH23.** The non-redundant TE sequence library from EDTA was provided as the 'transcriptome' to salmon. Each of the EH23 RNA-seq samples was mapped to the TE transcriptome. Similar to the gene expression analysis, the minimum TPM threshold for a given TE was ≥0.1 TPM in ≥20% of samples[126]. The top 50 expressed TEs were visualized as a heatmap, showing $\log_2 TPM$ to represent log fold change.

**Observed/expected CpG.** 'CpG islands' are defined as unmethylated regions spanning >200 bp, GC content >50% and observed/expected CpG ratio >0.6. Cytosine methylation over time results in a loss of CpG dinucleotides after cytosine is deaminated to thymine. With cytosine methylation, the expectation is that CpG dinucleotides (CG, CHG, CHH (where H is A, T, or C)) will have greater methylation activity. The observed/expected CpG ratio calculation[127,128]

is: (CpG dinucleotide count/$L$)/(C count/$L$ × G count/$L$). Observed/expected CpG patterns were visualized in Fig. 2h,k.

**Analysis of TEs directly flanking SVs.** For each of the SV subtypes (inversions (INVS), duplications (DUPS), translocations (TRANS) and inverted translocations (INVTR)), the flanking region 500 bp upstream and downstream of each breakpoint (1 kb total for each breakpoint) was surveyed for TE content, using both intact and fragmented annotations. The set of 78 scaffolded, chromosome-level genomes were included, grouped by population. To compare with the genome at large, a random window was retrieved from the same genome and chromosome, with the same length as each of the SVs with bedtools shuffle, and the flanking windows were retrieved for each of the simulated breakpoints. Only cases where a specific type of TE was associated with both breakpoints of a single SV were further assessed with bedtools intersect. Both fragmented and intact TEs were included in this analysis. Statistical significance was assessed using Welch's two-sided $t$-test in SciPy[71]. TEs occur more frequently near SV breakpoints (500 bp upstream and downstream of the breakpoint; 1 kb total) than in randomly selected regions of the same length from the same chromosome and genome. To overcome differences in abundance, the randomly shuffled regions of the genome were bootstrapped (1,000 replicates), with the requirement that each of the simulated, shuffled TE datasets match the number of observed breakpoints in the population. The TE content of observed and simulated data was assessed for statistical significance with Welch's two-sided $t$-test in scipy[71] and Benjamini–Hochberg multiple test correction (alpha=0.5, method = 'indep', is_sorted=False)[120]. A test statistic and $P$ value was generated for each of the 1,000 bootstrap replicates. The average test statistic and $P$ value were then calculated (Supplementary Table 13).

**Orthofinder and synteny analysis.** We ran Orthofinder version 2.5.4 to aid in analysis of the 193 cannabis proteomes. Two runs were completed. The first was focused on our highest quality cannabis assemblies and only included scaffolded assemblies along with dozens of other plant samples from Plaza and a few samples from NCBI. Another run, including all of our cannabis pangenome assemblies, along with close relatives sourced from Plaza, was also produced to allow for detailed protein level analysis of the remaining assemblies. In all cases, only the primary (longest isoform unless otherwise annotated) protein sequence was used. Orthofinder results were analysed using a variety of methods, including Orthobrowser[129], which is capable of generating static web pages that allow for simultaneous visualization of gene tree dendrograms, gene tree multiple sequence alignments, and synteny of the selected gene and surrounding genes across all of the genomes (https://resources.michael.salk.edu/root/home.html).

Non-cannabis genomes included in the scaffolded cannabis Orthofinder run: (1) *Amborella trichopoda*; (2) *Aquilegia oxysepala*; (3) *A. thaliana*; (4) *C. sativa*; (5) *Carpinus fangiana*; (6) *Carya illinoinensis*; (7) *Ceratophyllum demersum*; (8) *Citrullus lanatus*; (9) *Corylus avellana*; (10) *Cucumis melo*; (11) *Cucumis sativus*; (12) *Fragaria vesca*; (13) *Fragaria X*; (14) *Lotus japonicus*; (15) *Magnolia biondii*; (16) *Malus domestica*; (17) *Manihot esculenta*; (18) *M. notabilis*; (19) *Nelumbo nucifera*; (20) *Oryza sativa*; (21) *Parasponia andersoni*; (22) *P. persica*; (23) *Quercus lobata*; (24) *Rosa chinensis*; (25) *Sechium edule*; (26) *T. orientale*; (27) *Trochodendron aralioides*; (28) *Vaccinium macrocarpon*; (29) *V. vinifera*; (30) *Z. jujuba*; and (31) *H. lupulus*.

Non-cannabis genomes included in the full cannabis Orthofinder run: (1) *F. vesca*; (2) *L. japonicus*; (3) *M. domestica*; (4) *P. persica*; and (5) *R. chinensis*.

**Calculation of sequence entropy for DNA and protein sequences.** We calculated sequence entropy for protein and DNA-based orthogroups on 193 genomes. High entropy corresponds to more diversity and variation among sequences in an orthogroup, and low entropy indicates less diversity and more similarity among orthogroup sequences. A minimum entropy value of 0 corresponds to matching identity. The maximum entropy corresponds to a random sequence of amino acids and is derived from the equation: $\log_2(20) = 4.32$, where 20 is the number of amino acids. For DNA, the maximum entropy[130] is $\log_2(4) = 2.0$. We computed the entropy for each column of the orthofinder multiple sequence alignment using the entropy function from scipy.stats[71] and then calculated the average entropy for the whole multiple sequence alignment. A minimum of five sequences per orthogroup were required for inclusion in the analysis. Pairwise comparisons were made for each orthogroup across populations, and the distribution of entropy values for each multiple sequence alignment was visualized as a joint histogram. This analysis was applied to both proteins (gene sequences) and DNA (TEs).

**Visualization and analysis of synteny with genespace.** To visually assess gene-level variation in the haplotype-resolved, chromosome-scale genomes with X and Y chromosomes (AH3M, BCM, GRM and KOMP), we used genespace version 0.9.3[131] within R version 4.2.2 (2022-10-31)[132]. We initially ran OrthoFinder[90] outside of the genespace environment and imported the results. To run the analysis, we used the synteny function, followed by plot_riparianHits. We built a pangenome representation with the pangenome function. We used the output file gffWithOgs.txt as the primary file used for obtaining syntenic gene pairs across all genomes in the subset. Gene IDs with an identical integer value in the 'og' column (last column) were retrieved as syntenic orthologues.

**SV analysis.** The 78 fully scaffolded assembly haplotypes were each aligned to the EH23a assembly using minimap2[75]. Syri was then used to call SVs on each alignment[98] and plotsr was used to visualize alignments and SVs[99]. CDS and TE content were analysed using bedtools intersect[121]. Inversion breakpoint repeats were called using blastn alignments of inversions with a minimum size of 10 kb. Windows of 8 kb centred around the start and end breakpoint of each inversion, and were aligned self-to-self, as well as to the breakpoint window pair on the opposing side of the inversion (start to end). Only one the top scoring alignment (excluding the full-length self–self alignment) was counted per breakpoint. Inverted repeats were called as alignments in opposing orientations and segmental duplications were called for alignments in the same orientation.

**Phased SNPs.** SNPs were also called using Syri[98] on the same assemblies and alignments as described above. SNPs from each of the two haplotypes per sample were merged into single phased genotype calls per sample, and sites with an N as the ALT call were removed (github.com/RCLynch414/SYRI_vcf.sh). Finally, vcftools was used to quality filter and thin SNP sites to a minimum of 1000 bp spacing: --remove-indels --minGQ 20 --remove-indv EH23a --min-alleles 2 --max-alleles 2 --thin 1000 --stdout --recode.

**LD calculations.** Phased SNPs from the scaffolded assemblies were first assessed for r2 correlations in with bin using plink[133]: --double-id --allow-extra-chr --set-missing-var-ids @:# --maf 0.01 --geno 0.1 --mind 0.5 --chr 7 --thin 0.1 -r2 gz --ld-window 100 --ld-window-kb 1000 --ld-window-r2 0 --make-bed. Then ld_decay.py was used to make decay curves (GitHub - erikrfunk/genomics_tools), which were plotted with ggplot in R. Separately LD heat maps were made using vcftools: --thin 50000 --recode; and plotted in with LDheatmap in R (sfustatgen.github.io/LDheatmap/).

**GO terms.** GO term enrichment tests were performed with the topGO package in R, using all high-confidence gene annotations from EH23a as the null distribution and classic Fisher test of significance[73].

**Selection scans with $F_{st}$ and XP-CLR.** $F_{st}$ values were calculated using vcftools for each phased SNP and the scaffolded assembly MJ and hemp population assignments; significance was calculated using the top 5% of these values. The XP-CLR model for selective sweeps was applied to the same SNPs and 20-kb genome widows 59; significance was calculated using the top 5% of these values.

**TreeMix.** The TreeMix model was run using only SNPs outside of gene models: -seed 69696969 -o out_stem -m 5 -k 50 -noss -root asian_hemp. One to 10 migration scenarios were simulated, and ranked based on the ln(likelihoods). Five migration events (-m = 5) was selected as the most likely final number.

**Local PCA.** The local PCA method was applied to the phased SNPs, with 1,000-bp minimum spacing between SNPs, and genome windows of 100 SNPs[134].

**Disease resistance gene analogue analysis.** Plant disease resistance gene analogues are defined by the presence of one or more highly conserved amino acid motifs in their encoded proteins. These motifs encode functional protein domains that determine pathogen specificity and subcellular localization. Depending on the particular pathosystem, resistance gene analogue proteins can be entirely cytoplasmic, or can span the cell membrane with cytoplasmic functional domains, extracellular domains, or both.

Drago2[135] was used to identify motifs conserved among plant disease resistance gene analogues for the 78 chromosome-level, haplotype-resolved genomes. Input files were transcript annotation fasta files for each genome. Sets of genes containing both nucleotide binding site (NBS) and leucine-rich repeat (LRR) domains were used as input to MEME to assess and compare amino acid composition in motifs over gene sets.

To identify genes related to powdery mildew resistance, the sequence of a marker mapped to chr. 2 in CBDRx was used as a blastn query against the EH23a anchor genome[136]. The resulting hit had 96% nucleotide identity on chr. 2 of EH23a at 77,292,037–77,291,397 bp. It was located in a cluster of 46 genes including 32 with kinase domains, six receptor-like kinases, two with nucleotide binding site plus transmembrane domains, one with coiled-coil and kinase domains, and one with coiled-coil, nucleotide binding site, and transmembrane domains. The blast hit itself was between two annotated kinase genes, EH23a.chr2.v1.g115480 and EH23a.chr2.v1.g115510.

The resulting top blast hits did not overlap with any gene annotations; however, 16 of the 38 genomes had blast hits on chr. 2 with >95% nucleotide identity to the CBDRx gene; of these, nine of these had 99–100% nucleotide identity over all three exons (1,745 bp, 1,448 bp and 287 bp), respectively. Sequences from five of the 16 genomes (H3S7a, OFBa, SZFBa, TKFBa and WCFBa) clustered separately from the rest. These were distinguished by a 1-bp insertion in the first exon, ten small indels (2–8 bp) in exonic space, and a 1,280 bp longer second intron. These regions were extracted and aligned with the CBDRx gene sequence, and the alignment was used to produce a maximum-likelihood tree (Extended Data Fig. 8).

Coiled-coil NBS–LRR genes (CNLs) showed a distinct pattern on chr. 3 and chr. 6. There were one to two CNL genes between 400–600 kb; two to four between 1–1.4 Mb; one to two at 6–8 Mb; a single CNL gene near the near the centromeric region of the chromosome at 35–37 Mb, and one to five (COFBa) CNLs between 78–84 Mb. Exceptions to this pattern were OFBa, H3S1a, and MMv31a, which lacked a CNL in the centromeric region. In SDFBa and SN1v3a, the centromeric CNLs were located at 42.8 and 47.5 Mb, respectively. SN1v3a had a CNL at 12.2 Mb, another exception to the overall pattern. Chr. 3 in this genome was larger than the others, at 90 Mb, compared to the rest at 80–85 Mb. Finally, GERv1a lacked a CNL in the 78–84 Mb region of chr. 3.

**Identification of terpene synthase genes.** Each of the *Cannabis* proteomes was aligned to a set of 40,926 protein sequences from UniProt (search criteria 'Embryophyta' and 'reviewed'; accessed on 20 September 2022) with blastp (version blast 2.6.0, build 7 December 2016)[137]. Alignment thresholds included an E-value threshold of less than $10^{-3}$, at least 20% query coverage, and a per cent identity based on the length of the alignment[138]. Terpene synthases were also identified based on the presence of Pfam domains, PF01397 and/or PF03936[139]. To assess domain content, each of the *Cannabis* proteomes was aligned to the Pfam-A.hmm database (last modified 15 November 2021; accessed 20 September 2022)[140] with hmmscan (HMMER 3.3.2 November 2020)[141] on default settings.

**Identification of genes in the precursor pathways for terpene and cannabinoid biosynthesis.** Terpene biosynthesis proceeds via two pathways: the chloroplastic methyl-D-erythritol phosphate pathway, which produces precursors for monoterpene and cannabinoid biosynthesis, and the cytosolic mevalonate pathway, which produces precursors for sesquiterpene biosynthesis. The protein sequences for these pathways[142–144] were aligned to each of the *Cannabis* proteomes with diamond version 2.1.4 on default settings[145].

**Synthase cassette analysis.** To identify full and partial length cannabinoid synthases in each of the 193 cannabis genomes, the reference cannabinoid synthase sequences were aligned to the genome with blastn. An enriched LTR sequence developed from CBDRx[11] was used as a reference to further aid in the identification of synthases. LTR08 is an LTR sequence from the CBDRx genome that is associated with the synthase cassettes. A Python script was written to take in cannabinoid synthase blast results and LTR08 blast results in table format. Synthase hits with length <500 bp were filtered out. LTR08 hits with bitscore <1,250 were filtered out. Synthase and LTR08 hits with mismatches <10 and zero gaps were labelled as 'Full' sequences. All other hits were labelled as 'Partial' sequences. Hits that shared the same starting position were then filtered to a single sequence and given one of the synthase labels according to the following. Full hits were retained and labelled as the corresponding functional synthase. Partial hits within 60 kb of an LTR08 hit upstream or downstream were labelled as *CBDAS* and retained. If there were no Full hits or hits with an LTR08 in proximity, the hit with the highest bitscore was labelled as the respective synthase and retained. The filtered and labelled synthases were then plotted onto a track to visualize cannabinoid synthase orientation for each region of a genome. A minimum of four synthase hits was required for visualization. Inkscape was used to visualize synthase cassette tracks. Manual edits were used to correct a few incorrect labels between *CBDAS* and *CBCAS*. Synthase cassettes are grouped by overall cassette shape.

**Cannabinoid synthase gene analysis.** First ORFinder was used to remove pseudogenes from the initial list of potential genes described above (ftp.ncbi.nlm.nih.gov/genomes/TOOLS/ORFfinder/linux-i64/). Then we used usearch11.0.667 to cluster synthase coding sequences: -cluster_fast -id 0.997 -sort length -strand both -centroids -clusters[146]. TranslatorX was then used to produce protein-guided multiple sequence alignments[147]. Synthase evolutionary history was inferred by using the maximum-likelihood method and General Time Reversible model in MEGA11[148].

**k-mer crossover analysis.** We used the anchoring function of PanKmer to locate crossover events in known trios of cannabis genotypes (Supplementary Table 15). Eleven trios included FB191 as a varin-donor parent and 6 trios included SSV as a varin-donor parent. The parents of FB191 are HO40 and FB30, while the parents of SSV are HO40 and SSLR; in both cases, HO40 was the varin donor. For each trio, the $F_1$ genome was haplotype-resolved and included one haplotype from

a varin-donor parent and one from a non-varin donor parent. In each case, we used PanKmer anchoring to identify the 'varin haplotype'. For FB191 trios, we generated a 31-mer index of the FB191 genome using 'pankmer index' with default parameters. Using a Python script importing PanKmer's API functions pankmer.anchor_region() and pankmer.anchor_genome()[21], we anchored the FB191 index in each haplotype of the cross, for example COFBa and COFBb. We identified the varin haplotype as the haplotype with higher 31-mer conservation in the FB191 index. We applied the same procedure to SSV trios using a PanKmer index of SSV. We then sought to trace potential varin alleles from HO40 to the varin haplotype of the cross. To represent HO40, we generated two single-genome 31-mer indexes: one for the HO40 genome and a second for the highly similar EH23a sequence. We also generated single-genome 31-mer indexes of FB30 and SSLR. For each FB191 cross, we anchored the HO40 index, EH23a index and FB30 index in the varin haplotype. We inferred crossover events at loci with a clear 'haplotype switch' indicated by $k$-mer conservation values. We repeated the same procedure for SSV trios, applying the SSLR index in place of the FB30 index. All scripts for this analysis are available on GitLab.

**Varin SNP association tests and genetics.** First, the BestNormalize package in R was used to select the ordered quantile (ORQ) method to transform the varin ratio data, which were initially deemed multimodal. Then the model BLINK from the GAPIT package in R[149] was used with PCA.total=6 to test associations between SNPs in the $F_2$ population and transformed varin ratio data (Supplementary Table 16). This PCA.total parameter was selected based on visual evaluation of QQ plots for PCA.total values 1–10, where 6 was the smallest number that did not show systemic inflation of $P$ values[149]. Next, gene and TE models were manually assessed in the regions surrounding the four FDR-corrected significant SNPs (Supplementary Table 16), in conjunction with the $k$-mer based crossover results. Of the four significant SNPs, we focused further analyses on the genes associated with the top two highest phenotypic variance explained (Supplementary Fig. 25). Then, Orthofinder groups for *BKR*, *ALT3* and *ALT4* were extracted, and the three *ALT3* and *ALT4* orthogroups were pooled into a single set of ALT gene counts. Phylogenies of BKR and ALT protein sequences were constructed in MEGA with the neighbour-joining method from the orthogroups using 100 bootstrap replicates[148]. The *BKR* alignment and translation displayed was made using the Geneious[150] alignment algorithm on default settings (Fig. 5).

**Sex chromosome SDR–PAR boundary identification and comparisons.** Y based $k$-mers (Y-mers) were mapped to X/Y haplotypes using BWA (v.0.7.17) mem, requiring perfect alignments and allowing multimapping up to 10 times. To determine putative SDR–PAR boundaries, we focused on extracting conserved orthologues in regions with decreased Y-mer mapping density for subsequent gene tree analysis. Orthologues were defined using OrthoFinder (v.2.5.4) with the multiple sequence alignment option. OrthoFinder was executed using proteins from all available male (XY) assemblies from this study, including all male and several female contig-level assemblies, and additional haplotype-resolved assemblies from other studies: (1) BOAXa; (2) BOAXb; (3) AH3Ma; (4) AH3Mb; (5) BCMa; (6) BCMb; (7) GRMa; (8) BCMb; (9) GRMa; (10) Carmagnola_HAP2[29]; (11) Futura75_HAP1[29]; (12) Futura75_HAP2[29]; (13) OttoII_HAP1[29]; (14) OttoII_HAP2[29]; (15) Uso31_HAP1[29]; (16) Uso31_HAP2[29]; (17) FIMv1a; (18) FIMv1b; (19) GVA-H-22-1061-002_hap1[34]; (20) GVA-H-22-1061-002_hap2[34]; (21) GVA-H-21-1003-002_hap1[34]; (22) GVA-H-21-1003-002_hap2[34]; (23) SAN2a; (24) SAN2b; (25) TIBv1a; (26) TIBv1b; (27) WFv1a; (28) WFv1b; (29) WIv1a; (30) WIv1b; (31) YMMv1a; and (32) YMMv1b.

Gene trees were estimated for ten conserved orthologues spanning putative SDR–PAR boundaries, to determine which orthologues were SDR- or PAR-linked in each assembly. For example, strong support for separate clades containing either X- or Y-linked orthologues is expected when the Y gametologue (1:1 orthologues on X and Y chromosomes) is tightly linked to the SDR[151].

For all ten conserved orthologues or gametologues, we: (1) used blastn (BLAST+ v.2.14.1) and bedtools (v.2.31.0) getfasta, to find and extract nucleotide sequences for full-length genes (including introns); (2) aligned each gene matrix with MAFFT (v.7.505), using the options '--localpair --maxiterate 1000'; and (3) inferred maximum-likelihood trees with IQ-TREE (v.1.6.12) with the options '-MFP -bb 1000'. Following our analysis of X–Y gametologue trees, we used gene coordinates corresponding to the first putative Y-specific, SDR-linked gene to define each SDR boundary, then padded starting coordinates by 10 bp. The start of X-specific regions (that is, region on the X that does not recombine with the Y and is collinear to the Y-SDR) was defined based on X-gametologue coordinates corresponding to the first Y-specific gene.

The SDR–PAR boundary was defined using gene trees of XY gametologues from SDR bordering regions, which we identified by mapping male-specific $k$-mers to each haplotype. Our gene tree analysis revealed two major Y haplotype groups with distinct SDR boundaries (Ya and Yb). The 'cloud boundary' represents variation in the SDR–PAR boundary within cannabis, based on XY gametologue relationships. Ya was more common in our dataset ($n = 6$), and exhibits an ~132-kb extended SDR that spans the cloud boundary; whereas this region remains PAR-linked in the less frequent, Yb, haplotype ($n = 2$). The Ya haplotype reported in the main text was found in BCMb (feral), GRMa (HC hemp), AH3Mb (MJ), and Carmagnola, which is a fibre hemp landrace from Northern Italy, and the Yb haplotype was found in Kompolti (Hungarian fibre cultivar), which was selected for superior fibre characteristics in the 1950s from an older Italian variety, and GVA-H-21-1003-002 (isolated feral population from NY, USA).

### Reporting summary

Further information on research design is available in the Nature Portfolio Reporting Summary linked to this article.

## Data availability

The NCBI BioProject ID for the cannabis pangenome is PRJNA1140642. All of the pangenome sequencing data at NCBI Sequence Read Archive (SRA) is under the BioProject accession PRJNA904266. The BioProject accession IDs for EH23a and EH23b are PRJNA1111955 and PRJNA1111956, respectively. Genomes and annotation files for all 193 assemblies (including links to corresponding US National Plant Germplasm System accessions), orthobrowser and Genome Jbrowse instances, and input and output files for graph pangenomes are available at https://resources.michael.salk.edu. Annotations for R-genes, terpene synthases, cannabinoid synthases and additional genome visualizations are available from https://figshare.com/projects/Cannabis_Pangenome/205555 (ref. 152) and https://doi.org/10.25452/figshare.plus.c.7248427.v1 (ref. 153). Links to specific genome datasets are provided in Supplementary Table 1 (https://doi.org/10.6084/m9.figshare.25869319.v1 (ref. 154)). Source data are provided with this paper.

## Code availability

Scripts and analysis pipelines are available at https://github.com/anthony-aylward/CannabisPangenomeShared (ref. 155) and https://github.com/padgittl/CannabisPangenomeAnalyses (ref. 156).

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

**Acknowledgements** The authors thank members of the Michael laboratory for discussion on this work; and T. Gordon and Z. Stansell for sending leaf material from lines from the GRIN collection. This work was funded in part by the Tang genomics fund (T.P.M.), a National Science Foundation Plant Genome Postdoctoral Research Fellowship to L.K.P.-C. (NSF-IOS PRFB 2209290), and the development of pangenome tools in the Michael laboratory was supported by Bill and Melinda Gates Foundation (INV-040541) (T.P.M.). Support for this work was also provided by the US Department of Agriculture National Institute of Food and Agriculture Postdoctoral Fellowship (USDA NIFA) no. 2022-67012-38987 (S.B.C.), USDA NIFA no. 2023-67013-39620 (A.H.) and National Science Foundation (NSF) IOS-PGRP CAREER no. 2239530 (A.H.).

**Author contributions** T.P.M., R.C.L., S.C., A.R.G., K.V. and L.K.P.-C. conceived and organized research efforts. R.C.L., L.K.P.-C., T.P.M., B.J.K., N.T.H., N.A., A.A., A.M., J.K.K., H.I.C., A.R.G., A.T., P.C.B., S.B.C. and A.H. analysed pangenome data. R.C.L., L.K.P.-C., A.R.G., T.P.M., K.C., E.R.M., T.D. and S.C. conducted greenhouse, field and laboratory experiments. R.C.L., L.K.P.-C., T.P.M., B.J.K. and K.V. wrote and edited the manuscript. R.C.L., L.K.P.-C. and T.P.M. revised the manuscript. All authors read and approved the manuscript.

**Competing interests** S.C. was a co-founder of Oregon CBD. A.R.G. and A.T. were employees of Oregon CBD. R.C.L. is a stakeholder in Saint Vrain Research LLC, which manufactures hemp-based products. T.P.M. is a founder of the carbon sequestration company CQuesta. A.H. is a co-founder of the genotyping company Veil Genomics. The other authors declare no competing interests.

**Additional information**
**Correspondence and requests for materials** should be addressed to Ryan C. Lynch, Lillian K. Padgitt-Cobb or Todd P. Michael.

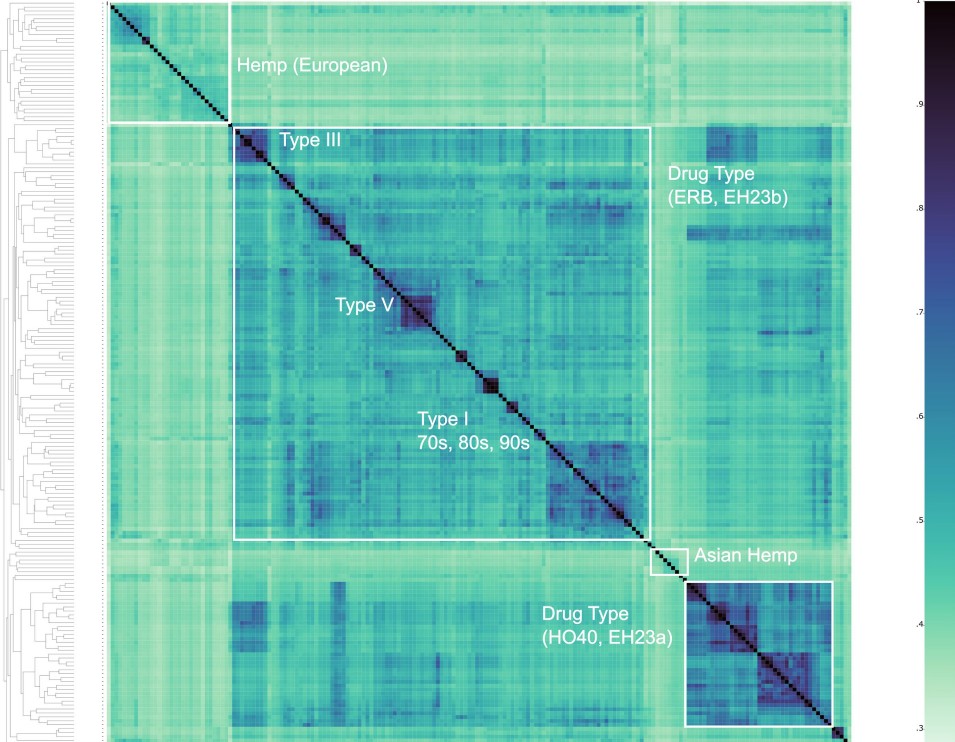

**Extended Data Fig. 1 | PanKmer Jaccard similarity matrix of 193 Cannabis genomes.** PanKmer (PK) was used to estimate the relationship between the genomes in the cannabis pangenome. A large portion of the pangenome included elite cultivars, breeding trios and foundational Marijuana (MJ) lines originating from breeding programs spanning the 1970s to present (Supplementary Fig. 1; Supplementary Table 1). These samples represented chemotypes showing high expression of pentyl or propyl (varin) homologs of CBDA or THCA, and cannabinoid free (type V) plants. Flowering time variation was also captured with the inclusion of both short-day (SD) and DN phenotypes.

The remaining cultivars came from the United States Department of Agriculture (USDA) Germplasm Resource Information Network (GRIN) and German federal genebank (IPK Gatersleben) repositories to ensure researchers will have access to plants for experimentation. These samples included European and Asian fiber and seed hemp, feral populations, North American marijuana (type I), hc yielding (CBDA or CBGA) hemp (type III and IV), male plants (XY; Fig. 1b) and monoecious plants (XX; Supplementary Table 1). Together, this comprehensive dataset provides a foundation for exploring cannabis genomic diversity, hybridization, and trait evolution. See Figshare for full resolution version.

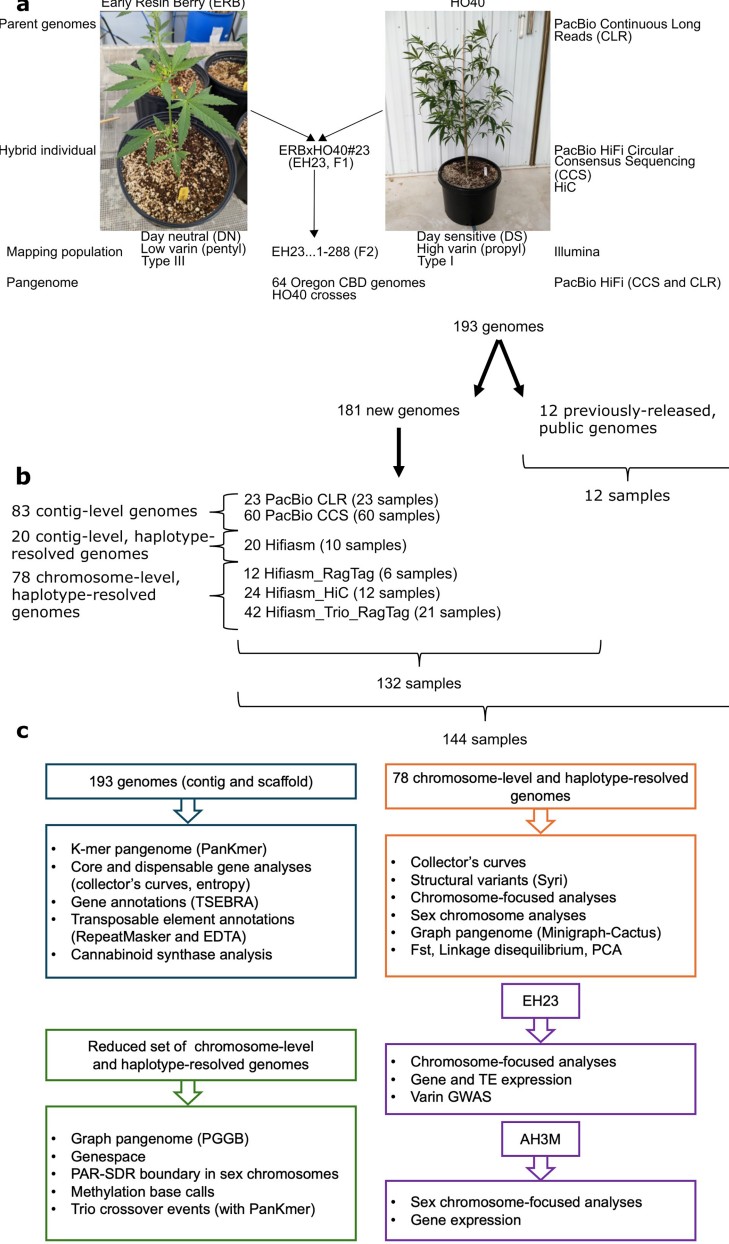

**Extended Data Fig. 2 | The EH23 anchor genome sequencing strategy and resulting populations.** A) The F1 hybrid EH23 (ERBxHO40#23) was generated by crossing the type III (high CBDA), day neutral (DN), Early Resin Berry (ERB) with the type I (high THC), day sensitive (DS), HO40. Both ERB and HO40 were sequenced with PacBio CLR, while EH23 was sequenced with PacBio HiFi (CCS) and scaffolded with High-throughput Chromatin Conformation Capture (Hi-C). The F2 mapping population (288 individuals) was sequenced with Illumina short reads. The remaining pangenome samples from OCBD are summarized in (Supplementary Table 1) with a pedigree chart (Supplementary Fig. 1). B) Organization schematic for 193 genomes of the cannabis pangenome. Two methods were used to achieve haplotype-resolved, chromosome-scale genomes. The first, a streamlined method, employed Hi-C data for both phasing and

scaffolding (Supplementary Table 1, Methods), generating 24 haploid genomes from 12 samples (Hifiasm_HiC). These served as scaffolding references for 42 genomes from 21 samples (Hifiasm_Trio_RagTag), resulting in trio-phased haploid assemblies. Together, these 78 genomes serve as the foundation for our pangenome analyses of transposable elements and structural variation. Additionally, we generated 20 haplotype-resolved contig-level assemblies (Hifiasm), along with 83 contig-level assemblies using older PacBio continuous long reads (CLR; 23 assemblies) and circular consensus sequencing (CCS; 60 assemblies) (Supplementary Table 1). C) Diagram of genomes used in different analyses for this study. For all assemblies we generated gene model annotations using both *ab initio* tools and RNA expression data, as well as TEs called using a RepeatModeler library (Supplementary Table 2, Methods).

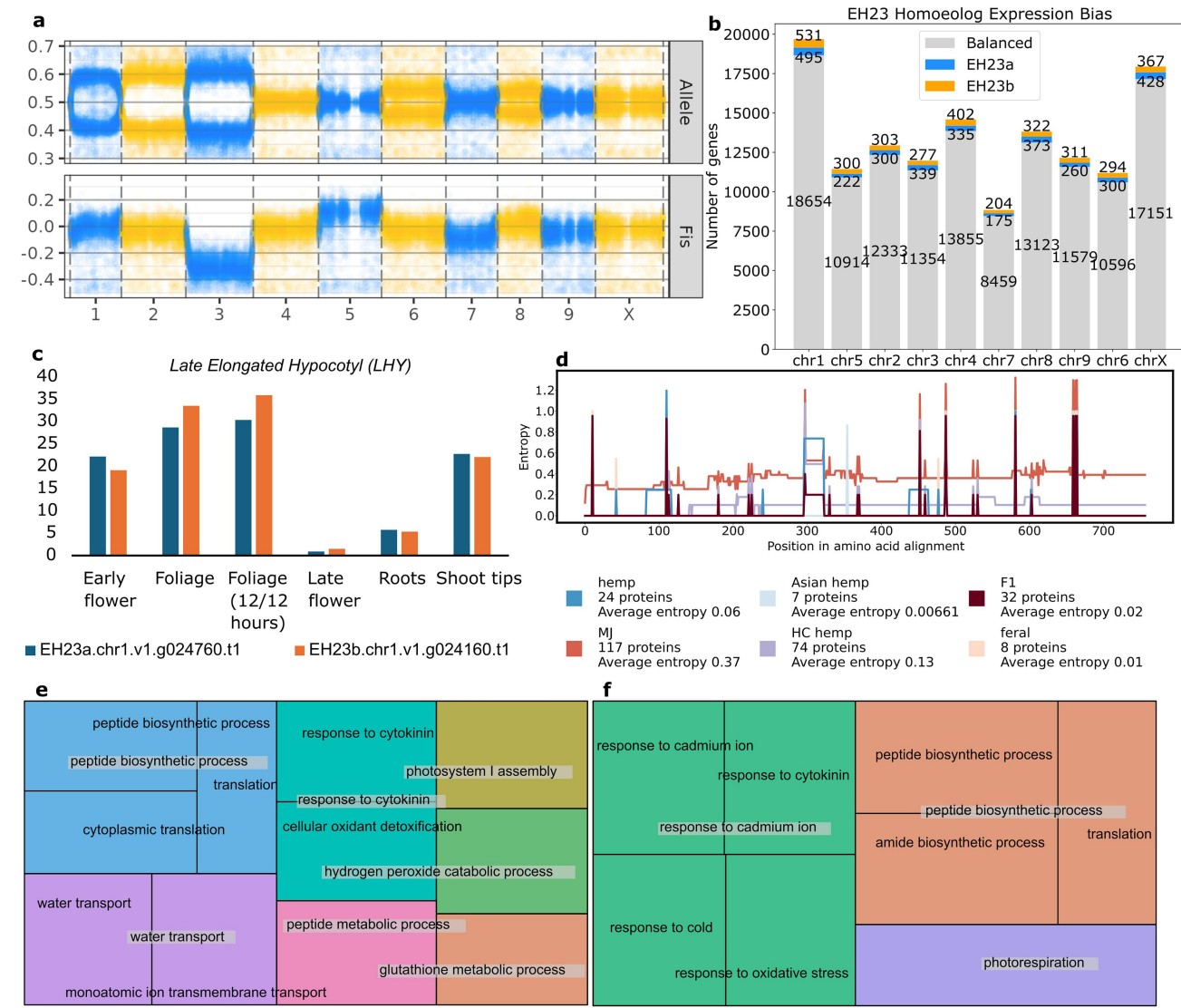

**Extended Data Fig. 3 | F1 hybrid (ERBxHO40_23; EH23a and EH23b) between two phenotypically and genetically divergent parents clarifies features of the genome missed in other studies to-date.** A) Inheritance of alleles across the genome from the F2 population. The upper panel presents the frequency of each allele and the lower panel shows FIS or the deviation from our evolutionarily neutral expectation of heterozygosity. B) Haplotype specific expression for all tissue types from EH23, grouped by chromosome. Haplotype gene pairs were either syntenic or reciprocal best hits. Balanced and biased gene expression was assigned according to TPM difference. A difference threshold of 5 TPM was required for gene pairs to be assigned as biased, otherwise gene pairs were assigned as balanced (see also Supplemental Table 2 for counts by tissue type). C) *LATE ELONGATED HYPOCOTYL (LHY)* showed biased gene expression in EH23b foliage under 12 h of light (12/12 h). D) The copy of LHY with biased expression also belonged to an orthogroup with high entropy in different populations, with the largest difference in entropy separating feral and MJ. E) GO term enrichment of biased gene expression for all tissues in EH23a; and F) GO term enrichment of biased gene expression for all tissues in EH23b. See also Supplemental Note 2.

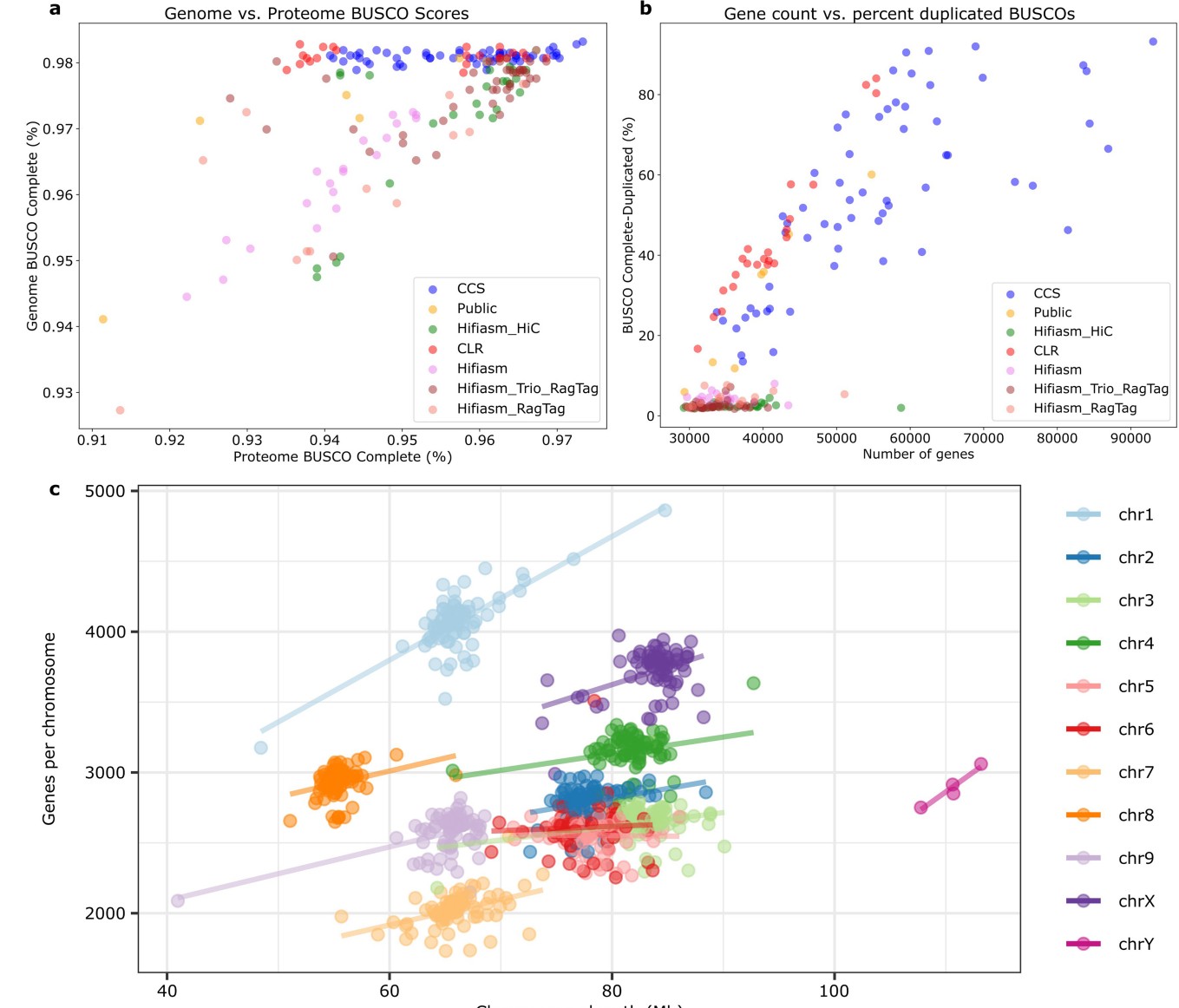

**Extended Data Fig. 4 | The cannabis pangenome and pangenes are high quality.** A) Benchmarking Universal Single-Copy Orthologs (BUSCO)[19] for both the genome and gene predictions suggest that they are both high quality and complete. Gene models were predicted based on homology and expression data from different tissues, including flowers, leaves, and roots (Supplementary Table 2) with TSEBRA. We evaluated the quality of gene models with BUSCO[19], which were around 95% complete on average for all assembly types. The scaffolded genomes contained 35,000 genes on average, and in the contig genomes, the number of genes scaled with the presence of duplications detected by BUSCO (Fig. 1e). B) The number of genes predicted contrasted with the number of BUSCO duplicate genes suggesting that the CCS and CLR contig-based assemblies were retaining significant duplicated sequence due to uncollapsed haplotypes. These haplotypes were not removed to retain the level of variation for downstream analysis. C) Scatter plot of chromosome lengths on the x-axis compared with gene counts per chromosome on the y-axis across the nine autosomes and both sex chromosomes.

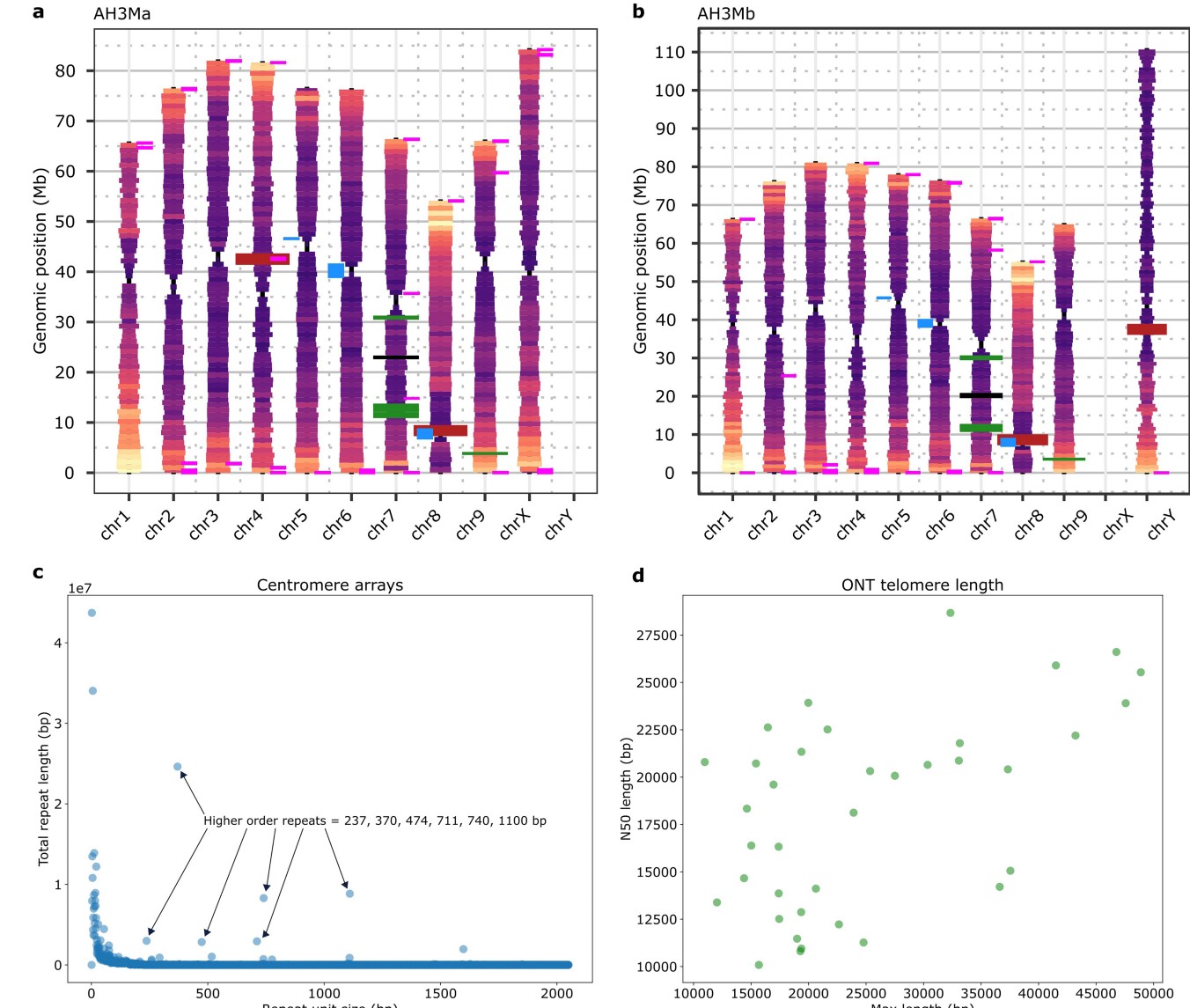

**Extended Data Fig. 5 | Cannabis centromere and telomere analysis shows higher order repeat structure.** A-B) The AceHigh3 (AH3M) chromosomal features of nine pairs of autosomes and one pair of sex chromosomes (X and Y). One million base pair rectangular windows extend outward from each pair of haplotypes at a width proportional to the absence of the CpG motif. Each rectangular window is colored by gene density with warm colors indicating high gene density and cool colors indicating low gene density. Each pair of haplotypes is connected by polygons indicating structural arrangement, with gray for syntenic regions and orange connecting inversions. Rectangles along each haplotype indicate select loci, including 45S (26S, 5.8S, 18S) rDNA arrays (firebrick red), 5S RNA arrays (black), 237 bp centromere repeat (blue), 370 bp CS-1 sub-telomeric repeat (pink) and cannabinoid synthases (forest green; *CBCAS, CBDAS, THCAS*, and *OAC*). Chromosomal plots for all 78 haplotype-resolved,

chromosome-scale genomes show similar trends (see Ideos.pdf at https://doi.org/10.25452/figshare.plus.28405079.v1). C) The centromere arrays identified in the AH3M genome (as an exemplar for the pangenome) with Tandem Repeat Finder (TRF). Two high copy number arrays were identified with base repeats of 237 and 370 bp, along with their higher order repeats (HOR). The 237 bp array is sparsely found in the genome (blue, panel A), although usually proximal to the high "CpG" sites. The 370 bp repeat is the same sequence as the sub-telomeric repeat CS-1[106] and found on the ends of the chromosomes (pink, panel A). D) A subset of the genomes were sequenced on Oxford Nanopore Technologies to estimate the telomere length in cannabis genomes[103]. The N50 ONT read length is plotted as a function of the max telomere repeat identified using the TeloNum software[103].

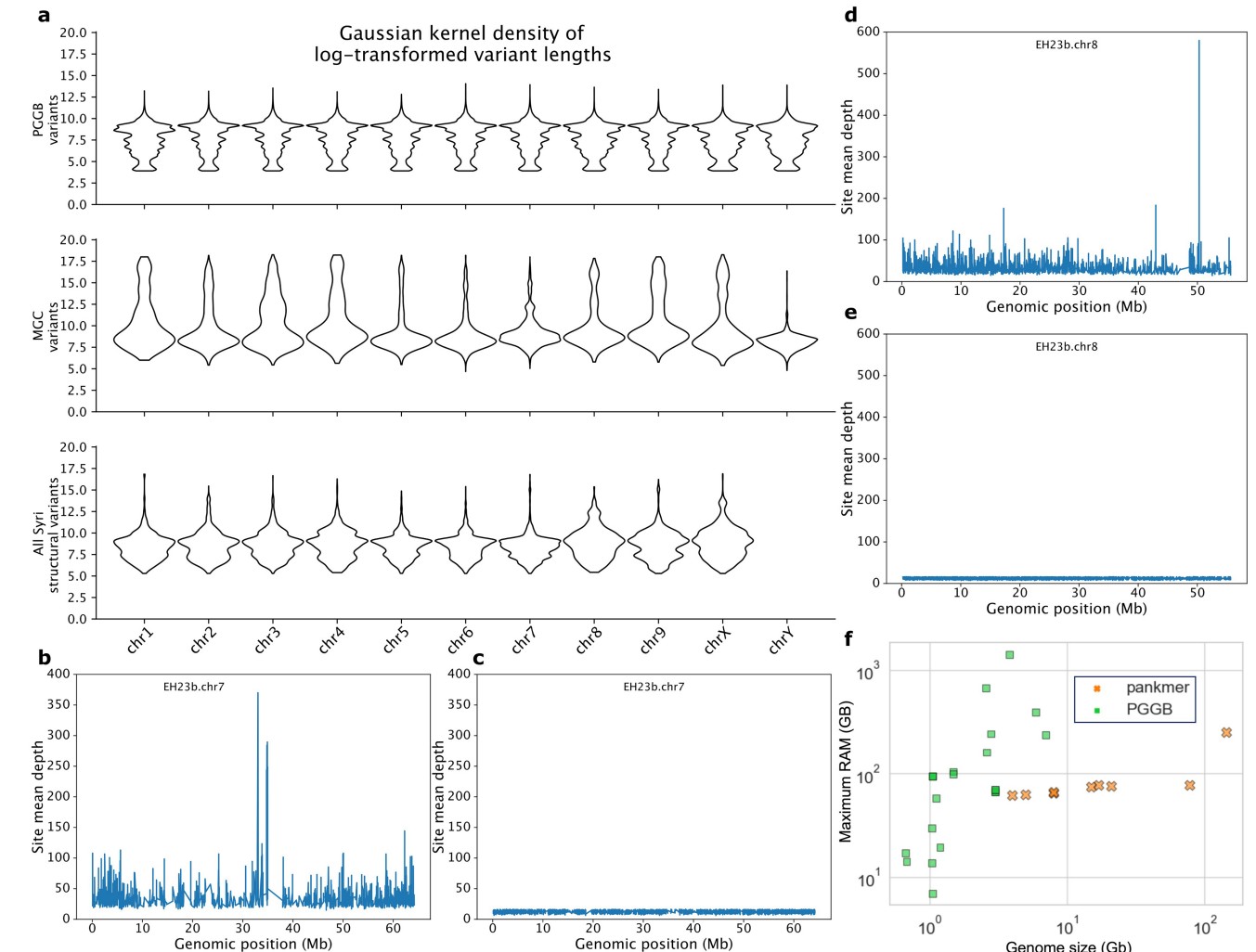

**Extended Data Fig. 6 | Comparison of Syri, Pan Genome Graph Builder, and Minigraph-Cactus structural variation (SV) calls.** A) Differences between Syri SV, Pan Genome Graph Builder (PGGB), and Minigraph-Cactus (MGC) variant lengths. This is a violin plot showing Gaussian kernel density estimates for PGGB variant lengths, Minigraph-Cactus variant lengths, and Syri SV lengths (all SV types combined, including duplications, inversions, inverted translocations, and translocations). The input data are log-transformed variant lengths. Lengths are log-transformed due to the very large range between smallest and largest lengths. The highest probability region in the violin plot is approximately at the same density for all three methods (-8). MGC shows a smoother distribution than Syri and PGGB. PGGB appears to be the most granular method, with more distinct groupings than the other methods. PGGB discovers more short variants, while MGC and Syri capture variants >= 50 bp. For comma-separated variants in the VCF file ("ALT" column), only the longest of the variants was counted. Plots showing average depth of EH23 F2 population short reads mapping to B) EH23b chromosome 7 as represented in the MGC pangenome graph; C) the linear reference sequence of EH23b chromosome 7; D) EH23b chromosome 8 as represented in the MGC pangenome graph; and E) the linear reference sequence of EH23b chromosome 8. F) Plot showing the maximum computational memory (RAM in units of gigabytes [GB]) required for analyzing pangenomes of varying sizes (in units of gigabases [Gb]) using PGGB and PanKmer.

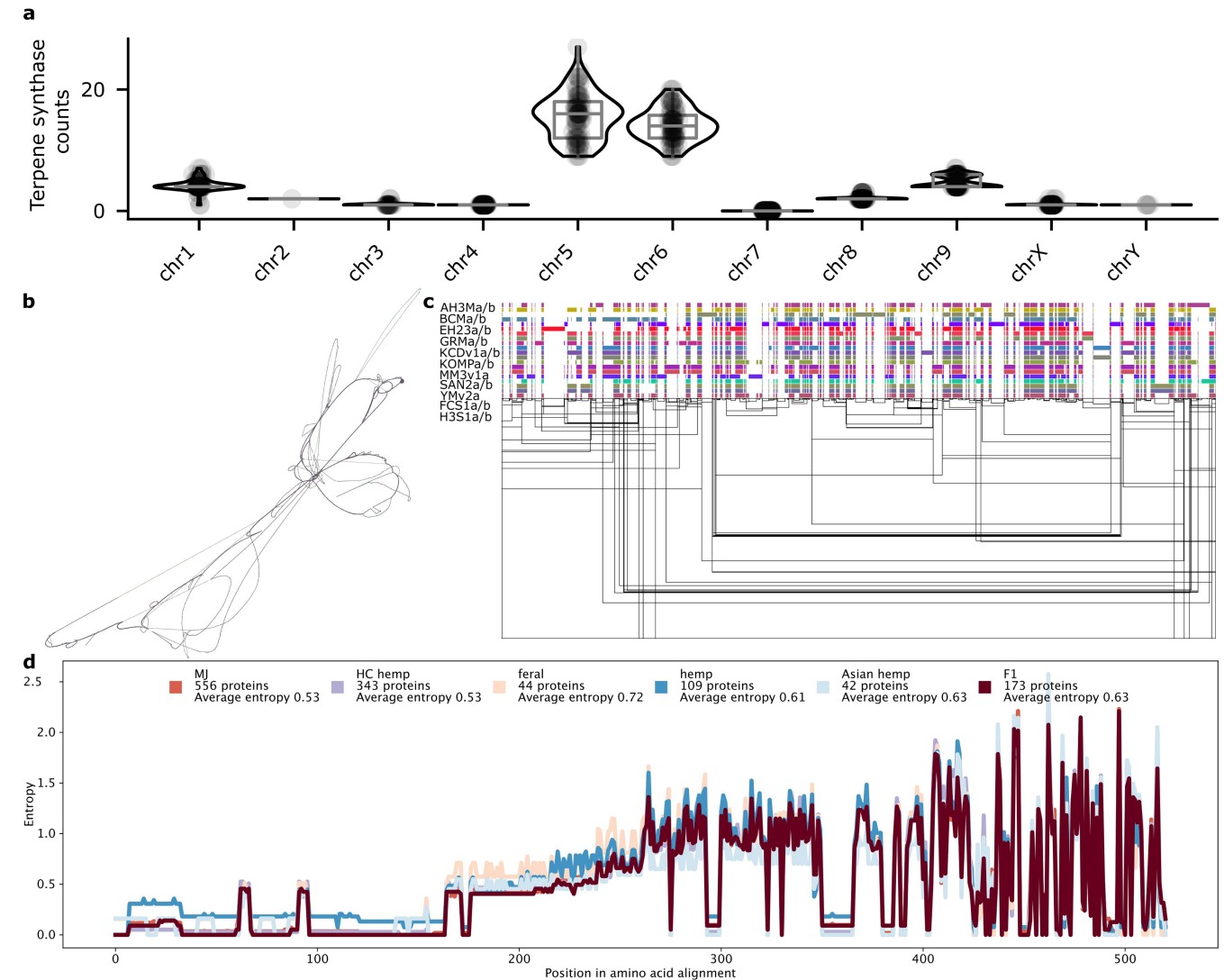

**Extended Data Fig. 7 | Terpene synthase genes across the cannabis pangenome.** A) Violin plot showing terpene synthase copy number in the cannabis pangenome. Chromosomes 5 and 6 are copy number "hotspots" in the cannabis pangenome. B) Odgi 2D visualization of EH23a.chr6.v1.g321150. t1, the highest expressed terpene synthase in all flower samples on EH23a.chr6, from reference-free PGGB pangenome graph (PGGB graph of chromosome 6 including AH3Ma/b, BCMa/b, EH23a/b, GRMa/b, FCS1a/b, H3S1a/b, KCDv1a/b, KOMPa/b, MM3v1a, SAN2a/b, YMv2a). C) Pangenome variation graph visualization of EH23a.chr6.v1.g321150.t1, showing interspersed regions of variation across the gene sequence. D) Visualization of entropy values for protein multiple sequence alignment showing low variation at the beginning of the alignment and high variation towards the end of the alignment.

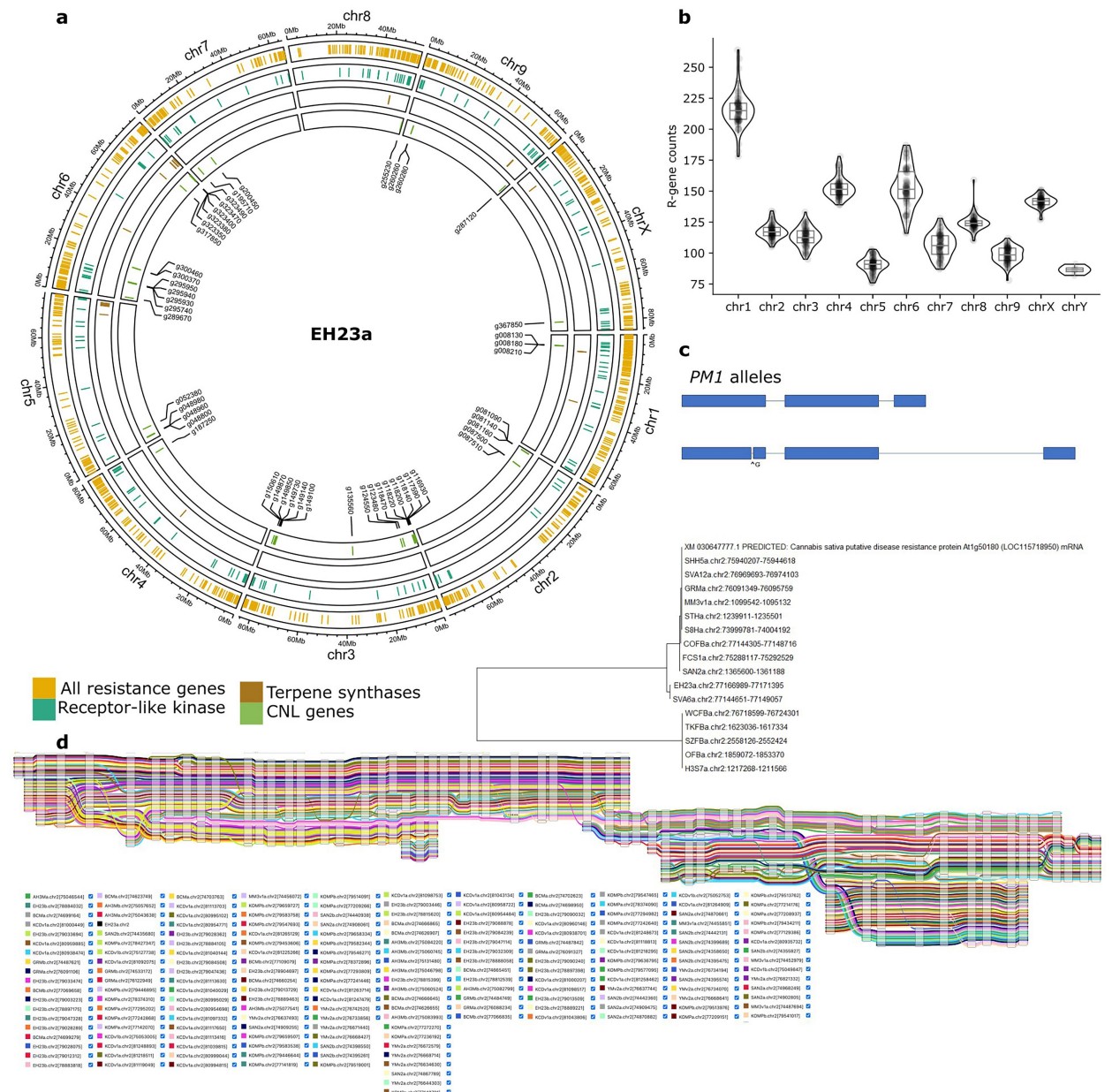

**Extended Data Fig. 8 | Disease resistance genes across the cannabis pangenome.** A) Circos plot showing the EH23a genome as an example of the chromosomal distribution of disease resistance gene analogs (RGAs). Outer track (gold)=all categories of RGAs identified by drago2; middle track (blue)= receptor-like kinases; interior track=coiled-coil nucleotide binding site leucine-rich repeat genes. B) Violin plot showing numbers of RGAs per chromosome in chromosome-level, haplotype resolved genomes. C) Maximum likelihood tree of coiled-coil NBS-LRR (CNL) genes on chromosome 2 with similarity to a gene associated with powdery mildew resistance. D) Sequence tube map visualization of gene near PM1 marker (EH23a.chr2.v1.g115410; EH23a.chr2:77164374-77165978).

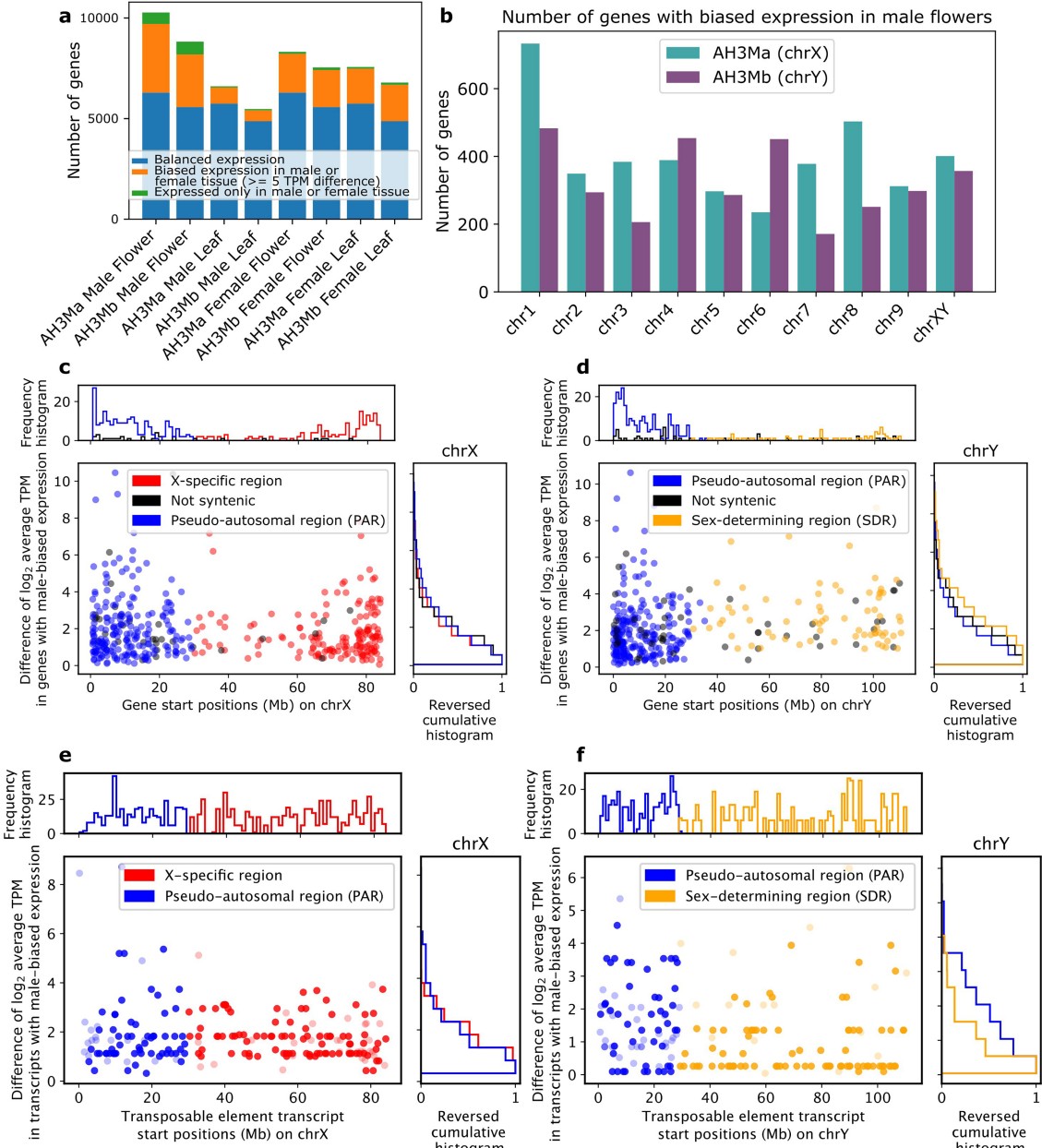

**Extended Data Fig. 9 | Expression patterns in the flowers and leaves of male and female AceHigh (AH3M) plants.** A) Stacked bar chart showing the number of genes with balanced, biased, or exclusive expression in male and female tissues. Overall, for a gene to be considered expressed, a minimum average TPM value of 1.0 across tissue replicates was required, grouped by sex. For balanced expression, genes were required to have a minimum average TPM of at least 1.0 in both sexes, grouped by tissue type, while also having less than a difference of 5 TPM between each sex. For biased expression, a difference of >= 5 TPM between sexes was required for each tissue type. For exclusive expression, a gene was required to have a minimum average TPM of at least 1.0 in one sex for a given tissue, without expression in the other sex for that tissue type (TPM = 0). On average, approximately 90% of genes with balanced or biased expression were syntenic across tissues and sexes; in contrast, approximately 80% of genes with exclusive expression were syntenic. The main exception was exclusively-expressed genes in female leaf tissue, in which approximately 90% of genes were syntenic. For this analysis, synteny was relative to the set of eight genomes with X and Y chromosomes, determined by GeneSpace. B) Chromosome-level counts of genes with biased expression in male flowers. C) and D) Scatter plots showing biased gene expression in male flowers across chromosomes X and Y, respectively. The x-axis shows gene start positions and the y-axis shows the difference of $\log_2$ TPM between male and female flowers, specifically showing genes with biased or exclusive expression in male flowers. The blue markers correspond to genes in the PAR and red markers correspond to genes in the X-specific region. E) and F) Biased expression of intact TEs in male flowers across chromosomes X and Y, respectively. GO term enrichment among genes with biased and exclusive expression in male flowers included a variety of metabolic pathways, including pollen development.

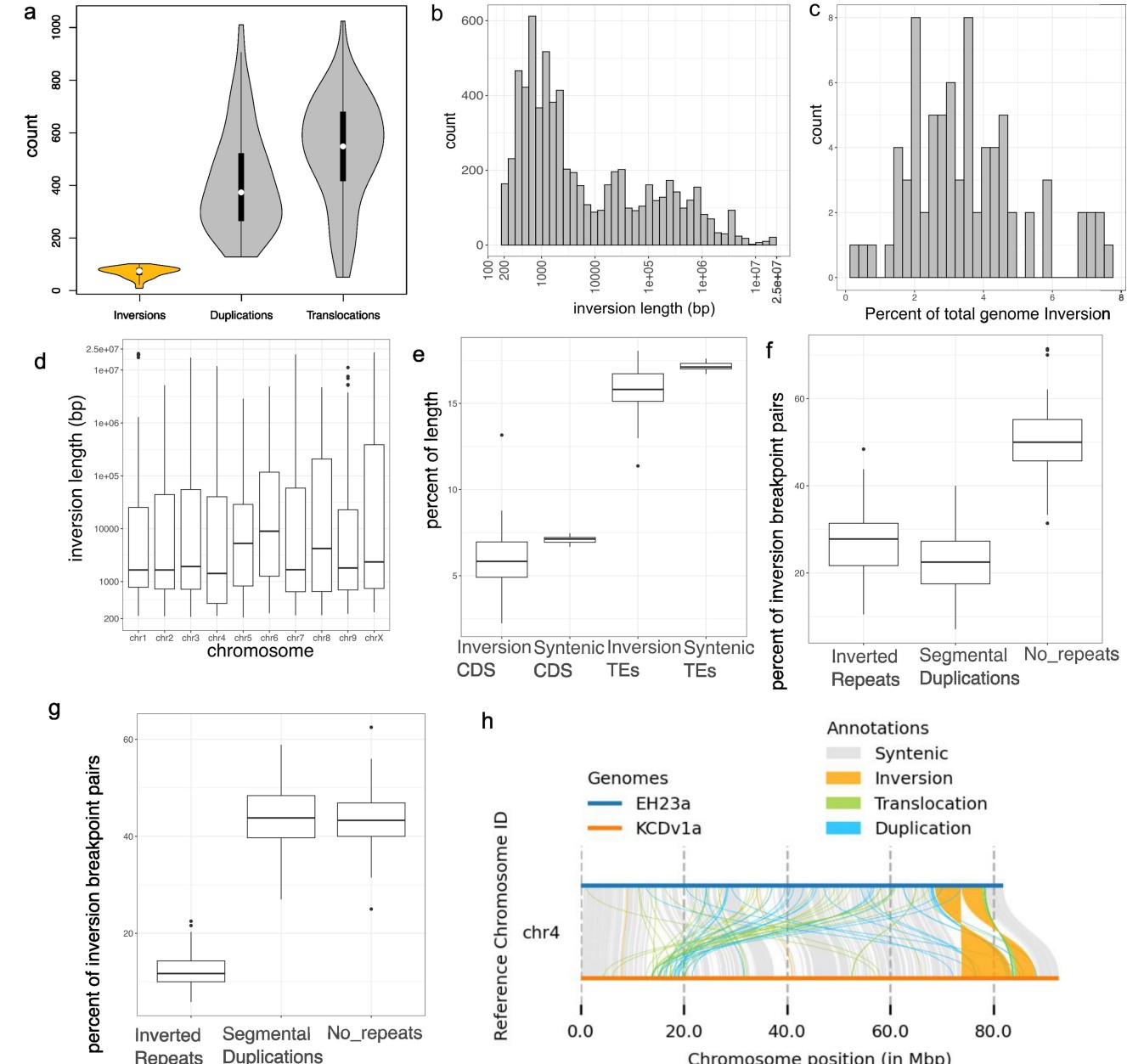

**Extended Data Fig. 10 | Cannabis pangenome reveals a wide range of structural variation (SV), on par with some of the values that have been reported for interspecies comparisons.** A) Distributions of three types of SVs across the 78 scaffolded assemblies of the cannabis pangenome. Each sample assembly was aligned to the EH23a haplotype assembly for SV calling. B) Multi-modal distribution of inversion lengths, for all inversions from all samples. C) Distribution of the total length of inversions in each assembly as a percent of total genome length. D) Distributions of inversion lengths, for all inversions from all samples. E) Distributions of coding sequences (CDS) and intact transposable elements (TEs) within all inversions and syntenic regions from each sample. Inversions are significantly depleted of CDSs compared to syntenic regions, while on average, TEs are present at nearly an equal level within inversions and syntenic regions. F) Inversion breakpoint (BP) pairs, defined as 8 kb windows centered at the start and end of each inversion larger than 10 kb, contain repetitive elements about 50% of the time. G) Inversion BPs show a higher rate of segmental duplications, but lower rate of inverted repeats, within self-to-self alignments for each 8 kb BP window, compared to the start-to-end pair alignments. F) Example alignment and SVs of a European hemp sample haplotype (KC Dora). The two mega base scale inversions are in a region of chromosome 4 that showed elevated $F_{st}$ values for SNPs in prior work comparing feral US hemp to marijuana populations[157].

# Reporting Summary

## Statistics

For all statistical analyses, confirm that the following items are present in the figure legend, table legend, main text, or Methods section.

| n/a | Confirmed | |
|---|---|---|
| ☐ | ☒ | The exact sample size (*n*) for each experimental group/condition, given as a discrete number and unit of measurement |
| ☐ | ☒ | A statement on whether measurements were taken from distinct samples or whether the same sample was measured repeatedly |
| ☐ | ☒ | The statistical test(s) used AND whether they are one- or two-sided<br>*Only common tests should be described solely by name; describe more complex techniques in the Methods section.* |
| ☐ | ☒ | A description of all covariates tested |
| ☐ | ☒ | A description of any assumptions or corrections, such as tests of normality and adjustment for multiple comparisons |
| ☐ | ☒ | A full description of the statistical parameters including central tendency (e.g. means) or other basic estimates (e.g. regression coefficient) AND variation (e.g. standard deviation) or associated estimates of uncertainty (e.g. confidence intervals) |
| ☐ | ☒ | For null hypothesis testing, the test statistic (e.g. $F$, $t$, $r$) with confidence intervals, effect sizes, degrees of freedom and $P$ value noted<br>*Give P values as exact values whenever suitable.* |
| ☒ | ☐ | For Bayesian analysis, information on the choice of priors and Markov chain Monte Carlo settings |
| ☐ | ☒ | For hierarchical and complex designs, identification of the appropriate level for tests and full reporting of outcomes |
| ☐ | ☒ | Estimates of effect sizes (e.g. Cohen's *d*, Pearson's *r*), indicating how they were calculated |

*Our web collection on statistics for biologists contains articles on many of the points above.*

## Software and code

Policy information about availability of computer code

| Data collection | Sequencing data were recorded as fastq files (see Methods), quantitative phenotypic data was collected manually in fields and greenhouses and recorded in Microsoft Excel (v18.85) (see Methods). |
|---|---|
| Data analysis | Each analysis in our study uses software that is described and referenced in our Methods section. Version information, parameters, and settings are also provided to allow for a clear understanding of how each analysis was performed. Scripts and analysis pipelines are available at https://github.com/anthony-aylward/CannabisPangenomeShared and https://github.com/padgittl/CannabisPangenomeAnalyses.<br><br>3D-DNA pipeline (version 180922), Apptainer (v1.1.8), assembly-stats (v1.0.1), bedops , bedtools , Bcftools , Blastn , Braker (v2.1.6), BUSCO (v5.4.3), Cactus (v2.6.7), CD-HIT , diamond (v2.1.4), Dorado (v0.3.4), EDTA (v1.9.6), eggNOG-mapper (v2.0.1), FALCON Unzip (PacBio SMRT Tools 9.0), FastTree , fastp , Freebayes , GAPIT (v3), Geneious Basic , genespace (v0.9.3), Hifiasm (v0.16.162), HiCanu (v2.2), HISAT2 (v2.2.1), IQ-TREE (v1.6.12), Juicebox (v1.11.086), Juicer (v1.6.2), MAFFT (v7.505), MEGA (v11), Minimap , Minimap2 (v2.24), Minigraph-Cactus , modkit , Nextflow (v24.04.3.5916), nf-core/pangenome (v1.1.2), OrthoFinder (v2.5.4), PanKmer , PGGB , plink , plotsr , R (v4.2.2), RagTag (v2.1.0), Salmon (v1.6.0), samtools , snakemake , Sourmash (v4.6.182), syRI , Tandem Repeat Finder (v4.09), TopGO , TranslatorX , Trim Galore , TSEBRA , vcfbub , vcftools , vg (v1.61.0) |

For manuscripts utilizing custom algorithms or software that are central to the research but not yet described in published literature, software must be made available to editors and reviewers. We strongly encourage code deposition in a community repository (e.g. GitHub). See the Nature Portfolio guidelines for submitting code & software for further information.

## Data

Policy information about <u>availability of data</u>

All manuscripts must include a <u>data availability statement</u>. This statement should provide the following information, where applicable:

- Accession codes, unique identifiers, or web links for publicly available datasets
- A description of any restrictions on data availability
- For clinical datasets or third party data, please ensure that the statement adheres to our <u>policy</u>

**Data Availability**

The NCBI BioProject ID for the cannabis pangenome is PRJNA1140642. All of the pangenome sequencing data at NCBI SRA is under the BioProject accession PRJNA904266. The BioProject accession IDs for EH23a and EH23b are PRJNA1111955 and PRJNA1111956, respectively. Genomes and annotation files for all 193 assemblies, including links to corresponding U.S. National Plant Germplasm System accessions are available from: resources.michael.salk.edu. Orthobrowser and Genome Jbrowse instances are hosted at: resources.michael.salk.edu. Input and output files for graph pangenomes are available at resources.michael.salk.edu. Annotations for R-genes, terpene synthases, cannabinoid synthases, and additional genome visualizations are available from: figshare.com/projects/ Cannabis_Pangenome/205555 and https://doi.org/10.25452/figshare.plus.c.7248427.v1. Links to specific genome datasets are provided in Supplementary Table 1: https://doi.org/10.6084/m9.figshare.25869319.v1.

## Research involving human participants, their data, or biological material

Policy information about studies with <u>human participants or human data</u>. See also policy information about <u>sex, gender (identity/presentation), and sexual orientation</u> and <u>race, ethnicity and racism</u>.

| | |
|---|---|
| Reporting on sex and gender | NA |
| Reporting on race, ethnicity, or other socially relevant groupings | NA |
| Population characteristics | NA |
| Recruitment | NA |
| Ethics oversight | NA |

Note that full information on the approval of the study protocol must also be provided in the manuscript.

# Field-specific reporting

Please select the one below that is the best fit for your research. If you are not sure, read the appropriate sections before making your selection.

☐ Life sciences     ☐ Behavioural & social sciences     ☒ Ecological, evolutionary & environmental sciences

For a reference copy of the document with all sections, see <u>nature.com/documents/nr-reporting-summary-flat.pdf</u>

# Ecological, evolutionary & environmental sciences study design

All studies must disclose on these points even when the disclosure is negative.

| | |
|---|---|
| Study description | Cannabis sativa plants were cultivated and crossed in controlled enviroments and fields in, OR, CO and CA. |
| Research sample | Cannabis sativa pangenome samples were selected from multiple sources to maximize the genetic diversity, history and agronomic value. A large portion of the pangenome comes from the Oregon CBD (OCBD) breeding program that includes elite cultivars; foundational marijuana lines potentially originating from the 1970s, 80s, 90s to present; and elite trios used for different aspects of the breeding program. The remaining cultivars come from the United States Department of Agriculture (USDA) Germplasm Resource Information Network (GRIN) and German federal genebank (IPK Gatersleben) repositories, as well as collections made by the Salk Institute from various breeders. The pangenome includes European and Asian fiber and seed hemp, feral populations, North American marijuana (type I), and North American high cannabinoid yielding (CBD or CBG) hemp (type III and IV). Additional cannabinoid diversity is represented with chemotypes presenting high expression of pentyl or propyl (varin) homologs of CBD or THC, and cannabinoid free (type V) plants. Flowering time variation is also captured with the inclusion of both regular short day and day neutral (autoflowering) phenotypes (Supplemental Table 1). |
| Sampling strategy | In addition to the whole genome sequencing data described above, ERBxHO40_23 was self-pollinated using STS induced masculinization of select flowers, to create an F2 mapping population. From this F2 population, individuals were scored for autoflower, varin content, and sequenced using Illumina 100 bp reads by NRGene (Nrgene Technologies Ltd, Israel). Illumina WGS |

genotyping runs were performed on 288 plants from this population, plus the ERBxHO40_23 parent.
Sample size for this population was dictated by available greenhouse space and assessment of prior mapping studies.

**Data collection**
For HMW DNA extractions leaf material was sampled as described in the methods. For RNAseq libraries samples were collected from six tissues during the course of of the ERBxHO40_23 individual: development: shoot tips, roots, late flower, leaf under short day, leaf under long day, and early flower

**Timing and spatial scale**
For RNAseq libraries, late flower = 8 weeks under 8-hour light / 16-hour dark conditions, leaf under short day = 4 weeks under 8-hour light and 16-hour dark, leaf under long day = 4 weeks under 12 hour light / 12 hour dark, and early flower 2 weeks under 8-hour light / 16-hour dark conditions.

**Data exclusions** NA

**Reproducibility** NA

**Randomization** NA

**Blinding** NA

Did the study involve field work? ☐ Yes ☒ No

# Reporting for specific materials, systems and methods

We require information from authors about some types of materials, experimental systems and methods used in many studies. Here, indicate whether each material, system or method listed is relevant to your study. If you are not sure if a list item applies to your research, read the appropriate section before selecting a response.

## Materials & experimental systems

| n/a | Involved in the study |
|---|---|
| ☒ | Antibodies |
| ☒ | Eukaryotic cell lines |
| ☒ | Palaeontology and archaeology |
| ☒ | Animals and other organisms |
| ☒ | Clinical data |
| ☒ | Dual use research of concern |
| ☐ | ☒ Plants |

## Methods

| n/a | Involved in the study |
|---|---|
| ☒ | ChIP-seq |
| ☒ | Flow cytometry |
| ☒ | MRI-based neuroimaging |

## Plants

**Seed stocks**
Links to corresponding U.S. National Plant Germplasm System accessions are available from: resources.michael.salk.edu

**Novel plant genotypes**
Breeding techniques are described clearly in the methods and a pedigree diagram is provided in Supplementary Figure 1.

**Authentication** NA

