## [Peer Review file · Nature]

Domesticated cannabinoid synthases amid a wild mosaic cannabis pangenome

Corresponding Author: Professor Todd Michael

Version 0:

Reviewer comments:

Referee #1

(Remarks to the Author)

The manuscript on "Domesticated cannabinoid synthases amid a wild mosaic cannabis pangenome" describes a pangenome analysis of 193 cannabis accessions, from which 181 were newly sequences. It is clear that this is a large effort and an useful tool for cannabis research.

My main concern about this manuscript is that it is very descriptive and reports to a some extent similar analysis to that performed earlier in the cannabis genome sequences that were available. I miss a comprehensive analysis of the results and implications for biology of cannabis and (potentially) other species. Although is has some original points I also miss the "ground breaking" science.

The data generated are of interest mostly to scientists working in cannabis species and potentially in related species such as hop.

For the reasons above I so not think that the manuscript has the quality nor the potential impact expected for a Nature publication. Therefore I so not endorse the publication of this manuscript in the current form in Nature.

Referee #2

(Remarks to the Author)

This study reports the pangenome of *Cannabis sativa*, a plant species that is relevant for fiber, drug and seed oil production. Widespread prohibition has restricted much of breeding and development of both genetic and genomic resources so far.

This work tries to shed light on genomic diversity and evolution, as well as on genes and gene families involved in the main synthase pathways. Main findings of this study include the detection of widespread structural variation (including some analyses on associated TEs), haplotype specific expression patterns, five distinct cannabis populations, lineage-specific TE expansion and diversification and patterns of contrasting diversity within the cannabinoid pathway.

Overall the results and conclusions are well supported by the data generated and the majority of analyses appears to be robust. The paper is well written and I'm confident that both the resources and results could have an impact on hemp breeding and utilization.

While the novelty of the results is in parts somewhat incremental to what has been reported before, there are areas such as the SVs and population structure that clearly require the pangenome.

I do have a few major points to address:

a.) On one hand side I appreciate that the authors utilize their newly generated pangenome resources in a slightly different way as done in many other pangenome studies these days. However, I don't really get the point why using anchor genomes would generally (or just in this study? And why then?) be preferable over constructing a "true" pangenome. Is it because of feasibility, your particular data structure (which consists of genomes of fairly different quality and contiguity) or is it beneficial/required to have a single reference to map to for certain analyses?

What does it mean that "...anchor genomes achieve similar high quality..." (line 117)? In what context? Compared to what? Is there a reference to cite for this claim?

b.) With the data framework in hand and a reasonably small genome size, wouldn't this be a good opportunity to try to construct a pangenome graph (I'm aware of the technical difficulties associated with...but it could eventually be exploited in some of the analyses)?

c.) Looking at Figure 1A it appears to me that there are less genes identified on chromosome 4 of haplotype 2. Is there an explanation for this?

d.) Expression analysis: I don't think Figure 1E is a good way to claim that haplotype EH23a has increased expression over haplotype b. More relevant would be how many genes are higher expressed, and what these genes are. What does the y-axis actually show?

Figure 1C: I assume only alleles/genes present in both haplotypes were analysed here...how do you reliably distinguish between identical/near identical orthologs?

Figure 1D: this is a bit misleading as it gives the impression that more than the aforementioned 10% of genes are actually affected by any type of haplotype bias.

Please add more explanations in the figure legends for Figures 1C-D (Y-axis!).

e.) Selection of lines for the pangenome: I understand that the authors have been limited in their choice by germplasm availability and/or current knowledge of hemp diversity. To get a better idea about the portion of diversity this pangenome actually covers it would be great to show a PCA (or similar) illustrating cannabis genetic diversity and the position of the selected lines within.

f.) The authors state that the pangenome should help shedding light on sex chromosome evolution in cannabis...I find very little on this aspect in the current manuscript.

g.) Figure 2D needs more explanation...what is shown with the colors here? Isn't there a more intuitive (clear) way of visualizing this (such as STRUCTURE plots)?

h.) gene annotation and construction of orthologous groups: I much appreciate the rationale to compensate for missed genes in the gene prediction procedure. However, after considering non-overlapping mappings as unannotated genes, I disagree to simply mark the "corresponding" (how was that determined?) orthogroup as present. Given the mapping criteria, orthology is not guaranteed unless determined by a new OrthoFinder run incorporating the recovered gene model.

i.) I'm not convinced about the full comparability (especially for the type of analysis performed here) between the cannabis and the *Humulus* genomes, as the later is of lower quality and contiguity.

j.) The TE part is interesting but in some parts fairly speculative (esp. line 316-324).

k.) Pan-gene analyses: the findings related to both Terpene synthase and disease resistance genes are very incremental (basically just a catalog of genomic positions and some CNV). This analysis could be enhanced by looking at sequence/expression diversification in the pangenome.

In addition to the targeted gene analyses, it would be interesting to learn at least something about general gene conservation and variability in the pangenome (aka core/shell/cloud gene portions).

l.) Data Availability: what does "pending" mean? Was the data submitted and accessions numbers assigned?

m.) I'm not sure the title is the best representation of the main work conducted in this study.

Referee #3

(Remarks to the Author)

In this manuscript, Lynch et al. report their work on developing a cannabis pangenome resource by generating a large number of haplotype resolved-chromosome scale assemblies spanning geographic origin, end use, and cannabinoid type. The authors perform extensive analysis of structural variation and find the region containing the cannabinoid synthase genes to be relatively conserved compared to other regions with large rearrangements or structural variants. Additional features such as transposable elements, resistance genes, and terpene synthase genes are characterized. A GWAS was conducted to provide insight into the genetic architecture of varin production in cannabis.

The manuscript presents original data and insight and the authors have used state-of-the-art approaches. The use of statistics throughout the manuscript appears appropriate and most of the conclusions are well supported. Many of the genomes are not accessible at the time of the review other than the 44 currently available at resources.michael.salk.edu but all code and software links work. The pangenome vastly expands the available genomic resource available in cannabis and will be widely used in the research community. The authors highlight the high level of species variation in cannabis and stress the need for germplasm preservation, especially in Asia, which is the proposed center of origin. Since cannabis is so embedded in human culture, this work will be broadly interesting to wide audience and is especially timely due to the rapidly changing legal landscape.

With that said, I have several suggested improvements to increase the manuscript quality and utility.

Main concerns

1. Since this is a pangenome paper, I'd like to see more comparisons across the whole pangenome or at least the 78

haplotype resolved-chromosome scale assemblies. How many core genes, dispensable genes and private genes are there? I'd like to see more statistics about how much more sequence/genes were added to the former reference. Could also add a genespace visualization between the identified subpopulations to showcase major structural variants.

2. Due to the outcrossing nature and high heterogeneity across the cannabis species, do you think having only one sample representing a subpopulation is sufficient to make claims about its relationship to the other groups? For example, pairwise F_{st} in Supplemental Table 3 and several of the PCAs.
3. L 1461 - Why was the proximal gene (next to the MLO marker) used – is there reason to believe this gene is the likely candidate gene for resistance or just linked? Was the observed sequence variation in the pangenome associated with the resistance scores in Supplementary Table 1?
4. Including the wild Asian sample makes visualizing the relationship between the other samples difficult to interpret and it would be interesting to see a PCA with it removed. Or run admixture $K = 2 - 6$ to see how groups drop out.
5. Why was the mapping in the F2 done via GWAS versus linkage mapping? Since phenotyping was unreplicated how reliable are the phenotypes? Were all the F2 samples female? You mention phenotyping autoflowering – is this data included? There does not appear to be a corresponding GWAS. Please remove if not used. Do the results emphasize the utility of the pangenome, meaning do they fall in newly discovered regions. What about the other two significant hits on Chr 9 and the X chromosome?
6. There should be a more discussion of the TPS and R gene results as compared to what was already known in the literature. What is novel?
7. Please discuss how the pangenome will be maintained and how additional data can be added/supplemented.

Minor concerns

1. Traditionally when discussing crosses in plant breeding, it is customary to write the female parent first. Therefore, it may make more sense for the masculinized parent to be written second (HO40#23 x ERB)
2. Supplemental Figure 1 is very important and is too difficult to see both in the manuscript and at the figshare link. Please improve the resolution and explain the red and blue lines.
3. It is unclear what counts is referring to in Figure 4B and if it is also representing the inversions too.
4. Why do you think chr 6 shows such long distance LD in Fig 4D? Is there anything interesting in this region?
5. Please expand on the EH23 RNASeq method section on page 26. How many plants were used and define tissue types more clearly.
6. Please add a color key for Figure 2D, change legend from Supplemental Figure 6d) D) to Supplemental Figure 6. D) and add percent variance explained to Figure 2E.
7. Can you add a frequency to each group in Figure 5C?
8. Why was 1kb selected for upstream and downstream analysis of TEs? It would be useful to look at the random distribution of TEs across different genomic features or even other genes to see if there is anything unique/interesting happening in the cannabinoid synthase genes.
9. It seems like Chr 8 should be flipped in Supplemental Fig 3B.
10. What are the apparent outlier samples for Chr 1 and 9 in Supplemental Figure 4.
11. The Supplemental Figure 6 link is only to the tree which does not appear to be the same as the one presented in the manuscript and the heatmap is missing.
12. Define line colors in Supplemental Fig 7B
13. The chr X label on the y axis in Supplemental Figure 12G and H is confusing. Would help to label all as X and Y or remove completely.
14. Can you add which bin the proposed centromere is located in the Supplemental Figure 14.
15. It is hard to understand or visualize where the elevated LD is in Supplemental Figure 18A. Please define the colors in 18B and add more info on the physical position to x axis for chromosome 1.
16. Why are the terpenes shown in different colors in the Supplemental Figure 19B?
17. What is the gene model representing in Supplemental Figure 19D? This is not well described in the text or figure legend.
18. I believe Supplemental Figure 22 is missing.
19. There appears to be an additional supplemental figure between Supplemental Figure 24 and 25.
20. Please define a, b, c, and d in Supplemental Table 9.
21. What is the parent 2 column in Supplemental Table 10 referring to?

Referee #4

(Remarks to the Author)

Lynch et al. report a comprehensive pangenome for cultivated cannabis, analyzing 181 new and 12 previously released genomes. They observed significant genetic diversity and propose a new understanding of cannabis population structure and hybridization history. Despite extensive genome-wide variation, they observed low diversity in critical cannabinoid synthase genes. I read this paper with interest and think there are many useful datasets and interesting analyses that will attract a broad readership. The paper could be strengthened for clarity, and many analyses are described in vague terms without detailed numbers or statistics, making it challenging to verify some of the claims. Below I have detailed my concerns which I feel when addressed will significantly strengthen the paper.

1. I am confused by the total number of genomes sequenced and what was included in the pangenome and various downstream analyses. The authors mention that a 'core' of 78 genomes were included in the pangenome, comprising 42 genomes from 21 samples and 24 haploid genomes, which does not sum to 78. It appears that the two distinct haplotypes from each cultivar or accession might be counted as separate genomes within the pangenome? Moreover, a larger pangenome of 193 genome assemblies is referenced throughout the paper, yet it is unclear which analyses pertain to the 78

genomes versus the 193 genomes, aside from their mention in the collector's curve.

2. The authors produced haplotype phased genome using both triobinning and HiC. Both of these approaches should produce more or less the same result, it is unclear why trio binning was performed, given the increased cost and complexity required to have parental genomes and their F1 sequenced.

3. Broadly, some of the analyses seem superficial or lack specific details. I recognize the authors may be aiming for brevity for the Nature format, but more details would be helpful.

For instance, the transposable element analyses describe TE abundance in the EH23 hybrid genome in detail, but differences in TE dynamics across the pangenome are only briefly mentioned. There are several TE analyses between Cannabis and Humulus and the authors use these differences to explain genome evolution, but these species diverged ~25 million years ago, making it difficult to compare centromere and TE differences and link them to the biology of these species. Similarly, the authors describe that structural variants are highly variable across the pangenome, but provide no statistics of how many, the average size, or how much of each genome is conserved in the anchor or dispensable. Figure 4 shows some really interesting results, but I do not think it is adequately described in the results. They claim that the number of inversions is similar to cross-species comparisons, but it would be great to have more details or data to support this.

I do not think it would take up much extra space to add these values and any relevant statistical test, but it would significantly strengthen the claims of this paper.

In the Terpene Synthesis section, the authors state that chromosomes 5 and 6 have terpene synthase hotspots but provide no details. How many are there? What is the copy number variation like across the genome? How does this link to the biology? Similarly, the authors describe that they annotated disease resistance genes, but provide no statistics or descriptions of how they vary across the pan-genome, simply that they developed a map of R genes and state that various diseases are emerging in Cannabis.

For the cannabinoid pathway section, I have a few questions. What does 'numerous pseudogenized paralogs' refer to? How many, how are pseudogenes being classified vs real genes, and how do these vary from THC producing and not producing accessions? Can the expression data be leveraged here to test for pseudogenization or subfunctionalization?

The authors claim that CBCAS paralogs appear not to be under strong selection. What evidence is there to support this? Numerous analyses could be done here to support this including Ka/Ks across the pangenome, or various selective sweep analyses using population data.

4. Line 260. Figure 2e seems to be a PCA or presumably the whole genome, but the results suggest that there are different relatedness based on different regions of the genome (e.g., chromosomes 2, 5, 7, and 9 are similar, the sex chromosomes show a different pattern). It seems that only a single value is plotted for each accession in this PCA. To better demonstrate the genetic structure suggested by the PCA, the authors might consider conducting population structure or admixture analyses, which could help verify the presence of the five distinct subpopulations illustrated in Figure 2C. From the PCA alone, distinctions among these subpopulations are not readily apparent, except possibly in the case of the EU hemp group. Additionally, the 'hc hemp' and 'feral' groups appear to be underrepresented in the genomic data, which might affect the clarity of subgroup distinctions.

5. The 25 Mb inversion in the Y specific region of the hemp sample Golden Redwood is quite interesting. I think this paper would benefit from a detailed analysis of the sex chromosomes in the pan genome of Cannabis. Generally, there are few pan-genome studies with species containing sex chromosomes (and perhaps none for plants?). This is talked about briefly throughout the paper, but one to two paragraphs discussing SVs, TE activity, gene differences in the non recombining regions compared to the rest of the genome would be of broad interest to readers.

Minor:

1. How much heterozygosity does a typical Cannabis cultivar/accession contain? Is EH23 a representative accession in this context or does it have higher diversity than most cultivars?

2. What is the X axis of Figure 3A showing? Is this percentage of the genome? Similarly for Figure 3B and C, does this show the age of different TEs across the pan genome or just EH23?

3. Line 216, how were the genomes annotated? This is described in the results, but one sentence here would be great, especially because genome annotation methods are quite contentious in the pan genome world.

4. Line 228, what does this mean? It would be helpful to report the average pairwise Fst or some averages for the 5 groups/populations to provide more context here. The Fst is presumably quite high,

5. Line 378. Is there any literature supporting this claim that 20 Mb inversion may function as supergenes in plants through overdominance?

Version 1:

Reviewer comments:

Referee #2

(Remarks to the Author)

The authors addressed most of my concerns in this revised manuscript. I especially appreciate the inclusion, comparison and discussion of graph representations of the cannabis pangenome. I believe this will be helpful for other researchers working in this area. Removing the anchor genome statement is fine: although I know, understand and appreciate the concept and use of an anchor genome here, I found its introduction quite confusing in the first version.

The only aspect I still find problematic is related to gene annotation: the authors state in their response that the gene set excluding the unannotated regions was used for grouping core vs dispensable genes. This is the type of analysis that would profit from a highly accurate and comparable gene annotation, that also contains the genes/regions which might have escaped the initial gene predictions. In any case this should be made transparent so that users can navigate the data sets accordingly.

Referee #3

(Remarks to the Author)

The authors have addressed my main concerns in this revised manuscript. In particular, they have added additional information on the core, dispensable and private genes, improved figures, made all data publicly available, and clarified questions regarding selected methods. The addition of the sex chromosome assembly data is very interesting and will offer insight into sex related traits in cannabis and sex chromosome evaluation more broadly. I have a few minor comments that may further help the clarity of the manuscript.

Minor comments:

Line 274 – Need period after Stack et al. 2025.

Figure 1B – what is the significance of the color (blue and red) for the X, Y and triangles? Also listing the Y-specific SDR as 79-84 Mb is confusing since the text says it spans 81 of the 110 Mb at Line 310). Also please explain what is meant by the X-specific region at 53-55 Mb?

What is the explanation for the increase in number in of genes for contig-level genomes versus haplotype phased genomes in Figure 1E?

In Figure 3C and 3D, it may be more meaningful to look at the 5 identified subgroups (Figure 1F) separately for LD as it is likely they have different decay rates.

Need to add new authors to author contribution section and likely update acknowledgements/funding.
Check ordering of chromosomes in Extended Data Fig 1B.

Please list color key in tube map for Supplementary Figure 14D as in Fig. 5D.

Supplementary Figure 18A – Please use more colors or clearly label the samples for the X and Y sample genotypes.

Referee #4

(Remarks to the Author)

The authors have addressed my previous concerns and I feel the revised manuscript is significantly improved in quality and clarity.

Nature manuscript 2024-05-09389

Referees' comments:

Referee #1 (Remarks to the Author):

The manuscript on "Domesticated cannabinoid synthases amid a wild mosaic cannabis pangenome" describes a pangenome analysis of 193 cannabis accessions, from which 181 were newly sequences. It is clear that this is a large effort and a useful tool for cannabis research. My main concern about this manuscript is that it is very descriptive and reports to a some extent similar analysis to that performed earlier in the cannabis genome sequences that were available. I miss a comprehensive analysis of the results and implications for biology of cannabis and (potentially) other species. Although it has some original points I also miss the "ground breaking" science. The data generated are of interest mostly to scientists working in cannabis species and potentially in related species such as hop. For the reasons above I do not think that the manuscript has the quality nor the potential impact expected for a Nature publication. Therefore I do not endorse the publication of this manuscript in the current form in Nature.

>>> We acknowledge your concerns about our manuscript and have substantially revised our work to emphasize its broad scientific impact, not only for cannabis research but also for the wider plant genomics community. Unlike previous efforts, which focused primarily on cannabinoid synthases, our research provides a genome-wide perspective that addresses fundamental questions about cannabis genetic diversity, structural variation, and evolutionary history. Our findings contribute significantly to plant biology in several ways:

→ Comprehensive Pangenome Analysis

- ◆ We constructed k-mer-based pangenomes and now in the revision add multiple graph-based that uncover genome-wide variation, enabling a deeper understanding of cannabis diversity beyond traditional linear references.
- ◆ Moreover, we have added identification of core and dispensable genes, highlighting the significance of the pangenome at the gene level.

→ New Insights into Cannabis Evolution and Diversity

- ◆ Cannabis diversity has been vastly under-sampled and misunderstood. Previous genome studies have been limited in scope, and our dataset represents the largest genome sampling effort of the genus. This allowed us to highlight the importance of sampling wild cannabis from Asia.
- ◆ Despite historical suppression, eradication efforts, and minimal public germplasm preservation, cannabis maintains an extraordinary degree of structural and genetic diversity.

→ Unprecedented Discovery of Fatty Acid Biosynthesis Pathway Variation

- ◆ Our study reveals that genetic and structural variations in the upstream fatty acid biosynthesis pathway underlie cannabinoid homolog variation and seed oil traits.
- ◆ Copy number and allelic variation in cannabis fatty acid biosynthesis genes exceed those found in major seed oil crops such as soybean and Brassica species, highlighting cannabis' untapped potential as a novel oilseed crop.

- ◆ Interestingly, we found that these variations may have been inadvertently preserved or enhanced in cannabinoid-rich drug-type gene pools, offering new directions for metabolic engineering and breeding.
- First Complete Assembly of Ancient Cannabis Sex Chromosomes
 - ◆ Cannabis harbors some of the largest and likely oldest heteromorphic sex chromosomes in flowering plants, yet these have never been fully assembled or characterized.
 - ◆ Our high-quality, haplotype-resolved assemblies allow us to define the PAR/SDR boundary in eight X/Y chromosome pairs for the first time, providing insights into their evolutionary dynamics.
 - ◆ We observed striking transposable element (TE) activity shaping the Y chromosome, uncovering biased gene expression in male flowers and patterns of syntenic gene retention that offer broader insights into sex chromosome evolution.
- Advancements in Pangenome Graph Construction
 - ◆ Cannabis' genome complexity, characterized by high repeat content, tandem duplications, and heterozygosity, presents challenges for graph-based assembly, which now we highlight.
 - ◆ We compare structural variants detected by Syri, Minigraph-Cactus (MGC), and PGGB, showing the advantages and limitations of different approaches. For cannabis we do not see an increased number of SNPs detected with the graph pangenome.
 - ◆ Graph pangenome construction still has its limitations, especially in a genome like cannabis with recent TE expansion, which exhausted our computational resources and required down sampling or masking of TEs.
 - ◆ We highlight that k-mer-based pangenomes enable discovery across a large set of genomes (193) and how this enables discoveries such as the varin locus.

While earlier studies have examined aspects of the cannabis genome, our work surpasses previous efforts in both scale and depth. The insights gained extend well beyond cannabinoid research and hold implications for plant evolutionary biology, sex chromosome research, and the domestication of complex secondary metabolic pathways. Given the historical and economic significance of cannabis, alongside its potential as a model for polygenic trait evolution and metabolic plasticity, we believe our study represents a groundbreaking contribution worthy of publication in Nature.

Referee #2 (Remarks to the Author):

This study reports the pangenome of *Cannabis sativa*, a plant species that is relevant for fiber, drug and seed oil production. Widespread prohibition has restricted much of breeding and development of both genetic and genomic resources so far. This work tries to shed light on genomic diversity and evolution, as well as on genes and gene families involved in the main synthase pathways. Main findings of this study include the detection of widespread structural variation (including some analyses on associated TEs), haplotype specific expression patterns, five distinct cannabis populations, lineage-specific TE expansion and diversification and patterns of contrasting diversity within the cannabinoid pathway. Overall the results and

conclusions are well supported by the data generated and the majority of analyses appears to be robust. The paper is well written and I'm confident that both the resources and results could have an impact on hemp breeding and utilization. While the novelty of the results is in parts somewhat incremental to what has been reported before, there are areas such as the SVs and population structure that clearly require the pangenome.

I do have a few major points to address:

a.) On one hand side I appreciate that the authors utilize their newly generated pangenome resources in a slightly different way as done in many other pangenome studies these days. However, I don't really get the point why using anchor genomes would generally (or just in this study? And why then?) be preferable over constructing a "true" pangenome. Is it because of feasibility, your particular data structure (which consists of genomes of fairly different quality and contiguity) or is it beneficial/required to have a single reference to map to for certain analyses? What does it mean that "...anchor genomes achieve similar high quality..." (line 117)? In what context? Compared to what? Is there a reference to cite for this claim?

>>>We have removed the anchor genome statement to eliminate confusion associated with the addition of novel terminology. However, in the pangenome era anchor genomes will play an important role just as reference genomes have enabled genome science to date. An anchor genome can be any genome in the pangenome set that enables pairwise or all-by-all analysis, which is chosen to test a specific hypothesis. For instance, our analysis of the varin region with PanKmer (PK) required the choice of an anchor genome that allowed us to identify the crossover events (**Fig. 5a, b, Supplemental Fig. 34**). This analysis demonstrated the power of both anchor genome choice as well as using a k-mer based pangenome method. Anchor genomes (or references) are still required for complex genomes such as cannabis because even to date building the complete reference-free graph pangenome with PGGB was not possible (**Supp. Fig. 7f**).

We relied on anchor (reference) genomes because of computational feasibility, and because pairwise whole genome alignments are still the standard approach for calling structural variants in plants ¹. We also highlighted a subset of individual high-quality genomes for which we had multiple data types, including gene expression, methylation base calls, and a Y chromosome (e.g. EH23 and Ace High [AH3M]). We featured EH23a/b in particular because it is the highest quality fully phased, haplotype-resolved, chromosome-scale cannabis genome to date. In addition, EH23 has the varin-producing background and importance in the Oregon CBD breeding program, as well as haplotypes from MJ and HC hemp with short day and day neutral photoperiodicity.

We leveraged a variety of approaches to analyze our pangenome, including graph, k-mer, and linear reference-based analyses. For the graph, we included only chromosome-level, haplotype-resolved genomes, while the k-mer based representation with PK allowed us to include the full set of 193 genomes to evaluate relatedness. Our PK pangenome captures DNA sequence diversity in cannabis, especially at the gene level (**Fig. 1c**) ². We found that Minigraph-Cactus, PGGB, and Syri discover a similar modal set of variants (**Supp. Fig. 7a**). For

repeat-rich genomes, mapping is still a challenge, considering that the highest mapping depth with the graph occurs within TEs (**Supplemental Figure 7b-e**). The presence of high levels of heterozygosity also adds complexity to graph construction (**Supplemental Figure 25**)³. Graph construction was extremely memory intensive, which limited our application to subsets of the full pangenome (**Supplemental Figure 7f**).

b.) With the data framework in hand and a reasonably small genome size, wouldn't this be a good opportunity to try to construct a pangenome graph (I'm aware of the technical difficulties associated with...but it could eventually be exploited in some of the analyses)?

>>> While the cannabis genome is modest in size (haploid genome, 750 Mb; diploid 1,500 Mb), it has ~79% TEs, many of which are recent, which means they have not had a chance to diverge and still are quite similar. The fact that the TEs are similar makes graph pangenome analysis a challenge computationally; we now detail this in the supplement (**Supplemental Figure 7; see also “Graph pangenome construction and visualization” in Supplementary Methods**). For these reasons, and our experience building pangenomes from 10 to 1000s of genomes, we built the cannabis pangenome using the PanKmer (PK) tool that enables rapid construction and analysis⁴. From a discovery standpoint we have found k-mer-based pangenomes much more informative for discovery; while this is rapidly changing, as of the writing of this the graph pangenome tools cannot handle hundreds of complex genomes like cannabis (**Supplemental Figure 7f**). The only way we could generate graph pangenomes for cannabis was either to down select to a set of informative genomes or masking the TEs, which still only allowed us to analyze a subset of the 193 genomes. We have spent substantial time and computational resources during this revision to fix these issues but to date there is no solution. Since the point of a pangenome is discovery (not a graph), we find that the PK-based pangenome is flexible, fast, and informative to define the variation important for crop improvement (see the varin section where we use the PK-based pangenome to identify genes associated with the production of minor cannabinoids).

However, we constructed a reference-based graph pangenome including all 78 chromosome-level, haplotype-resolved assemblies, as well as reference-free graphs generated with PanGenome Graph Builder (PGGB). Graph pangenome output files (VCF, HAL, GAF, GBZ, FASTA, GFA, PAF, OG, snarls, stats) are now available at <https://resources.michael.salk.edu> under the heading for “Graph Pangenomes.” Excessive memory requirements challenged our efforts at constructing a reference-free graph pangenome for all chromosome-level samples (**Supplemental Figure 8f**). We opted to focus on representative subsets of the scaffolded genomes that did not overwhelm available resources. We were able to construct a reference-free graph by generating a graph for each chromosome separately for the representative subset of 16 genomes. We found that the graph pangenome did not increase the mapping rate with short read sequencing data from Ren et al., in contrast to what has been found in other pangenome efforts. The average mapping rate for both mapping to the graph pangenome and the EH23a linear reference genome was ~95%.

To address the concerns we added: 1) comparisons of structural variant lengths identified with Syri, MGC, and PGGB; 2) mapping rates to a linear reference genome vs the MGC graph; 3) challenges of pangenome graph construction in cannabis (**Supplemental Figures 4,7,25**) gene visualizations (**Figure 5d, Supplemental Figures 11b,c, 12d, 14d**).

c.) Looking at Figure 1A it appears to me that there are less genes identified on chromosome 4 of haplotype 2. Is there an explanation for this?

>>> EH23a has 3,208 gene models and EH23b has 3,126 gene models on chromosome 4, corresponding to a difference of 82 gene models. Gene model density is represented by color, with warm colors indicating high gene density and cool colors indicating low gene density. The width of each rectangular window is relative to the absence of the CpG motif. We think the graph may look different due to the CpG content difference in chromosome 4b that is likely associated with the telomere, which we have clarified in the figure caption. The full set of ideograms for all haplotype-resolved, chromosome-level assemblies showing variation in these motifs is located here: <https://doi.org/10.25452/figshare.plus.28405079.v1>.

d.) Expression analysis: I don't think Figure 1E is a good way to claim that haplotype EH23a has increased expression over haplotype b. More relevant would be how many genes are higher expressed, and what these genes are. What does the y-axis actually show?

>>> We agree that detailed analysis focusing on the number and functional characteristics of expressed genes provides a stronger basis for interpretation. We updated and streamlined this analysis by focusing on chromosome and tissue-specific differences (**Extended Data Figure 1b, Supplemental Table 3**). We also re-ran salmon with an additional parameter, `--keepDuplicates`, which prevented the removal of some haplotype gene pairs. We highlighted the core circadian clock transcription factor *LATE ELONGATED HYPOCOTYL (LHY)*, which showed haplotype-biased gene expression in foliage tissue. *LHY* also showed population-specific differences in protein sequence entropy (diversity); MJ had higher entropy than other populations (**Extended Data Figure 1c, d**). We chose *LHY* to highlight because we found that the agronomically important circadian, light and flowering time genes had above average F_{st} (**fixation index; Supplemental Table 6**) consistent with their role in other crops⁵. We also showed that GO term enrichment among genes with biased expression was distinct in each of the haplotypes, although in general the GO terms are similar, including stress and light response (**Extended Figure 1e, f**).

Figure 1C: I assume only alleles/genes present in both haplotypes were analysed here...how do you reliably distinguish between identical/near identical orthologs?

>>> Only genes present in both haplotypes were included in this analysis. Gene pairs were identified based on a combination of reciprocal best hits and synteny using blastp and MCScanX. A threshold difference of 5 transcripts per million (TPM) was required for biased expression. We added additional detail to clarify this approach in the Supplementary Methods section under the heading "EH23 haplotype expression analysis."

Figure 1D: this is a bit misleading as it gives the impression that more than the aforementioned 10% of genes are actually affected by any type of haplotype bias.

>>>We updated Fig.1d (now **Extended Figure 1b**) with a stacked bar chart with y-axis explicitly showing the gene count. The majority of the genes have balanced expression but we observe subtle differences between tissues (**Supplemental Table 3**).

Please add more explanations in the figure legends for Figures 1C-D (Y-axis!).

>>> We removed panels C, D, and E from Figure 1. We moved Figure 1D to **Extended Figure 1b** and clarified axis labels.

e.) Selection of lines for the pangenome: I understand that the authors have been limited in their choice by germplasm availability and/or current knowledge of hemp diversity. To get a better idea about the portion of diversity this pangenome actually covers it would be great to show a PCA (or similar) illustrating cannabis genetic diversity and the position of the selected lines within.

>>>Selection of lines for the pangenome project maximized diversity using existing knowledge about sample provenance, phenotypic diversity and breeding potential. Due to the challenges surrounding international sample collection, and limited public germplasm availability, we anticipated that our analysis would likely contain limited representation of Asian cannabis diversity. As we highlighted in our initial submission, our sampling efforts of 144 diverse lines appeared not to have completely circumscribed the *Cannabis* genus (**Fig. 1c**). In our revised submission, we used 31-mers⁶ to better characterize the pangenome samples in the context of global cannabis diversity by comparing our full set of assemblies to a collection of published short read samples from across Asia and Europe--including some samples described as basal and feral drug-type.⁷ In this expanded hierarchical clustering analysis, we found additional support for the distinctiveness of our Asian hemp assemblies, and the publicly available Wild Tibetan assembly (**Supplementary Figure 16, Fig share link**). Additionally, the samples classified as basal or drug feral from Ren et al. 2021 clearly represented additional populations. While the authors of Ren et al. 2021 interpret their findings as evidence for these groups representing "historical escapes from domesticated forms", Gao et al. 2002, who published the wild Tibet assembly, classified their sample as "wild" based on geographic isolation: "The Kyirong Gully is a plateau gorge with an altitude of 1,700–6,000 m located on the south slope of the Himalayas and is very isolated from the outside world". We conclude there is uncertainty regarding wild vs feral samples from Asia, but given the genetic dissimilarity between many of these samples compared to North American and European cultivars, propose to represent largely wild type gene pools. We have updated our text to summarize these findings (**lines 233 - 254**).

f.) The authors state that the pangenome should help shedding light on sex chromosome evolution in cannabis...I find very little on this aspect in the current manuscript.

>>> Originally we planned a second manuscript addressing sex chromosomes with our collaborators. However, we appreciate that there were aspects of the sex chromosomes that would benefit from a pangenome view so we have updated the manuscript with a new sex chromosome expression dataset as well as deeper analysis of the sex chromosomes that provided evidence of unique patterns of genome evolution. Our new analyses include identification of the variable PAR/SDR boundary based on Y-specific k-mers (**Fig. 1b**), distinct patterns of solo:intact LTRs (**Fig. 2d-I**), male-specific gene expression in our Ace High genome (**Extended Data Figure 2**), the presence of syntenic genes (**Supplemental Figure 17**), and structural variants (**Supplemental Figure 18**).

g.) Figure 2D needs more explanation...what is shown with the colors here? Isn't there a more intuitive (clear) way of visualizing this (such as STRUCTURE plots)?

>>>Thank you for the suggestion. We have replaced this figure with an expanded, but simplified relatedness analysis of all pangenome assemblies and diverse short read samples (**Fig. 1g**). We have updated our text based on this analysis and believe it strengthens our hypothesis that wild cannabis persists in Asia.

h.) gene annotation and construction of orthologous groups: I much appreciate the rationale to compensate for missed genes in the gene prediction procedure. However, after considering non- overlapping mappings as unannotated genes, I disagree to simply mark the "corresponding" (how was that determined?) orthogroup as present. Given the mapping criteria, orthology is not guaranteed unless determined by a new OrthoFinder run incorporating the recovered gene model.

>>> The recovered unannotated genes were only used in the collector's curve to showcase diversity present at the genome level (**Fig. 1c**). This was intended to be an evaluation of overall presence/absence variation (PAV), but not necessarily coding potential and functionality. The inclusion of these unannotated genes biased the collector's curve towards less diversity (**Fig. 1c**). The method for assigning unannotated genes to a corresponding orthogroup is described in Supplementary Material, section "Analysis of gene-based pangenome." We took a different, but complimentary, approach toward grouping genes as core vs. dispensable (**Fig. 1e**), which was based on orthogroup membership, and did not include any unannotated regions (see Supplementary Methods section "**Identification of pangenome core and dispensable genes**"). We found that unique and cloud genes occurred at very low frequency, supporting our finding from the collector's curve that we sampled the majority of cannabis gene diversity with our pangenome (**Fig. 1c**).

i.) I'm not convinced about the full comparability (especially for the type of analysis performed here) between the cannabis and the Humulus genomes, as the later is of lower quality and contiguity.

>>> We removed this analysis based on multiple reviewer comments. .

j.) The TE part is interesting but in some parts fairly speculative (esp. line 316-324).

>>> We are pleased that the reviewer found this interesting. We have revised the TE and SV section to provide more support for the relationships between TEs, SVs and genes starting on **line 345**.

k.) Pan-gene analyses: the findings related to both Terpene synthase and disease resistance genes are very incremental (basically just a catalog of genomic positions and some CNV). This analysis could be enhanced by looking at sequence/expression diversification in the pangenome. In addition to the targeted gene analyses, it would be interesting to learn at least something about general gene conservation and variability in the pangenome (aka core/shell/cloud gene portions).

>>> We agree that a larger-scale investigation of pan-gene content was missing from our study, and that the terpene synthase and disease resistance sections were descriptive. We have moved those sections (**Supplemental Figs. 11, 12**) and added a core/shell/cloud analysis (**Fig. 1e, Supplemental Table 4**) as well as highlighting agronomically important circadian, light and flowering time variation (**Extended Data Fig. 1, Supplemental Fig. 14**). Also, the section on the varin and fatty acid biosynthesis pathway highlights both the SV and nucleotide diversity we exposed with the pangenome.

We identified and analyzed core and dispensable gene content of the full pangenome assembly set (**Figure 1e, Supplemental Figure 10**), and also analysed GO term enrichment of core genes (**Supplemental Table 4**). We found GO terms associated with terpene biosynthesis and defense response were strongly enriched among core genes (**Supplemental Table 4**). The most frequent significant GO term among core genes in the pangenome was “sesquiterpene biosynthetic process” (GO:0051762), which was associated with the most highly expressed terpene synthase copies on chromosomes 5, 6, and 9 (EH23a.chr5.v1.g077560.t1, EH23a.chr6.v1.g321150.t1, EH23a.chr9.v1.g282260.t1), and was significant in every genome except for PBBK, a public, previously released genome.

We analyzed sequence variation in orthogroups, including all 193 genomes, to find broad patterns of variation at the population level (**Supplemental Figures 11b-d, 12d, 14**). We discovered that some categories of genes, including agronomically important circadian, light and flowering time genes such as *GIGANTEA*, had substantially different levels of amino acid sequence variation (**Supplemental Figure 14c,d**). For instance, the circadian, light and flowering time genes displayed well above average F_{st} comparing hemp and MJ populations (**Supplemental Table 6**), and we highlight the biased expression of the core circadian clock transcription factor *LHY* (**Extended Figure 1c, d**).

l.) Data Availability: what does “pending” mean? Was the data submitted and accessions numbers assigned?

>>> All data is now publicly available: the NCBI BioProject ID for the cannabis pangenome is PRJNA1140642 (<https://www.ncbi.nlm.nih.gov/bioproject/?term=PRJNA1140642>). The BioProject Accession ID for the sequencing data is PRJNA904266. The BioProject Accession IDs for EH23a and EH23b are PRJNA1111955 and PRJNA1111956, respectively.

m.) I'm not sure the title is the best representation of the main work conducted in this study.

>>> Although cannabis is currently considered a minor drug crop and often viewed with suspicion by recent generations, its historical significance spans social, religious, economic, and agronomic domains. Our pangenome analysis reveals the remarkable extent to which cannabinoids have been subject to intense selection (domestication), while the broader genome reflects the diversity of human selection pressures and adaptations to various needs. While the discovery that cannabinoid synthase exhibits low variability may seem incremental, it was only possible through a comprehensive pangenome approach. More importantly, our most striking finding, relevant not only to cannabis but to other economically significant crops, is the genome's extraordinary dynamism (1-2.5% SNP; >20% SV) and the persistence of this diversity despite the highly conserved cannabinoid synthase organization.

Referee #3 (Remarks to the Author):

In this manuscript, Lynch et al. report their work on developing a cannabis pangenome resource by generating a large number of haplotype resolved-chromosome scale assemblies spanning geographic origin, end use, and cannabinoid type. The authors perform extensive analysis of structural variation and find the region containing the cannabinoid synthase genes to be relatively conserved compared to other regions with large rearrangements or structural variants. Additional features such as transposable elements, resistance genes, and terpene synthase genes are characterized. A GWAS was conducted to provide insight into the genetic architecture of varin production in cannabis.

The manuscript presents original data and insight and the authors have used state-of-the-art approaches. The use of statistics throughout the manuscript appears appropriate and most of the conclusions are well supported. Many of the genomes are not accessible at the time of the review other than the 44 currently available at resources.michael.salk.edu but all code and software links work. The pangenome vastly expands the available genomic resources available in cannabis and will be widely used in the research community. The authors highlight the high level of species variation in cannabis and stress the need for germplasm preservation, especially in Asia, which is the proposed center of origin. Since cannabis is so embedded in human culture, this work will be broadly interesting to wide audience and is especially timely due to the rapidly changing legal landscape.

With that said, I have several suggested improvements to increase the manuscript quality and utility.

>>> We have updated our Data Availability statement with links for all of our genomes, which are hosted on FigShare, NCBI, and/or our lab resource page. We created a data availability

spreadsheet on our Supplemental Table 1 (<https://doi.org/10.6084/m9.figshare.25869319.v2>) that lists the location for all of the datasets that we made available under the tab "DATA_AVAILABILITY."

Main concerns

1. Since this is a pangenome paper, I'd like to see more comparisons across the whole pangenome or at least the 78 haplotype resolved-chromosome scale assemblies. How many core genes, dispensable genes and private genes are there? I'd like to see more statistics about how much more sequence/genes were added to the former reference. Could also add a genespace visualization between the identified subpopulations to showcase major structural variants.

>>> We thank the reviewer for this suggestion about adding an analysis focused on the core, dispensable, and private genes. This is now included as **Figure 1e** and **Supplemental Figure 10**. Additionally, we have included a Genespace visualization showcasing all of the samples with X and Y chromosomes (**Supplemental Figure 17**).

2. Due to the outcrossing nature and high heterogeneity across the cannabis species, do you think having only one sample representing a subpopulation is sufficient to make claims about its relationship to the other groups? For example, pairwise F_{st} in Supplemental Table 3 and several of the PCAs.

>>>We agree with the reviewer that small samples do likely affect pairwise population estimates of F_{st} and overall inferences about population structure. In the case of genome wide F_{st} values in **Supplemental Tables 5** and **6** (with updated numbering), we used the Weir and Cockerham weighted F_{st} estimates, which was developed to help correct for small and uneven sample sizes.⁸ In the main text we have made updates to emphasize focusing on our largest sample sized populations, the MJ and European hemp, which showed significant differentiation (weighted average F_{st} = 0.2). In addition, we have generated a new k-mer based "cannabis tree of life" (see above; <https://doi.org/10.6084/m9.figshare.27883008.v1>) that includes all pangenome assemblies and the most diverse available Asian and European short read libraries, which adds support to our claims about differentiation of some Asian hemp from European hemp, and shows more clearly where remaining undersampled cannabis diversity exists (i.e. China, Tibet, Pakistan, Thailand, South America, and Africa).

3. L 1461 - Why was the proximal gene (next to the MLO marker) used – is there reason to believe this gene is the likely candidate gene for resistance or just linked? Was the observed sequence variation in the pangenome associated with the resistance scores in Supplementary Table 1?

>>>The marker from Mihalyov and Garfinkel (2021) was for a non-MLO type of powdery mildew resistance, PM1. The text has now been revised to describe the gene cluster in EH23a in which the PM1 marker is found (see section "Disease resistance gene analog (RGA) analysis" in Supplemental Material). We do not have reason to believe that the proximal gene

(XM_030647777.1) is the candidate gene for resistance. We assessed pangenome-level variability by mapping XM_030647777.1 to the full set of 193 genomes with minimap2, collected all gene models that overlapped the hit with bedtools intersect, and aligned corresponding proteins with mafft. The resulting tree did not show any clear groupings that corresponded to the resistance scores or populations (see fullPangenomeMLOTree.png at <https://doi.org/10.25452/figshare.plus.28405079.v1>). We also expanded our analysis of XM_030647777.1 to the pangenome by visualizing variation contained in the pangenome graph (**Supplemental Figure 12d**).

4. Including the wild Asian sample makes visualizing the relationship between the other samples difficult to interpret and it would be interesting to see a PCA with it removed. Or run admixture K = 2 - 6 to see how groups drop out.

>>>Based on feedback from multiple reviewers, we have removed the windowed Local PCA that was previously shown in **Fig. 2d**, and added a k-mer based MDS plot to as **Fig. 1g** that includes all pangenome assemblies and the most diverse short read samples from across Asia and Europe. We also have added a high-resolution hierarchical clustering based tree of all these samples to the supplemental materials that includes each sample name (**Supplemental Fig. 16**, <https://doi.org/10.6084/m9.figshare.27883008.v1>). **Supplemental Fig. 15a** also shows a PCA of the scaffolded assemblies, colored by their known use type and provenance. Additionally, we attempted to create unified SNP calls across all assembly types and diverse public short read libraries in order to test various population scenarios with Admixture. Unfortunately, we were unable to confidently complete this process due to technical limitations using a mix of SNP calling methods required for the various sequencing types. Using Admixture to test K = 1 - 10 on only the scaffolded assemblies produced a most likely value of K = 2, delineating the broad split between hemp and drug type samples that has been reported in the literature previously (data not shown). We think this likely reflects the under sampling of Asian and wild cannabis, which we have highlighted in the manuscript conclusions. Ultimately, we believe that neither this large pangenome nor earlier resequencing studies have fully captured the full diversity of cannabis, which is apparent from the results of our unified k-mer analysis (**Supplemental Figs. 8,9**), and collector's curves (**Fig. 1c,d**). This is on one hand surprising for a plant subjected to a century of prohibition and eradication efforts, but also a call for action to study and conserve wild cannabis.

5. Why was the mapping in the F2 done via GWAS versus linkage mapping? Since phenotyping was unreplicated how reliable are the phenotypes? Were all the F2 samples female? You mention phenotyping autoflowering – is this data included? There does not appear to be a corresponding GWAS. Please remove if not used. Do the results emphasize the utility of the pangenome, meaning do they fall in newly discovered regions. What about the other two significant hits on Chr 9 and the X chromosome?

>>>We opted to use the modern Bayesian-information and Linkage-disequilibrium Iteratively Nested Keyway (BLINK) association model, since the major QTL for the varin trait had already been discovered in a prior study, but we suspected that important minor QTL(s) remained unknown. With this model we could test each of the SNPs (93,251) independently, in

conjunction with linkage-disequilibrium criteria⁹, which has been shown to improve rates of both false positives and false negatives under different genetic architectures of simulated and real plant traits—for example in a study of wheat NAM populations¹⁰ and in randomly mated maize populations.¹¹ Therefore, we believe the BLINK association model incorporates linkage parameters of varying distance across the genome much like QTL interval mapping (e.g. EM maximization), but potentially improves precision when combined with high density SNP data. In our initial submission we reported 4 significant hits, but two were in gene-poor repeat-rich regions (chr9 and chrX) and were not discussed in the text. Upon further examination, we realized that these results were produced from the raw non-normalized data (**Supplemental Figure 32**). After reanalyzing with the normalized data (normality is an assumption in our GWA model, **Supplemental Fig. 33b**) we have corrected the results table with three hits on chr4 and one on chr7 (**Supplemental Table 16**). We chose to discuss the SNPs producing phenotypic variance explained (PVE) values above 5% in the GWA model and combined this with our crossover linkage analysis to propose gene candidates: the *BKR* gene (major QTL, PVE: $r = 0.86$, **Supplemental Figure 33**) in addition to the *ALT* gene containing region (PVE: $r = 0.28$). We do think these results demonstrate the utility of the pangenome. The region of chr7 containing *ALT* genes hosts significant nucleotide and structural diversity, discussed in the varin section (**Fig. 5b-d**), which would not have been discovered without using the pangenome. Likewise, we used the high quality assemblies and gene models to identify the fine scale genetic diversity across diverse accessions at the *BKR* locus (**Fig. 5e,f**).

We believe the varin phenotyping is reliable (*BKR* was previously identified in a different population by a separate group providing support that this is a robust dataset and analysis), although each plant was only assayed once for cannabinoids. Since the F2 population ($n = 271$) is derived from a selfed F1, the association of SNPs from commonly inherited haplotypes to varin production is a type of replication. Rare haplotypes or genotyping errors were also removed (min MAF = 0.25), meaning the final genotype to varin associations are statistically significant and replicated in many individuals of the F2 population (**Supplemental Figure 33 c,d**) Although data was collected, we did not conduct a GWAS on the autoflowering data due to several published studies on this trait^{12,13,14}. Autoflowering and varin data is provided in Supplementary Table 1 in the F2_population_phenos tab so that other researchers can make use of this data.

Yes, all of the F2 plants are XX females (often called feminized seed), since ERBxHO40_23 (XX) was self-pollinated using a silver thiosulfate treatment to induce masculinization of select flowers, which then pollinated the remaining female flowers. Some male flower expression was recorded in this population, as is common across feminized seed populations (see "Herm" column in F2_population_phenos sheet).

6. There should be a more discussion of the TPS and R gene results as compared to what was already known in the literature. What is novel?

>>> The extent of copy number variation across chromosomes and populations was previously not known (**Supplemental Figure 11a,12a,b**). We leveraged our new graph pangenome to

assess sequence variation (**Supplemental Figure 11b,c, 12d**). Terpene synthase genes are enriched among “core” genes (Supplemental Table 4). “Core” terpene synthases are among the highest expressed genes (EH23 gene expression). We also found substantial variability (measured by sequence entropy) across the length of the multiple sequence alignment for a core terpene synthase orthogroup (**Supplemental Figure 11d**). We have moved the terpene synthase and R gene discussion to Supplementary Note 1.

7. Please discuss how the pangenome will be maintained and how additional data can be added/supplemented.

>>> Data is publicly available on NCBI, FigShare, Michael lab resources page, and GitHub, and includes many lines from USDA-GRIN that have available seeds. Data is available for download from multiple sources: NCBI and FigShare, as well as the lab resources page. Our graph pangenomes are publicly available on our lab website. New genomes could be incorporated into the graph and k-mer-based pangenomes, and we envision that this will be a dynamic dataset as new tools and additional data become available.

Minor concerns

1. Traditionally when discussing crosses in plant breeding, it is customary to write the female parent first. Therefore, it may make more sense for the masculinized parent to be written second (HO40#23 x ERB)

>>> We recognize that the naming may be inconsistent with traditional plant breeding customs. However, the naming is passed down from early in the project (going back more than 10 years) and it is impractical to rename all records.

2. Supplemental Figure 1 is very important and is too difficult to see both in the manuscript and at the figshare link. Please improve the resolution and explain the red and blue lines.

>>> We have updated this figure in the text and at figshare. We recommend downloading this figure directly from the figshare link, because it is still difficult to fully visualize in the text. We have added a legend to the figure to explain the red and blue lines (female and male lines, respectively). The figure is available here: <https://doi.org/10.6084/m9.figshare.27981758.v1>

3. It is unclear what counts is referring to in Figure 4B and if it is also representing the inversions too.

>>>The count scale bar (purple) in **Fig. 3b** (updated numbering) is only representative of translocations, which we have labeled more clearly. We have added an additional scale bar for the yellow inversions and red duplications.

4. Why do you think chr 6 shows such long distance LD in Fig 4D? Is there anything interesting in this region?

>>>The elevated LD bins of chromosome 6 are surprising, given the larger inversions found on other chromosomes. However chromosomes 6 (and 5) have terpene synthase “hotspots,” where the majority of copies are located in a multi-megabase region near the end of the chromosome. Since terpenes are a known breeding selection target, this could help explain some of the elevated long distance LD. Additionally, from **Supplemental Figure 17c**: “sample GRMb is annotated as an outlier, with 2,104 non-syntenic genes at the beginning of chromosome 6, most of which lack similarity to a known gene, or have similarity to ribosomal or photosynthesis genes,” which could contribute to the chr6 LD outliers.

5. Please expand on the EH23 RNASeq method section on page 26. How many plants were used and define tissue types more clearly.

>>> Please see **Supplemental Table 2**, “Support for gene models,” for a more detailed description of the tissue types used in this analysis from EH23. Briefly, six tissue types were sampled from one plant, including: early flower buds (“Early flower”), main stem leaves (“Foliage”), leaves collected 12 hours after light treatment (“Foliage12”), late-stage flower buds (“Late flower”), primary and secondary root tissues (“Roots”), and apical shoot tips with meristematic regions (“Shoot tips”). Expression data is available from our lab resources page (<https://resources.michael.salk.edu/root/home.html>) under the tab, “Expression Files.”

6. Please add a color key for Figure 2D, change legend from Supplemental Figure 6d) D) to Supplemental Figure 6. D) and add percent variance explained to Figure 2E.

>>> We have modified Fig. 2 based on feedback from multiple reviewers by removing the previous Fig. 2e, shifting it now to **Fig. 1** and adding a new panel (g), which is an MDS of the pangenome samples in the context of a broader global cannabis diversity.

7. Can you add a frequency to each group in Figure 5C?

>>> We have added a count next to each of the cassette tracks in **Fig. 4c (updated numbering)**, corresponding to the number of cassettes across the pangenome that were visually similar to the cassette summary figure.

8. Why was 1kb selected for upstream and downstream analysis of TEs? It would be useful to look at the random distribution of TEs across different genomic features or even other genes to see if there is anything unique/interesting happening in the cannabinoid synthase genes.

>>> The 1 kb flanking window was selected based on the distances presented in **Supplemental Table 10**. Specifically, the average distance between genes and TEs in the 78 chromosome-level, haplotype-resolved genomes is approximately 500 bp, and the average standard deviation for all of the TE categories is almost exactly 1 kb. Based on this, 1 kb was selected as a reasonable flanking window size. **Supplemental Figure 19** also shows a 1 kb flanking window.

Supplemental Table 13 includes enrichment and depletion of TEs at the breakpoints of structural variants across populations. A conclusion from this analysis is that TEs are more prevalent at breakpoints of inversions and inverted translocations in MJ and F1 populations. Hemp does not show patterns of significant enrichment in any structural variant breakpoints. Across all populations, structural variants, and TE types, Mutator DNA TEs show the greatest enrichment in MJ in duplications.

For the trees presented in **Fig. 4e** and **Supplemental Fig. 31**, a flanking distance of 2 kb was selected to maximize the number of genomes presented, since fewer genomes were included in the 1 kb flanking distance analysis (in other words, there were fewer genomes with TEs present at a distance of 1 kb). In particular, at a distance of 1 kb, helitrons were more frequently associated with *THCAS* and *CBCAS*, relative to *CBDAS*. The conclusion is consistent for both 1 kb and 2 kb flanking distances. For comparison, please see the 1 kb distance trees included below (**Response Figs. 1, 2**). Proximal TE sequences were grouped according to full length cannabinoid synthase. Related to the trees: **Supplemental Table 14** presents enrichment of TEs surrounding cannabinoid synthases (2 kb flanking distance); the results in this table provided the motivation for focusing on helitron DNA TEs (that are enriched upstream of *THCAS*).

To further address this comment, we added an analysis to assess the potential enrichment of TEs near genes more generally. Across the pangenome, genes related to transposition, including transcription, recombination, and repair, were most frequently associated with Ty3-LTRs, more than any other type of TE. In contrast, Ty1-LTRs were associated with defense and metabolite biosynthesis (**Supplemental Table 11**).

Response Figure 1. Maximum likelihood tree of helitron sequences flanking (1 kb upstream or downstream) cannabinoid synthases in the 78 scaffolded assemblies (see also Fig. 4e for 2 kb flanking distance).

Response Figure 2. Maximum likelihood tree of Ty1 LTR-RT sequences flanking (1 kb upstream or downstream) cannabinoid synthases in the 78 scaffolded assemblies (see also Supplemental Fig. 31 for 2 kb flanking distance).

9. It seems like Chr 8 should be flipped in Supplemental Fig 6B.

>>>We have corrected this so all chromosomes are consistently oriented with EH23a .

10. What are the apparent outlier samples for Chr 1 and 9 in Supplemental Figure 4.

>>> OFBa.chr1 has a length of 48.4369 Mb. In contrast, OFBb.chr1 is 84.76276 Mb. The other outlier is SN1v3b.chr9, with a length of 40.96772 Mb. This appears to be a combination of natural variation in chromosome length combined with assembly efficiency (now **Supplemental Figure 3c**).

11. The Supplemental Figure 6 link is only to the tree which does not appear to be the same as the one presented in the manuscript and the heatmap is missing.

>>>We have added the high resolution heat map to the Supplemental Figure 6 link, which is now **Supplemental Figure 8**. We have now also added an expanded "cannabis tree of life" (**Supplemental Figure 16**, see above; https://figshare.com/articles/figure/All_cannabis_assembly_Ren_etal_samples_tree/27883008?file=50693184)

12. Define line colors in Supplemental Fig 7B.

>>>We have added a legend for migration weights represented in **Supplemental Fig. 15b**.

13. The chr X label on the y axis in Supplemental Figure 12G and H is confusing. Would help to label all as X and Y or remove completely.

>>>We have replaced the previous dot plots with updated figures showing syntenic regions between X and Y samples (**Supplemental Figure 18**).

14. Can you add which bin the proposed centromere is located in the Supplemental Figure 14.

>>> Location of centromere (see **Fig. 1a, Supplemental Figure 6**) is based on ideos (see Ideos.pdf at <https://doi.org/10.25452/figshare.plus.28405079.v1>). **Supplemental Figure 14** is now **Supplemental Figure 22**.

15. It is hard to understand or visualize where the elevated LD is in Supplemental Figure 18A. Please define the colors in 18B and add more info on the physical position to x axis for chromosome 1.

>>>This figure is now **Supplemental Figure 26**. We have improved the resolution of **Supplemental Figure 26a** and added annotations indicating the inversion location. We also added annotations and physical locations to **Supplemental Figure 26b**.

16. Why are the terpenes shown in different colors in the Supplemental Figure 19B?

>>> The colors corresponded to different groupings based on sequence similarity. The legend showed the correspondence between colors and groupings. We removed this figure and replaced it with pangenome graph visualizations (**Supplemental Figure 11b,c**), as well as a visualization of protein sequence variability (**Supplemental Figure 11d**; see Methods section titled, “Calculation of sequence entropy for DNA and protein sequences”).

17. What is the gene model representing in Supplemental Figure 19D? This is not well described in the text or figure legend.

>>> The gene model in Supplemental Figure 19D (now **Supplemental Figure 12c**) is the proximal gene (XM_030647777.1 from CBDRx; EH23a.chr2.v1.g115410 from reference genome EH23a) near the marker from Mihalyov and Garfinkel (2021) for a non-MLO type of powdery mildew resistance, PM1 (see section “Disease resistance gene analog (RGA) analysis” in Supplemental Material). We also assessed pangenome-level variability by mapping XM_030647777.1 to the full set of 193 genomes with minimap2, collected all gene models that overlapped the hit with bedtools intersect, and aligned corresponding proteins with mafft (see fullPangenomeMLOTree.png at <https://doi.org/10.25452/figshare.plus.28405079.v1>). We have moved our discussion of disease resistance genes to Supplemental Note 1.

18. I believe Supplemental Figure 22 is missing.

>>> This is now **Supplemental Figure 30**. It is available through FigShare (<https://doi.org/10.6084/m9.figshare.25872250>) and is linked in the document.

19. There appears to be an additional supplemental figure between Supplemental Figure 24 and 25.

>>> Thank you for catching this. This is now **Supplemental Figure 34**, which shows the relationship between the high varin HO40 crosses. This figure accompanies **Supplemental Table 15**.

20. Please define a, b, c, and d in Supplemental Table 9.

>>> We have updated the description for **Supplemental Table 14**. Please also see the section titled “Enrichment of transposable elements flanking genomic features” in the Supplementary Materials for a description of the variables. From this section: “Briefly, “X” represents a specific type of TE and “Y” encompasses all TEs. The total number of X located upstream or downstream of a specific genomic feature (for example, cannabinoid synthases) is denoted as “a”; the total number of X located upstream or downstream of all genomic features (for example, all genes) is “b”; the total number of Y located upstream or downstream of a specific genomic feature (cannabinoid synthases) is “c”; and the total number of Y located upstream or downstream of all genomic features (all genes) is “d”. An enrichment score (ES) is defined as $ES = (a/b)/(c/d)$, and the p-value is defined as $p = (a + b)!(c + d)!(a + c)!(b + d)! / (a! b! c! d! N!)$, where N is the sum of a, b, c, and d.”

21. What is the parent 2 column in Supplemental Table 10 referring to?

>>> This is now **Supplemental Table 15**. Please also refer to our response to comment 19. The parent 2 column refers to the second parent in the cross with a plant that was the result of a cross with the original high varin HO40 plant. The other parent ID has now been added to this table.

Referee #4 (Remarks to the Author):

Lynch et al. report a comprehensive pangenome for cultivated cannabis, analyzing 181 new and 12 previously released genomes. They observed significant genetic diversity and propose a new understanding of cannabis population structure and hybridization history. Despite extensive genome-wide variation, they observed low diversity in critical cannabinoid synthase genes. I read this paper with interest and think there are many useful datasets and interesting analyses that will attract a broad readership. The paper could be strengthened for clarity, and many analyses are described in vague terms without detailed numbers or statistics, making it challenging to verify some of the claims. Below I have detailed my concerns which I feel when addressed will significantly strengthen the paper.

1. I am confused by the total number of genomes sequenced and what was included in the pangenome and various downstream analyses. The authors mention that a 'core' of 78 genomes were included in the pangenome, comprising 42 genomes from 21 samples and 24 haploid genomes, which does not sum to 78. It appears that the two distinct haplotypes from each cultivar or accession might be counted as separate genomes within the pangenome? Moreover, a larger pangenome of 193 genome assemblies is referenced throughout the paper, yet it is unclear which analyses pertain to the 78 genomes versus the 193 genomes, aside from their mention in the collector's curve.

>>> We agree this is confusing and could benefit from greater clarity. We updated the text where relevant to state what set of genomes was used for a given analysis. We also added a visual breakdown of the pangenome, showing which data were used in each analysis (**Supplemental Figure 2b,c**).

2. The authors produced haplotype phased genome using both triobinning and HiC. Both of these approaches should produce more or less the same result, it is unclear why trio binning was performed, given the increased cost and complexity required to have parental genomes and their F1 sequenced.

>>> Since we sequenced through the OCBF breeding program to identify specific traits selected in these populations we had trios (**see the pedigree Supplemental Figure 1 and section on varin analysis**) and therefore, we did not generate Hi-C for these and used the parental information instead to phase these genomes. For example, the CCS reads from CALO and FB191 were used as parental reads to trio-bin the COFB genome.

3. Broadly, some of the analyses seem superficial or lack specific details. I recognize the authors may be aiming for brevity for the Nature format, but more details would be helpful. For instance, the transposable element analyses describe TE abundance in the EH23 hybrid genome in detail, but differences in TE dynamics across the pangenome are only briefly mentioned. There are several TE analyses between Cannabis and Humulus and the authors use these differences to explain genome evolution, but these species diverged ~25 million years ago, making it difficult to compare centromere and TE differences and link them to the biology of these species.

>>> We agree about the importance of making sound comparisons between species. We removed this analysis to improve the focus of our manuscript, and revised the TE and SV sections to provide more detailed analyses about the connections between TEs, SVs, genes and populations.

Similarly, the authors describe that structural variants are highly variable across the pangenome, but provide no statistics of how many, the average size, or how much of each genome is conserved in the anchor or dispensable. Figure 4 shows some really interesting results, but I do not think it is adequately described in the results. They claim that the number of inversions is similar to cross-species comparisons, but it would be great to have more details or data to support this. I do not think it would take up much extra space to add these values and any relevant statistical test, but it would significantly strengthen the claims of this paper.

>>>We updated the SV section to include more detailed statistics about inversion frequencies and sizes, and have more clearly referenced our detailed analyses in **Supplementary Fig. 24** about SVs. In terms of the statement that the percent of inversions for cannabis falls within the range found in cross-species comparisons, we are citing ¹⁵, who compared genomes from 32 pairs of species, each within a genus (but not necessarily sister species). They found a range of values for species pairs from ~1% (Eucalyptus) to ~37% (Fragaria). Our values for cannabis ranged from 0% (from a close relative of EH23a reference lineage) to 7.6% (TWSVa) (**Supplementary Fig. 24c**). A recent major finding is that inversions are not correlated with nucleotide identity, and do not accumulate in a clock-like manner ¹⁵, thus making the overlap of the cannabis genome interesting compared to these species pairs as a hypothesis generating statistic, but not definitive for taxonomic delineation, especially given that within species ranges for this metric have not been published to date for other taxa (cannabis may be the first). We have also added **Supplemental Figure 25**, which provides a detailed breakdown of structural variants across each genome.

In the Terpene Synthesis section, the authors state that chromosomes 5 and 6 have terpene synthase hotspots but provide no details. How many are there? What is the copy number variation like across the genome? How does this link to the biology? Similarly, the authors describe that they annotated disease resistance genes, but provide no statistics or descriptions of how they vary across the pan-genome, simply that they developed a map of R genes and state that various diseases are emerging in Cannabis.

>>> In **Supplemental Figure 11a** and **Supplemental Figure 12a,b** we showed chromosome-level copy number variation for terpene synthases and R genes, respectively. We have also added pan-gene analysis and found terpene biosynthesis and defense response genes are significantly enriched among core genes (**Supplemental Table 4**). The most frequent significant GO term among core genes in the pangenome is “sesquiterpene biosynthetic process” (GO:0051762), which is associated with the most highly expressed terpene synthase copies on chromosomes 5, 6, and 9 (EH23a.chr5.v1.g077560.t1, EH23a.chr6.v1.g321150.t1, EH23a.chr9.v1.g282260.t1), and is significant in every genome except for PBBK, a public, previously released genome. In order to improve the focus of the main text, we have now moved the description of R genes and terpene synthases to **Supplemental Note 1**.

For the cannabinoid pathway section, I have a few questions. What does ‘numerous pseudogenized paralogs’ refer to? How many, how are pseudogenes being classified vs real genes, and how do these vary from THC producing and not producing accessions? Can the expression data be leveraged here to test for pseudogenization or subfunctionalization?

>>> We have revised the main text to more accurately reflect our analysis methodology in this section (**line 508**). The clusters of pseudogenes and functional synthases are visualized in **Fig. 4c** and **4d**, and are now more clearly described in the **Fig. 4** caption as well. The methods to classify full length and partial length synthases (based on BLAST alignment bit scores and gaps) are described in detail in the “Synthase cassette analysis” section of the Supplementary Material. Additionally, the synthase phylogeny (**Fig. 4b**) was filtered using ORFinder to remove partial synthase matches with premature stop codons. We also checked the full length synthases described above by remapping them to our EH23a and EH23b assemblies, and found them to overlap with gene models (e.g. THCA synthase: EH23a.chr7.v1.g200100). In terms of how these genes and pseudogenes are arranged across THC producing (Type I) and the various other chemotypes, **Fig. 4d** shows the contrast between Type I and III (also visible in the Type II haplotypes). This graph shows how each assembly contains only one *THCAS* or *CBDAS*, that the locations of the functional synthases are not usually identical, and how the arrangement of pseudogenes around the synthase varies. We have updated the text to emphasize the findings more clearly (**lines 494-516**).

The authors claim that CBCAS paralogs appear not to be under strong selection. What evidence is there to support this? Numerous analyses could be done here to support this including Ka/Ks across the pangenome, or various selective sweep analyses using population data.

>>> The first reason for this claim is that while *CBCAS* genes were present in 56% (110/193) of genomes, CBCA is rarely produced in the breeding program samples of the pangenome (data not shown). Additionally, it is known from the literature that few if any cannabis plants produce CBCA, as discussed in **lines 521-524**. We also added text about RNA expression data that showed only low expression of the *CBCAS* genes in EH23a. Furthermore, we have included several new genome wide analyses to bolster this section and other areas of the manuscript: population level F_{st} and XP-CLR tests (<https://doi.org/10.6084/m9.figshare.28049363>), and

orthogroup entropy calculations (**Supplemental Fig. 14a-c**), which did not find any significant signals within cannabinoid synthases. Finally, we used Fisher's Exact Test of Neutrality (K_a/K_s based) on alignments for the *CBCAS* genes, which did not produce any significant P-values, meaning we cannot reject the null hypothesis of strict-neutrality for these loci based on our dataset (data not shown). Overall, we revised the *CBCAS* paragraph to more accurately reflect what is known and not known about CBCA in cannabis.

4. Line 260. Figure 2e seems to be a PCA or presumably the whole genome, but the results suggest that there are different relatedness based on different regions of the genome (e.g., chromosomes 2, 5, 7, and 9 are similar, the sex chromosomes show a different pattern). It seems that only a single value is plotted for each accession in this PCA. To better demonstrate the genetic structure suggested by the PCA, the authors might consider conducting population structure or admixture analyses, which could help verify the presence of the five distinct subpopulations illustrated in Figure 2C. From the PCA alone, distinctions among these subpopulations are not readily apparent, except possibly in the case of the EU hemp group. Additionally, the 'hc hemp' and 'feral' groups appear to be underrepresented in the genomic data, which might affect the clarity of subgroup distinctions.

>>>We have removed this analysis (**Fig. 2e**) due to difficulty interpreting the results. We also clarified this section of the text by adding an inline reference to the **Supplemental Figure 15a**, which is a PCA (whole genome SNPs) of the scaffolded assembly coded by population assignment. In **Supplemental Figure 15a** we can see that hc_hemp, feral and F1 samples are indeed largely contained within the European hemp and MJ clusters that are split across the X axis. The Asian hemp sample (YunMa) is set apart from these groups by its location on the Y-axis. Additionally, using these same full genome SNPs, an Admixture analysis produced a most likely K value of 2 (data not shown). The claim of "Pangenome uncovers at least five populations" stems from the use of k-mers to bridge analyses across the various assembly types, shown in simplified form in **Fig. 1f**, and full form for all 193 assemblies in **Supplemental Figure 8**. Unfortunately, we were unable to derive unified SNP calls for all assembly types and short read libraries, which limits the type of analyses we can perform at this time (i.e. Structure or Admixture). Instead, we have increased the genetic breadth of our k-mer clustering with new **Fig. 1g** that includes a broader representation of European and Asia short read samples. We think this provides a more clear overview of cannabis diversity and sampling. Overall, cannabis diversity has not been fully sampled, therefore inferences about population structure remain tentative, but this pangenome can serve to help guide a more complete ascertainment of cannabis diversity in the near future, especially through sampling and preserving Asian germplasm.

5. The 25 Mb inversion in the Y specific region of the hemp sample Golden Redwood is quite interesting. I think this paper would benefit from a detailed analysis of the sex chromosomes in the pan genome of Cannabis. Generally, there are few pan-genome studies with species containing sex chromosomes (and perhaps none for plants?). This is talked about briefly throughout the paper, but one to two paragraphs discussing SVs, TE activity, gene differences in

the non recombining regions compared to the rest of the genome would be of broad interest to readers.

>>> As mentioned above, we were coordinating with a collaborator on a distinct manuscript on the Y chromosomes, but based on your comments and those of the other reviewers we also agree a pangenome perspective would greatly increase the impact of this manuscript. Therefore, we have updated the manuscript with a detailed analysis of the X and Y chromosomes as well as a new sex-specific expression dataset. As addressed in a different response, we identified the PAR/SDR boundary based on Y chromosome-specific k-mers (**Fig. 1b, Supplemental Figure 18**), providing a granular delineation of the recombining and non-recombining regions and proposing hypotheses about the evolution of this region. Additionally, we observed distinct patterns of solo:intact LTRs (**Fig. 2d-I**), particularly among Ty1 LTRs on chromosome Y, and discussed this in terms of expansion and degeneration of the SDR. The dramatic spike in the solo:intact LTR ratio is a result of the increased abundance of solo LTRs, as opposed to a decrease in the number of intact LTRs. **Supplemental Figs. 17 and 18** also show regions of structural variation in the X and Y chromosomes.

Minor:

1. How much heterozygosity does a typical Cannabis cultivar/accession contain? Is EH23 a representative accession in this context or does it have higher diversity than most cultivars?

>>> Cannabis is generally described as heterozygous due to its outcrossing nature and hybridization history. While the range of heterozygosity varies in the literature based on methods and samples, we found in our assemblies that heterozygosity of SNPs range from about 1% - 2.5% (**Supplementary Fig. 4**). The EH23 sample is on the lower end ~1% heterozygous, despite the parents of this hybrid being selected for contrasting cannabinoid and flowering time traits. This makes sense since both the parents of the hybrid are generally of drug-type origin, while sample H3S7, an F1 cross between a fiber hemp and MJ sample (Hercules x Santhica-27), shows the highest heterozygosity in our assemblies (2.5%). Therefore, we find EH23 to be representative of the typical drug-type hybrids, although wider, more diverse crosses between hemp, drug-type and wild populations are possible, as we suggest in our conclusions. In our revised manuscript we now also report heterozygosity across a range of SV types (**Supplemental Fig. 25**), which average 20.6% of the genome length (including non-alignable regions). Using this metric, the EH23 sample is slightly below average at 18.8% heterozygous.

2. What is the X axis of Figure 3A showing? Is this percentage of the genome? Similarly for Figure 3B and C, does this show the age of different TEs across the pan genome or just EH23?

>>> Yes, **Figure 2a** is showing the percent of each genome that is covered by TEs, grouped by population ID. The y-axis is a Gaussian kernel density estimation (KDE). **Figures 2a-c** include all genomes (not just EH23), which we have clarified in the figure caption.

3. Line 216, how were the genomes annotated? This is described in the results, but one sentence here would be great, especially because genome annotation methods are quite contentious in the pan genome world.

>>> We have added a sentence about gene annotations in the main text (**line 160**). We have also expanded the methods section (“Gene and repeat prediction”) in the Supplementary Material. Finally, we have added more detail to the section, “Support for gene models,” (**Supplemental Table 2**) which includes the samples used for gene prediction and expression analyses.

4. Line 228, what does this mean? It would be helpful to report the average pairwise F_{st} or some averages for the 5 groups/populations to provide more context here. The F_{st} is presumably quite high,

>>>We have updated the text in this sentence and have provided a supplementary table with further details on all population pairwise comparisons of average genome wide F_{st} (**Supplemental Table 5**), but see also our above response to reviewer #3 about sample size and F_{st} . Additionally, we have added some discussion and supplementary material about F_{st} outliers (**lines 220-231**).

5. Line 378. Is there any literature supporting this claim that 20 Mb inversion may function as supergenes in plants through overdominance?

>>>Although, the term 'supergene' is used less frequently in the plant literature, we explored this hypothesis based on findings from sunflower species.¹⁶ Similar to Todesco et al. 2020, we found abundant megabase scale inversions in cannabis, but unlike in the multi-species sunflower comparison, we did not find the same level or extent of elevated LD in the inverted regions, as discussed in **lines 467-469**. We have added a reference to **line 457** that reviews the theory and empirical evidence for the idea that inversions can function to form supergenes, often in the context of balancing selection and associative overdominance¹⁷. Interestingly this concept generally overlaps with the evolution of sex chromosomes, which are thought to sometimes originate from inversions.

References:

1. Jayakodi, M. *et al.* Structural variation in the pangenome of wild and domesticated barley. *Nature* **636**, 654–662 (2024).
2. Sherman, R. M. & Salzberg, S. L. Pan-genomics in the human genome era. *Nat. Rev. Genet.* **21**, 243–254 (2020).
3. Kaur, H., Shannon, L. M. & Samac, D. A. A stepwise guide for pangenome development in crop plants: an alfalfa (*Medicago sativa*) case study. *BMC Genomics* **25**, 1022 (2024).

4. Aylward, A. J., Petrus, S., Mamerto, A., Hartwick, N. T. & Michael, T. P. PanKmer: k-mer-based and reference-free pangenome analysis. *Bioinformatics* **39**, (2023).
5. Steed, G., Ramirez, D. C., Hannah, M. A. & Webb, A. A. R. Chronoculture, harnessing the circadian clock to improve crop yield and sustainability. *Science* **372**, (2021).
6. Titus Brown, C. & Irber, L. sourmash: a library for MinHash sketching of DNA. *J. Open Source Softw.* **1**, 27 (2016).
7. Ren, G. *et al.* Large-scale whole-genome resequencing unravels the domestication history of. *Sci Adv* **7**, (2021).
8. Meirmans, P. G. & Hedrick, P. W. Assessing population structure: F(ST) and related measures. *Mol Ecol Resour* **11**, 5–18 (2011).
9. Huang, M., Liu, X., Zhou, Y., Summers, R. M. & Zhang, Z. BLINK: a package for the next level of genome-wide association studies with both individuals and markers in the millions. *Gigascience* **8**, (2019).
10. Sandhu, K. S., Burke, A. B., Merrick, L. F., Pumphrey, M. O. & Carter, A. H. Comparing performances of different statistical models and multiple threshold methods in a nested association mapping population of wheat. *Front. Plant Sci.* **15**, 1460353 (2024).
11. Gyawali, A., Shrestha, V., Guill, K. E., Flint-Garcia, S. & Beissinger, T. M. Single-plant GWAS coupled with bulk segregant analysis allows rapid identification and corroboration of plant-height candidate SNPs. *BMC Plant Biol* **19**, 412 (2019).
12. Garfinkel, A. R. *et al.* Genetic Mapping of SNP Markers and Candidate Genes Associated with Day-Neutral Flowering in *Cannabis sativa* L. *bioRxiv* 2023.04.17.537043 (2023) doi:10.1101/2023.04.17.537043.
13. Leckie, K. M. *et al.* Loss of daylength sensitivity by splice site mutation in *Cannabis* pseudo-response regulator. *Plant J.* **118**, 2020–2036 (2024).
14. Toth, J. A., Stack, G. M., Carlson, C. H. & Smart, L. B. Identification and mapping of major-effect flowering time loci *Autoflower1* and *Early1* in *Cannabis sativa* L. *Front. Plant*

- Sci.* **13**, 991680 (2022).
15. Hirabayashi, K. & Owens, G. L. The rate of chromosomal inversion fixation in plant genomes is highly variable. *Evolution* **77**, 1117–1130 (2023).
 16. Todesco, M. *et al.* Massive haplotypes underlie ecotypic differentiation in sunflowers. *Nature* **584**, 602–607 (2020).
 17. Jay, P. *et al.* Supergene Evolution Triggered by the Introgression of a Chromosomal Inversion. *Curr Biol* **28**, 1839–1845.e3 (2018).

Referees' comments:

Referee #2:

The authors addressed most of my concerns in this revised manuscript. I especially appreciate the inclusion, comparison and discussion of graph representations of the cannabis pangenome. I believe this will be helpful for other researchers working in this area. Removing the anchor genome statement is fine: although I know, understand and appreciate the concept and use of an anchor genome here, I found its introduction quite confusing in the first version.

Thank you for your helpful suggestions. We also think that they improved the manuscript. We decided to leave the “anchor genome” concept out as not to introduce new concepts.

The only aspect I still find problematic is related to gene annotation: the authors state in their response that the gene set excluding the unannotated regions was used for grouping core vs dispensable genes. This is the type of analysis that would profit from a highly accurate and comparable gene annotation, that also contains the genes/regions which might have escaped the initial gene predictions. In any case this should be made transparent so that users can navigate the data sets accordingly.

We agree that highly accurate gene models are beneficial for analyzing the pangenome landscape. We analyzed the pangenes with multiple methods because we have found that gene model construction is an iterative and imperfect process. We added a clarifying sentence to the Supplementary Methods under the section heading, “Identification of pangenome core and dispensable genes”:

“This analysis was restricted to high confidence gene models predicted with the TSEBRA pipeline. In contrast, the collector’s curve analysis of gene content also included unannotated genome regions lacking gene model predictions, but with similarity to known genes, as a way to capture unsampled diversity (Fig. 1c, d; Supplementary Figure 4; see also section “Analysis of gene-based pangenome”).”

Our gene-based collector’s curve plateaued at ~100-125 genomes, signifying that cannabis gene diversity was well sampled. With the TSEBRA gene models alone, more than 75% of genes were present in core (~23%) or nearly-core (~55%) groups. The overall fraction of unique (0.7%) and cloud (0.4%) genes was only ~1% of the pangenome universe. Taken together, our analysis of pangenes captured a comprehensive set of genes and “gene-like” content.

Referee #3:

The authors have addressed my main concerns in this revised manuscript. In particular, they have added additional information on the core, dispensable and private genes, improved figures, made all data publicly available, and clarified questions regarding selected methods. The addition of the sex chromosome assembly data is very interesting and will offer insight into

sex related traits in cannabis and sex chromosome evaluation more broadly. I have a few minor comments that may further help the clarity of the manuscript.

Thank you.

Minor comments:

Line 274 – Need period after Stack et al. 2025.

Corrected.

Figure 1B – what is the significance of the color (blue and red) for the X, Y and triangles? Also listing the Y-specific SDR as 79-84 Mb is confusing since the text says it spans 81 of the 110 Mb at Line 310). Also please explain what is meant by the X-specific region at 53-55 Mb?

We have added more detail in the figure legend explaining the red = X homologs and blue = Y homologs and the tip triangles = collapsed monophyletic clades of X or Y homologs. The X specific region is the fraction of the X chr that does not undergo recombination with the Y chr (although it undergoes recombination in XX females).

What is the explanation for the increase in number in of genes for contig-level genomes versus haplotype phased genomes in Figure 1E?

The contig assemblies have residual, uncollapsed haplotypes resulting in an increased number of genes. We have updated the figure legend to make this clear as well as Extended Data Figure 4, which shows the number of genes scaled with the presence of duplications detected by BUSCO.

In Figure 3C and 3D, it may be more meaningful to look at the 5 identified subgroups (Figure 1F) separately for LD as it is likely they have different decay rates.

While we appreciate this suggestion, we do not want to replace the current breakdown of LD decay by chromosomes, which support results from other aspects of the manuscript (X chr and chr1).

Need to add new authors to author contribution section and likely update acknowledgements/funding.

We have updated the author contributions with all new authors and also added funding for the new authors.

Check ordering of chromosomes in Extended Data Fig 1B.

This figure is now Extended Data Figure 9. Panel A of this figure is all chromosomes grouped by sex and tissue. Panel B is grouped by chromosome. Panels C-F are specific

to X and Y chromosomes, and the x-axis is the gene position along the length of either the X or Y chromosome (in Mb).

Please list color key in tube map for Supplementary Figure 14D as in Fig. 5D.

We added the legend to this figure.

Supplementary Figure 18A – Please use more colors or clearly label the samples for the X and Y sample genotypes.

We updated the figure legend and labeled each genome track separately.

Referee #4:

The authors have addressed my previous concerns and I feel the revised manuscript is significantly improved in quality and clarity.

We appreciate your time and feedback.